# The transcription factor Zeb1 controls homeostasis and function of type 1 conventional dendritic cells

Yan Wang[1,5], Quan Zhang[2,3,5], Tingting He[1], Yechen Wang[1], Tianqi Lu[1], Zengge Wang[1], Yiyi Wang[1], Shen Lin[3], Kang Yang[1], Xinming Wang[1], Jun Xie[1], Ying Zhou[2,3], Yazhen Hong[1], Wen-Hsien Liu[1], Kairui Mao[1], Shih-Chin Cheng[1], Xin Chen[1], Qiyuan Li[2,3,4] ✉ & Nengming Xiao[1] ✉

Type 1 conventional dendritic cells (cDC1) are the most efficient cross-presenting cells that induce protective cytotoxic T cell response. However, the regulation of their homeostasis and function is incompletely understood. Here we observe a selective reduction of splenic cDC1 accompanied by excessive cell death in mice with *Zeb1* deficiency in dendritic cells, rendering the mice more resistant to Listeria infection. Additionally, cDC1 from other sources of *Zeb1*-deficient mice display impaired cross-presentation of exogenous antigens, compromising antitumor CD8+ T cell responses. Mechanistically, Zeb1 represses the expression of microRNA-96/182 that target *Cybb* mRNA of NADPH oxidase Nox2, and consequently facilitates reactive-oxygen-species-dependent rupture of phagosomal membrane to allow antigen export to the cytosol. Cybb re-expression in *Zeb1*-deficient cDC1 fully restores the defective cross-presentation while microRNA-96/182 overexpression in *Zeb1*-sufficient cDC1 inhibits cross-presentation. Therefore, our results identify a Zeb1-microRNA-96/182-Cybb pathway that controls cross-presentation in cDC1 and uncover an essential role of Zeb1 in cDC1 homeostasis.

Dendritic cells play a crucial role in bridging innate and adaptive immunity by priming and activating T cells as professional antigen-presenting cells (APC)[1,2]. Since their discovery by Ralph Steinman & Zanvil Cohn in 1973, two cell types have been classified as members of dendritic cell (DC) family including conventional DC (cDC) and plasmacytoid DC (pDC), whereas Langerhans cells (LC) and monocyte-derived DC (moDC) are clearly distinct from DC based on ontogeny and gene expression profiles[1,2]. Both cDC and pDC arise from common dendritic cell progenitor (CDP), although pDC also develop from a lymphoid progenitor[2,3]. pDC are best characterized by the unique capacity to rapidly produce large amounts of type I interferons upon viral infection, but they tend to be less efficient in presenting antigens

to T cell than cDC[4]. cDC comprise two functionally distinct subsets, cDC1 and cDC2. Transcription factors Irf8, Batf3, Nfil3 and Id2 are required for the development of cDC1[3,5]. Among these, Irf8 is the master lineage-determining transcription factor. Id2 is mutually repressed by Zeb2, and Zeb2, in turn, is repressed by Nfil3. The Nfil3-Zeb2-Id2 transcription factor regulatory circuit switches the E-protein-dependent +41 kb *Irf8* enhancer in dendritic cell progenitors to Batf3-dependent +32 kb *Irf8* enhancer in mature cDC1[6–8]. Notch2 and Klf4 have been shown to be required for the development of two functionally distinct cDC2 subsets[9,10]. EBI2 and CD97 selectively control splenic cDC2 homeostasis by positioning or retaining them in the spleen, respectively[11,12], while the lymphotoxin-β receptor promotes

[1]State Key Laboratory of Cellular Stress Biology, Innovation Center for Cell Signaling Network, School of Life Sciences, Faculty of Medicine and Life Sciences, Xiamen University, Xiamen, Fujian 361102, China. [2]National Institute for Data Science in Health and Medicine, Xiamen University, Fujian 361102, China. [3]School of Medicine, Xiamen University, Xiamen, Fujian 361102, China. [4]Department of Hematology, The First Affiliated Hospital of Xiamen University, Xiamen 361003, China. [5]These authors contributed equally: Yan Wang, Quan Zhang. ✉e-mail: qiyuan.li@xmu.edu.cn; nengming@xmu.edu.cn

splenic cDC2 proliferation through Rel-B[13,14]. However, it is unclear how the homeostasis of splenic cDC1 is regulated.

The capability to present exogenous antigens on major histocompatibility complex class I molecules (MHC-I)[15], a process called cross-presentation, is crucial for induction of protective cytotoxic T cell (CTL) response against tumor and virus, even when DC are not infected directly[16,17]. To date, two major pathways of antigen cross-presentation have been proposed to explain how MHC-I molecules are loaded by peptides derived from extracellular sources. In the vacuolar pathway, the degradation of internalized antigens by lysosomal proteases such as Cathepsin S and loading of processed peptides onto MHC-I occur entirely within endocytic or phagocytic compartments[18,19]. In the cytosolic or phagosome-to-cytosol (P2C) pathway, internalized antigens from phagosomes have to be translocated into the cytoplasm where they undergo proteasomal degradation. The derived peptides are subsequently transported by TAP into the endoplasmic reticulum (ER) or back into the phagosomes to be loaded onto MHC-I molecules[18,19]. Most studies reported that the P2C pathway contributes more to cross-presentation of cell-associated antigens than does the vacuolar pathway.

How internalized antigens gain access to the cytosol remains elusive. However, available evidence suggest that ER-associated degradation (ERAD) machinery selectively transport polypeptide substrates across phagosomal membrane[20,21]. Alternatively, the antigens may passively leak from phagosomal compartments through membrane disruption caused by phagosomal reactive oxygen species (ROS) produced by NADPH oxidase Nox2[22,23]. The extent to which extent each mechanism contributes to the export of different types of antigens is uncertain. cDC1 are generally considered the most efficient cross-presenting cells due to their intracellular biological specialization, including adaptation in trafficking and low activity of lysosomal proteases, which preserve antigenic information. In addition to membrane rupture, ROS produced by Nox2 also causes phagosomal alkalinization and inhibits lysosomal proteases[24]. However, how the activity of Nox2 is regulated is not completely understood.

Published datasets of Assay for Transposase-Accessible Chromatin with high-throughput sequencing (ATAC-seq) reveal that the motifs for zinc-finger E-box binding homeobox 1 (Zeb1) are more highly associated with increased chromatin accessibility in cDC1 than in cDC2. This points to a potential role for Zeb1 in regulating some aspects of cDC1 and prompt us to intensively investigate the precise role of Zeb1 in the development and function of DC.

We show here that mice with *Zeb1* deficiency in dendritic cells are more resistant to infection with *Listeria monocytogenes*, because of a selective reduction in splenic cDC1 population associated with excessive cell death. On the other hand, cDC1 from other origins of *Zeb1*-deficient mice display impaired capability to present exogenous antigens, resulting in defective antitumor CD8$^+$ T cell response. The loss of Zeb1 in DC unleashes the expression of microRNAs miR-96 and miR-182, which, in turn, suppresses the expression of *Cybb* that encodes Nox2 subunit gp91$^{phox}$. Consequently, phagosomal ROS-dependent rupture of phagosomal membrane is diminished, leading to impaired antigen export to the cytosol. Our data identify a Zeb1-miR-96/182-Cybb regulatory axis that controls cross-presentation in cDC1 and uncover an essential role of Zeb1 in cDC1 homeostasis and function.

## Results

### Selective reduction of splenic cDC1 in DC-specific *Zeb1*-deficient mice

The Zeb protein family of transcription factors consist of two members Zeb1 and Zeb2, which are best known to drive epithelial to mesenchymal transition (EMT) by repressing epithelial genes[25]. Zeb1 and Zeb2 are widely expressed in murine myeloid immune cells, but the expression of Zeb1 in cDC1 is higher than that of Zeb2[26,27]. It has been demonstrated that Zeb2 switches the DC fate specification from cDC1

to pDC or cDC2 by antagonizing Id2 expression[28–30]. The high association of Zeb1 binding motif with increased chromatin accessibility in cDC1 suggests a potential role for Zeb1 in regulating some aspects of cDC1 function[31]. However, further investigations are needed to examine the precise role of Zeb1 in dendritic cells. To address this, we generated mice with floxed *Zeb1* alleles using a gene targeting strategy, in which exon 4 was flanked by *LoxP* sites, and then crossed them with mice harboring a transgene encoding Cre recombinase under the control of the *Itgax* promoter (CD11c-cre) to generate DC-specific Zeb1 conditional knockout mice (*Zeb1*$^{fl/fl}$ CD11c-cre, called Zeb1-dcKO hereinafter) (Supplementary Fig. 1a). Immunoblot analysis confirmed specific and efficient deletion of Zeb1 in CD11c$^+$ dendritic cells, but not in T cells and B cells (Supplementary Fig. 1b). Flow cytometric analysis showed a threefold decrease in the frequencies of cDC1 populations in the spleen of Zeb1-dcKO mice compared to those in wild-type (WT) littermates (*Zeb1*$^{fl/fl}$ or *Zeb1*$^{+/+}$ CD11c-cre), with marked reductions in the absolute numbers of both cDC1 and cDC2 due to lower abundance of myeloid cells in the spleen of Zeb1-dcKO mice (Fig. 1a, b and Supplementary Fig. 1c). However, the proportions and absolute numbers of cDC1 and cDC2 in other lymphoid tissues (peripheral lymph node (pLN) and thymus) and non-lymphoid tissues (liver and lung) were not affected by the absence of Zeb1 (Fig. 1a–d). The frequencies of pDC were similar in the lymphoid tissues of WT and Zeb1-dcKO mice, nevertheless, the numbers of pDC were dramatically decreased in the spleen of Zeb1-dcKO mice (Fig. 1e, f). All other myeloid cells were normally present in the spleen of Zeb1-dcKO mice, with the exception of a small decrease of neutrophils and inflammatory monocytes due to lower abundance of total splenic myeloid cells (Supplementary Fig. 1c–e). Alveolar macrophages (AM), distinct from cDC, are the prototypical macrophages in the lung that express high levels of the integrin CD11c and the lectin Siglec-F. Zeb1-dcKO mice showed comparable numbers of AM in both bronchoalveolar lavage (BAL) fluid and lung with wild-type counterparts (Fig. S1f, g). Zeb1-dcKO mice had an increased frequency of CD8 single positive thymocytes, which was also observed in Zeb1 mutant *Cellophane* mice[32], but possessed normal frequencies and numbers of mature T cells in the second lymphoid organs (Supplementary Fig. 1h, i). This selective reduction of splenic cDC1 was further confirmed by uniform manifold approximation and projection (UMAP) analysis (Fig. 1g) and kernel density estimation using the plot_density function (Supplementary Fig. 1j and Supplementary Data 1) with single-cell RNA sequencing (scRNA-seq) data of the splenic cDC samples from WT and Zeb1-dcKO mice. Taken together, these data suggested that *Zeb1* deficiency in DC selectively reduced the cDC1 population in the spleen but not in the other lymphoid and non-lymphoid tissues.

We next investigated the stage at which Zeb1 regulates cDC1 development by analyzing the pre-DC compartment in the bone marrow (BM) and spleen. Pre-DC give rise to cDC and are derived from common DC progenitors (CDP) in the BM. It has been reported that commitment to the cDC1 and cDC2 lineage is already apparent as Siglec-H$^-$Ly6C$^-$ pre-cDC develop exclusively into cDC1 whereas Siglec-H$^-$Ly6C$^+$ pre-cDC preferentially give rise to cDC2[33]. We observed no significant difference in CD135$^+$ pre-DC and their sub-populations distinguished by Siglec-H and Ly6C expression in the BM of WT and Zeb1-dcKO mice (Supplementary Fig. 1k, l). Although the frequencies of Siglec-H$^-$Ly6C$^-$ pre-cDC1 was only mildly decreased in the spleen of Zeb1-dcKO mice, the absolute numbers of splenic pre-DC and pre-cDC1 were notably reduced due to lower total numbers of splenic myeloid cells (Supplementary Fig. 1m, n). To investigate further whether absence of Zeb1 affected the development of DC progenitors into cDC1, we then examined in vitro development of BM cells from WT and Zeb1-dcKO mice upon stimulation with Flt3L, which generates bona fide counterparts of the splenic DC subsets[34]. In this context, WT and Zeb1-dcKO BM precursors exhibited similar capacity to develop into pDC, cDC1 and cDC2 (Supplementary Fig. 1o, p), suggesting that Zeb1

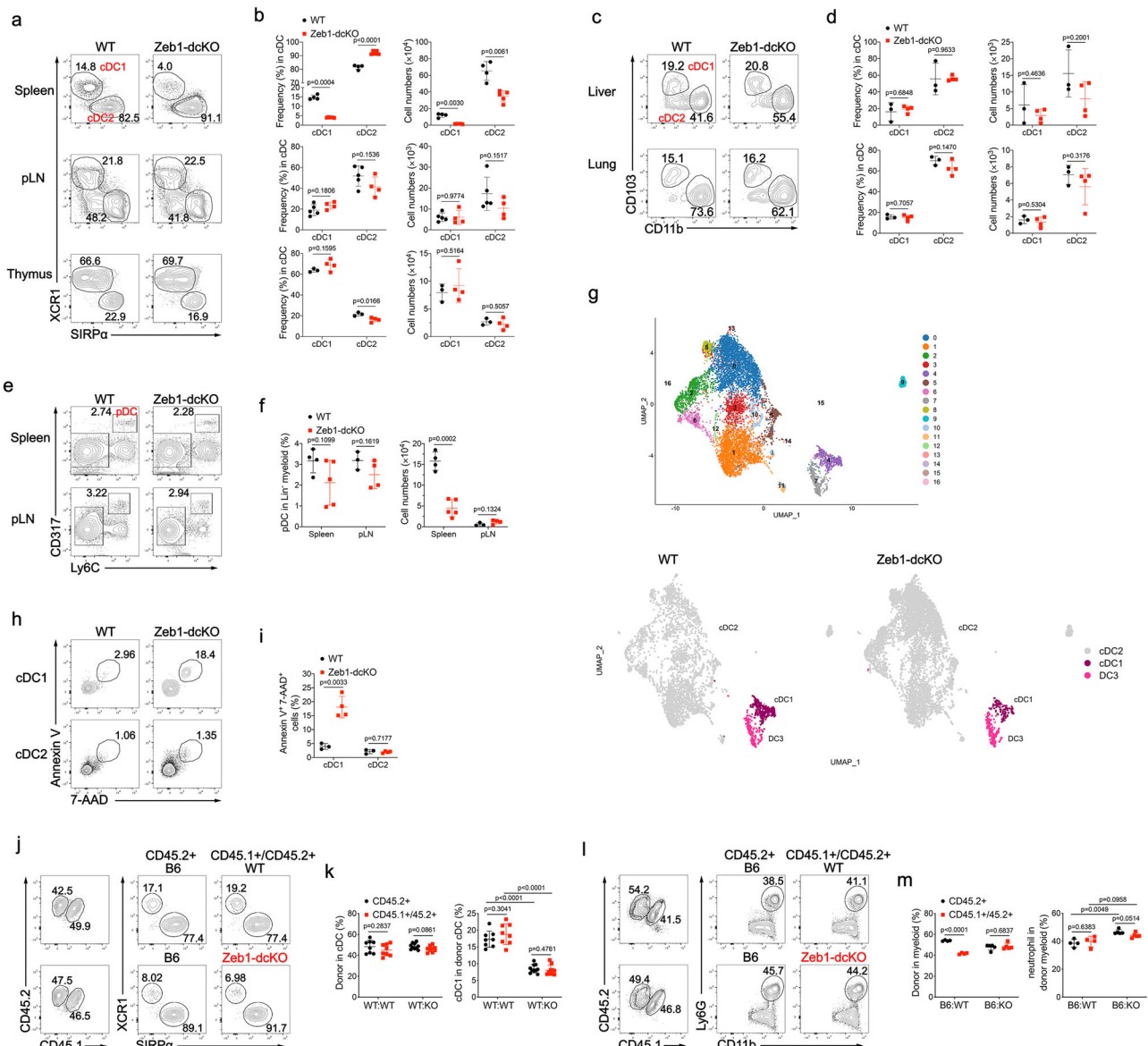

**Fig. 1 | DC-specific ablation of Zeb1 selectively reduced splenic cDC1. a** Flow cytometry of cDC in spleen (top row), pLN (middle row) and thymus (bottom row) from WT and Zeb1-dcKO mice. **b** Frequencies and numbers of XCR1⁺SIRPα⁻ cDC1 and XCR1⁻SIRPα⁺ cDC2 among cDC in spleen, pLN, thymus from mice as in **a** (spleen: WT, n = 4; Zeb1-dcKO, n = 5; pLN: WT, n = 5; Zeb1-dcKO, n = 4; thymus: WT, n = 3; Zeb1-dcKO, n = 4). **c** Flow cytometry of cDC in liver (top row) and lung (bottom row) from WT and Zeb1-dcKO mice. **d** Frequencies and numbers of CD103⁺CD11b⁻ cDC1 and CD103⁻CD11b⁺ cDC2 among cDC in liver, lung from mice as in **c** (WT, n = 3; Zeb1-dcKO, n = 4). **e** Flow cytometry of Lin⁻ myeloid cells (Lin⁻=CD3⁻CD19⁻ hereinafter) in spleen (top row) and pLN (bottom row) from WT and Zeb1-dcKO mice. **f** Frequency and number of CD317⁺Ly6C⁺ pDC among Lin⁻ myeloid cells from mice as in **e** (spleen, WT, n = 4; Zeb1-dcKO, n = 4; pLN, WT, n = 3; Zeb1-dcKO, n = 4). **g** UMAP plot showing unsupervised clustering of total splenic cDC (top row) and the population individually (bottom row) from WT and

Zeb1-dcKO mice by scRNA-seq analysis. **h** Flow cytometry of splenic cDC1 (top row) and cDC2 (bottom row) from WT and Zeb1-dcKO mice. **i** Frequencies of dying (Annexin V⁺ 7-AAD⁺) splenic cDC1 and cDC2 among total cDC1 or cDC2 from mice as in **h** (WT, n = 3; Zeb1-dcKO, n = 4). **j** Flow cytometry of splenic cDC from mixed BM chimeric mice. **k** Frequencies of donor cDC (among total cDC) and cDC1 (among donor cDC) in spleen from mixed BM chimeric mice as in **j** (B6:WT, n = 8; B6:KO, n = 10). **l** Flow cytometry of CD11b⁺ cells in spleen of mixed BM chimeric mice. **m** Frequencies of donor CD11b⁺ cells (among total CD11b⁺ cells) and CD11b⁺Ly6G⁺ neutrophils cells (among donor CD11b⁺ cells) in spleen of mixed BM chimeric mice as in l (B6:WT, n = 4; B6:KO, n = 5). Each symbol represents an individual mouse, small horizontal lines indicate the mean (± s.d.). Data are representative of two (**h–m**) or three (**a–f**) independent experiments. Data are presented as mean ± s.d. Statistical analysis was performed using two-tailed unpaired Student's t-test. Source data are provided as a Source Data file.

may control the homeostasis of splenic cDC1 rather than their development. To explore how Zeb1 regulated the homeostasis of splenic cDC1, we performed single-cell sequencing data analysis using the R package Seurat (v3.0). Feature plots revealed that residual Zeb1-deficient splenic cDC1 maintained their cDC1 identity by expressing similar levels of cDC1 and cDC2 signature genes, including *Xcr1, Clec9a, Irf8, Sirpa, CD209a, Irf4*, to that of WT cDC1. However, *CD8a* expression was decreased, and *Itgae* (encoding CD103) expression was increased

in *Zeb1*-deficient splenic cDC1 (Supplementary Fig. 2a). Nevertheless, there were 106 genes up-regulated and 49 genes down-regulated in *Zeb1*-deficient cDC1 (Supplementary Fig. 2b). Gene ontology (GO) analysis revealed that in addition to pathways of tight junction and cell adhesion, pathways of TNF signaling, apoptosis and necroptosis were also enriched in these differentially expressed genes (DEGs) (Supplementary Fig. 2c). We next investigated whether the depletion of splenic cDC1 in Zeb1-dcKO mice might be caused by impaired survival or

proliferation. By staining fresh isolated splenocytes with AnnexinV and 7-aminoactinomycin D (7-AAD), we detected a dramatically increased percentage of dying cells (AnnexinV⁺ 7-AAD⁺) in splenic cDC1 but not splenic cDC2 from Zeb1-dcKO mice (Fig. 1h, i). In contrast, the percentage of either apoptotic (active caspase-3⁺) or proliferating (Ki67⁺) splenic cDC1 was not affected by *Zeb1* deficiency (Supplementary Fig. 2d–g). Taken together, these results suggested that Zeb1 was required for the homeostasis of splenic cDC1 by maintaining their survival and that loss of Zeb1 led to excessive non-apoptotic cell death in splenic cDC1.

We next investigated whether the reduction of splenic cDC1 in Zeb1-dcKO mice was due to cell-intrinsic factors or external environmental factors. To do so, we generated mixed bone marrow chimeric mice, by reconstituting sub-lethally irradiated B6.SJL (CD45.1⁺) mice with a mixture of C57BL/6 J (B6) (CD45.2⁺) BM cells and either WT (CD45.1⁺CD45.2⁺) or Zeb1-dcKO (CD45.1⁺CD45.2⁺) BM cells at a ratio of 1:1. Eight weeks later, in this competitive environment, Zeb1-dcKO (CD45.1⁺CD45.2⁺) BM cells reconstituted lower percentage of cDC1 in the spleen of mixed bone marrow chimeric mice, similar to that in non-chimeric Zeb1-dcKO mice (Fig. 1j, k). Unexpectedly, B6 (CD45.2⁺) BM cells and Zeb1-dcKO BM cells in the same recipient generated comparably fewer splenic cDC1, whereas B6 (CD45.2⁺) BM cells together with WT (CD45.1⁺CD45.2⁺) BM cells in the same host developed into normal frequencies of splenic cDC1 (Fig. 1j, k). Deletion of Zeb1 in CD11c⁺ cells did not affect the generation of neutrophils, which do not express CD11c, from both genotypes of BM cells in spleens of mixed

bone marrow chimeric mice (Fig. 1l, m). The reduced generation of splenic cDC1 from B6 BM cells in the same recipient with Zeb1-dcKO BM cells was probably associated with cell death triggered by phagocytosis of dying or dead *Zeb1*-deficient cDC1, as *Zeb1*-sufficient splenic cDC1 from B6 mice underwent much more cell death when co-cultured with apoptotic or necroptotic splenocytes than with live splenocytes (Supplementary Fig. 2h, i). Together, these results suggested that the effect of Zeb1 deletion on splenic cDC1 was cell-intrinsic and even dominant, as it also affected *Zeb1*-sufficient splenic cDC1 but did not affect other myeloid cells, including cDC2 and neutrophils, in the same host.

### Resistance of DC-specific *Zeb1*-deficient mice to Listeria infection

Previous studies have established that CD8α⁺ dendritic cells (cDC1) are the obligate cellular entry points for productive infection by intracellular bacterium *L. monocytogenes*. Lack of cDC1 enhances host resistance to the infection due to the loss of access into periarterial lymphocyte sheath (PALS)[35,36]. To examine the host defense of Zeb1-dcKO mice against listeria infection, we infected WT and Zeb1-dcKO mice intravenously with *L. monocytogenes*. All WT mice succumbed to infection with a high dose ($1 \times 10^6$ CFU) of *L. monocytogenes*, whereas almost all Zeb1-dcKO mice survived (Fig. 2a). There were 100 ~ 1000 times fewer CFU of bacteria in the spleen and liver of Zeb1-dcKO mice than in WT littermates at day 2 after infection with a low dose ($2.5 \times 10^4$ CFU) of *L. monocytogenes* (Fig. 2b). The decreased listeria burden

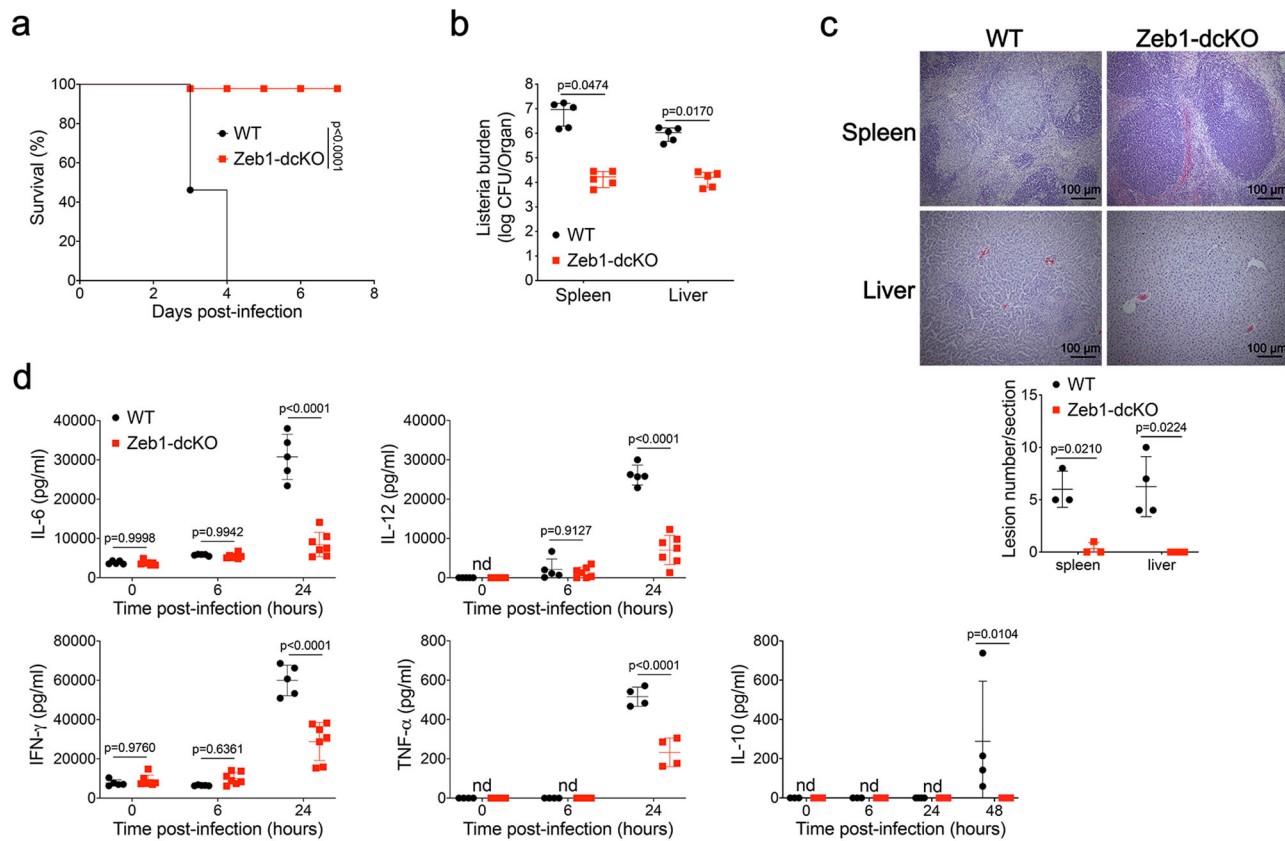

**Fig. 2 | DC-specific Zeb1-deficient mice are resistant to Listeria infection.**
**a** Survival of WT and Zeb1-dcKO mice infected intravenously (i.v.) with $1 \times 10^6$ CFU of *L. monocytogenes* (*n* = 10). **b** Listeria CFUs per spleen and liver of WT and Zeb1-dcKO mice at day 2 after i.v. infection with $2.5 \times 10^4$ CFU of *L. monocytogenes* (*n* = 5). **c** Histopathology (H&E) of spleen and liver from WT and Zeb1-dcKO mice at day 2 after i.v. infection with $2.5 \times 10^4$ CFU of *L. monocytogenes* (scale bars: 100 μm). Numbers of lesions in the sections were enumerated and shown at the bottom. **d** ELISA of cytokines secreted to serum of WT and Zeb1-dcKO mice at indicated

time points after i.v. infection with $1 \times 10^6$ CFU of *L. monocytogenes* (WT, *n* = 5; Zeb1-dcKO, *n* = 7; For IL-10, *n* = 4). Each symbol represents an individual mouse, small horizontal lines indicate the mean (± s.d.). Data are representative of three independent experiments. Data are presented as mean ± s.d. nd: not detected. Statistical analysis was performed using two-tailed unpaired Student's *t*-test (**b**) or two-way ANOVA with Sidak's multiple comparisons test (**d**), or log-rank (Mantel–Cox) test of survival curve (**a**). Source data are provided as a Source Data file.

observed in Zeb1-dcKO mice was in complete accordance with the dramatically reduced mortality. Furthermore, histological analysis showed that Zeb1-dcKO mice displayed substantially reduced lesions in the spleen and liver compared to WT littermates at day 2 after infection. Micro-abscesses with depletion of apoptotic lymphocytes or necroptotic hepatocytes and infiltration by inflammatory cells were observed in white pulp of spleen or near vessels of liver from WT mice but not from Zeb1-dcKO mice (Fig. 2c). Collectively, these data suggested loss of Zeb1 in DC enhanced host defense against Listeria infection.

To investigate the innate immune responses of Zeb1-dcKO mice to Listeria infection, we measured the secretion of inflammatory cytokines into serum of mice within 24 h post infection using enzyme-linked immunosorbent assay (ELISA). As anticipated, Zeb1-dcKO mice showed a severe reduction in secretion of all tested serum cytokines, including IL-6, IL-12, TNF-α, IFN-γ and the anti-inflammatory cytokine IL-10 (Fig. 2d). We then sought to determine whether the impaired production of inflammatory cytokines was due to the decreased Listeria burden or defective innate signaling in dendritic cells. To address this issue, we challenged purified cDC, or subpopulations cDC1 and cDC2 from Flt3L-cultured BM cells with different Toll-like receptor ligands, including the TLR4 ligand LPS, TLR9 ligand CpG-B DNA, TLR1/2 ligand Pam3CSK4, TLR3 ligand Poly(I:C), heat-killed *L. monocytogenes* (HKLM), and measured cytokine production by intracellular staining or by ELISA. In both experiments, *Zeb1*-deficient Flt3L-cDC or sub-populations cDC1 and cDC2 produced similar amounts of all tested cytokines, including IL-12, IL-6, and TNF-α, compared to WT counterparts (Supplementary Fig. 3a–e). Taken together, these results strongly suggested that the increased resistance to Listeria infection of Zeb1-dcKO mice was likely due to the impaired transmission of Listeria into PALS, resulted from selective reduction of splenic cDC1, as the innate immune response was intact in *Zeb1*-deficient cDC.

## Ablation of Zeb1 in DC dampens antitumor immunity

A previous study has suggested that *L. monocytogenes* may utilize the cross-presentation pathways of cDC1 for the productive infection[36]. Although the decreased Listeria burden in the liver can be attributed to the severe reduction of splenic cDC1 in Zeb1-dcKO mice, it is necessary to examine the capability of cross-presentation in residual cDC1 in these mice. Accumulating evidence has established that cDC1 are particularly adept at uptake of dead tumor cells and at cross-priming tumor-specific CD8+ T cells within tumor microenvironment or after migration to tumor draining lymph nodes (tumor dLNs)[37,38]. To examine the role of Zeb1 in the function of cDC1, we challenged WT and Zeb1-dcKO mice subcutaneously with $2 \times 10^5$ B16F10 melanoma cells. B16F10 tumors grew much faster and caused shorter host survival in Zeb1-dcKO mice than in WT littermates (Fig. 3a, b). Similar to unchallenged Zeb1-dcKO mice, tumor-bearing Zeb1-dcKO mice also had decreased frequencies and numbers of splenic cDC1, as compared with tumor-bearing WT mice (Supplementary Fig. 4a, b). By contrast, tumors from WT and Zeb1-dcKO mice displayed a comparable composition of cDC1 and cDC2 (Fig. 3c, d). Likewise, tumor dLNs from Zeb1-dcKO mice contained similar numbers of resident and migratory cDC1 compared to that from WT mice, but had slightly fewer resident cDC2 (Fig. 3e, f). Collectively, these data suggested that Zeb1 in DC was required for control of tumor growth, although the loss of Zeb1 in DC did not affect tumor infiltration of cDC1 despite selective reduction of splenic cDC1.

Consistent with the increased tumor growth, Zeb1-dcKO mice showed a marked decrease in the recruitment and activation of effector CD8+ T cells and a mild reduction in numbers of effector CD4+ T cells in B16F10 tumors despite normal cDC1 infiltration (Fig. 3g, h). Similarly, the frequencies and numbers of IFN-γ-, Granzyme B- and perforin-producing CD8+ T cells were dramatically reduced in B16F10 tumors from Zeb1-dcKO mice, while the frequencies and numbers of

IFN-γ-producing CD4+ T cells were unaffected (Fig. 3I, j). Exhausted (PD-1+Tim3+) CD8+ T cells resulting from chronic antigen stimulation were barely generated in B16F10 tumors from Zeb1-dcKO mice (Supplementary Fig. 4c, d). Therefore, the deletion of Zeb1 in DC resulted in a decrease in both the number and effector function of CD8+ T cells in tumors (Fig. 3g–j), which was unlikely due to regulatory T (Treg) cells, as the numbers of tumor-infiltrating Treg cells in WT and Zeb1-dcKO mice were comparable (Supplementary Fig. 4e, f). The numbers of splenic CD4+ and CD8+ T cells were slightly but significantly declined in tumor-bearing Zeb1-dcKO mice, consistent with the decreased numbers of splenic cDC1 (Supplementary Fig. 4g, h). In contrast, the CD4+ and CD8+ T cells in tumor dLN remained poorly activated, and their numbers were comparable between WT and Zeb1-dcKO tumor-bearing mice (Supplementary Fig. 4i, j). These results suggested that either cDC1-mediated priming of tumor infiltrating CD8+ T cells did not occur at tumor dLN, or most of activated CD8+ T cells in tumor dLN migrated to the tumor site. To exclude the influence of deficits in splenic cDC1 numbers on the antitumor T cell response in Zeb1-dcKO mice, we performed splenectomy before tumor inoculation. Interestingly, splenectomy slightly inhibited B16F10 melanoma growth in both WT and Zeb1-dcKO mice, consistent with an early study (Supplementary Fig. 5a)[39]. However, splenectomised Zeb1-dcKO mice still had compromised tumor control compared to splenectomised WT mice. Moreover, splenectomy neither abrogated the difference of antitumor T cell response between WT and Zeb1-dcKO mice, nor hindered infiltration of cDC1 into tumors (Supplementary Fig. 5a–g). Taken together, these data demonstrated that Zeb1 expression in DC was required for the recruitment and activation of tumor infiltrating CD8+ T cells, and strongly suggested that loss of Zeb1 in DC may attenuate cross-presentation of tumor antigens to CD8+ T cells by cDC1 at tumor sites or tumor dLNs.

## Deletion of Zeb1 abrogates the presentation of exogenous antigen by cDC1

Given that Zeb1 expression in DC was required for activation of tumor infiltrating CD8+ T cells, we next investigated the role of Zeb1 in cross-presentation of cell-associated antigens. To address this issue, we transferred carboxyfluorescein diacetate succinimidyl ester (CFSE)-labeled OT-I T cells to B6 recipient mice followed by immunization with irradiated ovalbumin (OVA)-loaded $β2m^{-/-}$ splenocytes. On day 3 after immunization, OT-I T cells showed much stronger proliferation in both spleen and pLNs when transferred into WT mice than when transferred into Zeb1-dcKO littermates (Fig. 4a, b). Since loss of Zeb1 in DC caused reduction of cDC1 only in the spleen but not in the pLNs, these data could not exclude the possibility that in vivo cross-presentation of cell-associated antigens by cDC1 from pLNs was attenuated by the absence of Zeb1. To directly evaluate the effect of *Zeb1* deficiency on cross-presentation in cDC1 from LNs, we purified cDC1 from pLNs and mesenteric LNs (mLNs) of WT and Zeb1-dcKO mice and test their capability to cross-present bacteria-associated antigen in the form of heat-killed *L. monocytogenes* expressing ovalbumin (HKLM-OVA) to OT-I T cells in vitro. cDC1 from pLNs and mLNs of WT mice induced strong T cell proliferation by cross presentation of HKLM-OVA in a dose-dependent manner. However, cDC1 from pLNs and mLNs of Zeb1-dcKO mice induced significantly less T cell proliferation, especially at a low dose of bacteria (Fig. 4c, d). These results confirmed that Zeb1 promoted cross-presentation of cell-associated and bacteria-associated antigens in cDC1.

To gain comprehensive insight into the role of Zeb1 in antigen presentation, we sorted cDC1 and cDC2 from BM cells cultured with Flt3L and performed in vitro antigen presentation assays. Compared to WT Flt3L-cDC1, *Zeb1*-deficient Flt3L-cDC1 exhibited a striking defect in cross-presentation of both cell-associated and bacteria-associated antigens that were presented by neither WT nor *Zeb1*-deficient Flt3L-cDC2 (Fig. 4e–h). This defect of cross-presentation was unrelated to the

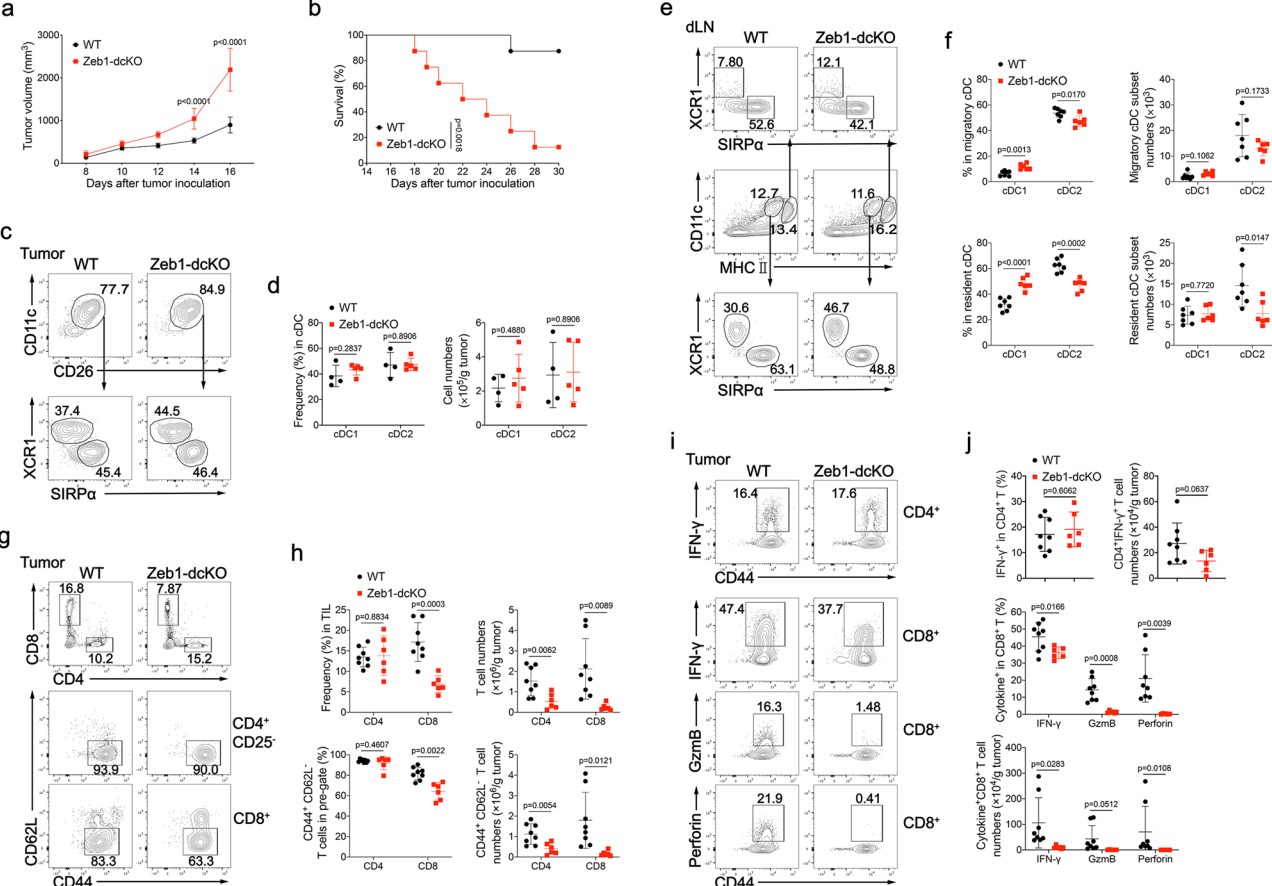

**Fig. 3 | DC-specific deletion of Zeb1 impairs antitumor immunity.** Tumor growth (**a**) and survival curves (**b**) of WT and Zeb1-dcKO mice injected subcutaneously (s.c.) with $2 \times 10^5$ B16F10 melanoma cells ($n = 8$). **c** Flow cytometry of Lin$^-$CD45$^+$CD64$^-$F4/80$^-$MHC II $^+$ myeloid cells was used to identify cDC1 (XCR1$^+$SIRPα$^-$) and cDC2 (XCR1$^-$SIRPα$^+$) subsets in tumor from WT and Zeb1-dcKO mice at day 12 after s.c. injection with $2 \times 10^5$ B16F10 melanoma cells. **d** Frequencies and numbers of cDC1 and cDC2 among CD11c$^+$CD26$^+$ cDC in tumors of WT and Zeb1-dcKO mice as in **c** (WT, $n = 4$; Zeb1-dcKO, $n = 5$). **e** Flow cytometry of live Lin$^-$Ly6C$^-$ myeloid cells was used to identify migratory and resident cDC1 and cDC2 in tumor dLN of mice as in **c**. **f** Frequencies and numbers of cDC1 and cDC2 among migratory or resident cDC in tumor dLN of mice as in **c** (WT, $n = 7$; Zeb1-dcKO, $n = 6$). **g** Flow cytometry of tumor infiltrating leukocytes (TILs) (top row), TIL CD4$^+$ T cells (middle row), TIL CD8$^+$ T cells (bottom row) in tumor from mice as in **c**. **h** Frequencies and numbers of TIL CD4$^+$ or CD8$^+$ T cells (among TILs) (top row) and CD44$^+$CD62L$^-$ CD4$^+$ (among TIL CD4$^+$ T cells) or CD44$^+$CD62L$^-$ CD8$^+$ (among TIL CD8$^+$ T cells) activated T cells (bottom row) in tumor from mice as in **c** (WT, $n = 8$; Zeb1-dcKO, $n = 6$). **i** Flow cytometry of TIL CD4$^+$ T cells (Top row), TIL CD8$^+$ T cells (bottom rows) in tumor from mice as in **c**. **j** Frequencies and numbers of IFN-γ-producing CD4$^+$ T cells (among TIL CD4$^+$ T cells) and of IFN-γ-, Granzyme B- and Perforin-producing CD8$^+$ T cells (among TIL CD8$^+$ T cells) in tumor from mice as in **c** (WT, $n = 8$; Zeb1-dcKO, $n = 6$). Each symbol represents an individual mouse, small horizontal lines indicate the mean (±s.d.). Data are representative of three independent experiments. Data are presented as mean ± s.d. Statistical analysis was performed using two-tailed unpaired Student's $t$-test (**d**, **f**, **h**, **j**), or two-way ANOVA with Sidak's multiple comparisons test (**a**), or log-rank (Mantel–Cox) test of survival curve (**b**). Source data are provided as a Source Data file.

survival of cDC1, because the deletion of Zeb1 did not enhance the cell death of cDC1 during cross-presentation (Supplementary Fig. 6a, b). Furthermore, *Zeb1*-deficient cDC1 showed a partially reduced efficiency for cross presentation of soluble OVA compared to WT cDC1, whereas WT and *Zeb1*-deficient cDC2 equally cross-presented soluble OVA (Fig. 4I, j). In addition, the direct presentation of processed antigen SIINFEKL peptide to OT-I T cells was equally efficient in WT and *Zeb1*-deficient cDC1, as well as in both genotypes of cDC2, suggesting that loading of processed antigen onto MHC-I was functionally normal in *Zeb1*-deficient cDC (Fig. 4k, l). The defect of antigen presentation in the absence of Zeb1 was not limited to cross-presentation to OT-I T cells by cDC1. The presentation of soluble and bacteria-associated antigen through MHC-II to OT-II T cells was also dramatically decreased in *Zeb1*-deficient cDC1 (Supplementary Fig. 6c, d), while the presentation of these antigens as well as cell-associated antigen through MHC-II was intact in *Zeb1*-deficient cDC2 (Supplementary Fig. 6e–g). Collectively, these results suggested that Zeb1 expression in DC was required not only for cross-presentation of exogenous antigens to CD8$^+$ T cells but also for presentation of these antigens to CD4$^+$ T cells by cDC1.

## Zeb1 orchestrates transcriptional program that favors cross-presentation

To explore the mechanism by which Zeb1 regulated cross-presentation, we performed high throughput RNA sequencing (RNA-seq) of WT and *Zeb1*-deficient Flt3L-cDC1 at steady state and after stimulation with HKLM-OVA. Principal component analysis revealed that HKLM-OVA stimulation enlarged the transcriptomic diversification in WT and *Zeb1*-deficient cDC1, compared to the steady state (Supplementary Fig. 7a). A comparison of WT and *Zeb1*-deficient cDC1 transcriptomes identified 246 genes activated and 364 genes repressed by Zeb1 at steady state (Supplementary Fig. 7b and Supplementary Data 2), whereas identified 1167 genes activated and 1084 genes repressed by Zeb1 after stimulation (Fig. 5a and Supplementary Data 3). Functional enrichment analysis of these DEGs revealed that the most enriched Kyoto Encyclopedia of Genes and Genomes (KEGG) pathways including cell adhesion molecules, tight junction and focal adhesion, were overactivated in steady *Zeb1*-deficient cDC1 relative to WT cDC1 (Supplementary Fig. 7c), consistent with the well-established function of Zeb1 in repressing epithelial genes[25]. The pathways of antigen

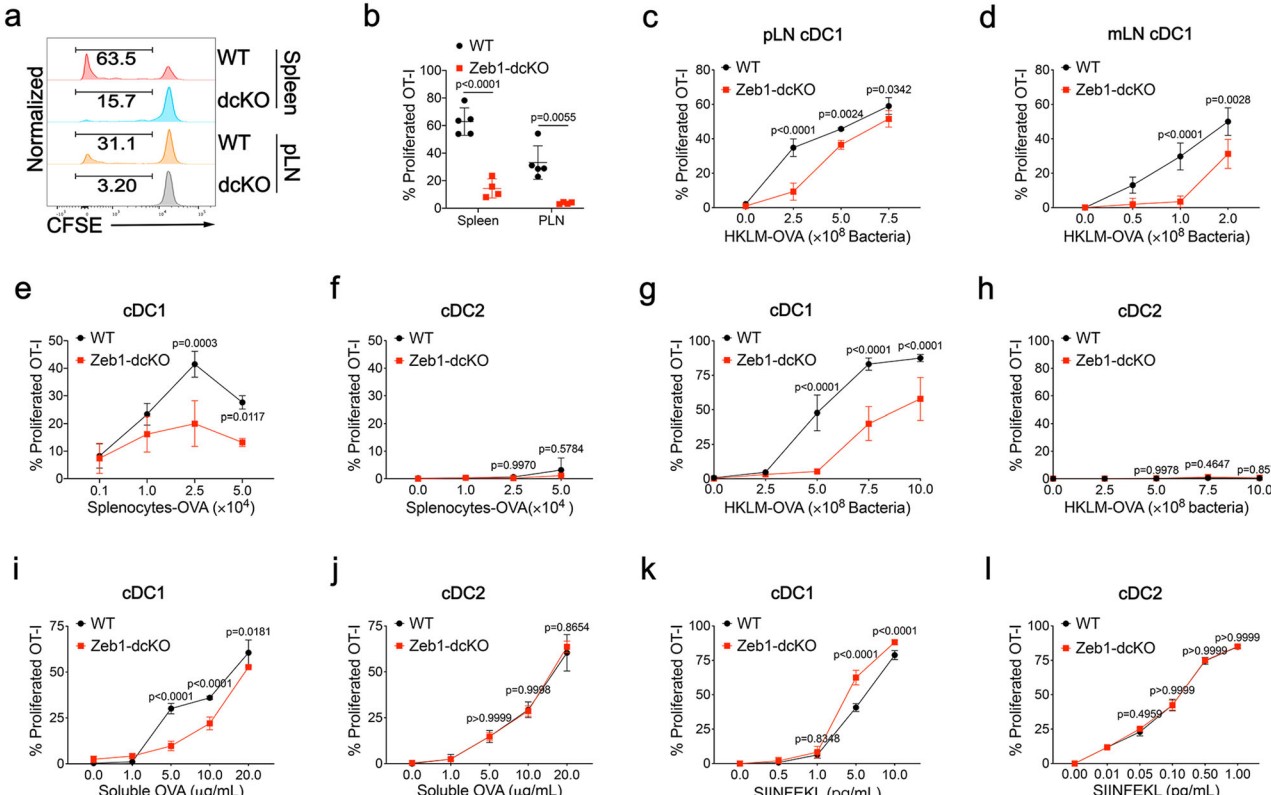

**Fig. 4 | Zeb1 is selectively required for cross-presentation by cDC1. a** Flow cytometry of OT-I T cells from WT and Zeb1-dcKO mice receiving adoptive transfer of $5 \times 10^5$ CFSE-labeled OT-I T cells, at day 3 after i.v. immunization of $5 \times 10^5$ irradiated ovalbumin (OVA)-loaded $\beta2m^{-/-}$ splenocytes. **b** Frequency of CFSE⁻CD44⁺ OT-I T cells among total OT-I T cells from mice as in **a** (WT, $n = 5$; Zeb1-dcKO, $n = 4$). Each symbol represents an individual mouse, small horizontal lines indicate the mean (± s.d.). cDC1 sorted from pLNs (**c**) or mLNs (**d**) of WT and Zeb1-dcKO mice were cultured for 3 days with CFSE-labeled OT-I T cells and different dose of HKLM-OVA, and assayed for OT-I proliferation and activation (CFSE⁻CD44⁺). WT and Zeb1-deficient Flt3L-cDC1 (**e**) or Flt3L-cDC2 (**f**) were cultured and analyzed as described in **c** and **d**, with different dose of irradiated OVA-loaded $\beta2m^{-/-}$ splenocytes as

antigens. Flt3L-cDC1 (**g**) or Flt3L-cDC2 (**h**) of both genotypes were cultured and analyzed as described in **c** and **d**, with various dose of HKLM-OVA as antigens. Flt3L-cDC1 (**i**) or Flt3L-cDC2 (**j**) of both genotypes were cultured and analyzed as described in **c** and **d**, with various dose of soluble OVA as antigens. Flt3L-cDC1 (**k**) or Flt3L-cDC2 (**l**) of both genotypes were cultured and analyzed as described in **c** and **d**, with different amounts of SIINFEKL peptides as antigens. Data are representative of three independent experiments. Data are presented as mean ± s.d. Statistical analysis was performed using two-tailed unpaired Student's *t*-test (**b**) or two-way ANOVA with Sidak's multiple comparisons test (**c–l**). Source data are provided as a Source Data file.

processing and presentation, lysosome and phagosome were enriched or became more significantly enriched after stimulation with HKLM-OVA (Fig. 5b). Gene Set Enrichment Analysis (GSEA) further uncovered that most of the genes in the pathways of antigen processing and presentation (Fig. 5c), phagosome (Fig. 5d) and some in the lysosome pathway (Fig. 5e) were down-regulated in *Zeb1*-deficient cDC1 after stimulation. Among the down-regulated genes in the pathways of antigen processing and presentation, and phagosome, genes including *H2-T22, H2-M2, H2-Q4*, etc., that encode MHC-Ib molecules might not be the causative factors for defective cross-presentation, as they have functions other than antigen presentation to conventional CD8⁺ T cells[40] (Fig. 5f). Two notable genes, *Cybb* and *Ncf2*, which encode subunits of Nox2, in the phagosome pathway, were down-regulated in *Zeb1*-deficient cDC1 after stimulation. Strong reduction of Cybb and slight reduction of Ncf2 were further confirmed by immunoblotting (Fig. 5g), suggesting a potential requirement of Zeb1 for phagosomal ROS production. Given that Nox2 activity is absolutely required for cross-presentation of particulate antigens[23,24], the reduction of Cybb and Ncf2 might contribute to the defective cross-presentation in *Zeb1*-deficient cDC1. Although most of the genes in the lysosome pathway were down-regulated in *Zeb1*-deficient cDC1, the genes (*Ap1b1, Ap1m2,* and *Ap1s3*) encoding subunits of the clathrin adapter AP-1 that controls not only the distribution of apical proteins, but also the trafficking of endocytic vesicles[41], were up-regulated in *Zeb1*-deficient cDC1 (Fig. 5f

and Supplementary Fig. 7d). Taken together, these results suggested that Zeb1 promoted cross-presentation probably by regulating the expression of a series of genes involved in the phagosome and lysosome pathways.

In most cases, Zeb1 directly represses the transcription of its target genes by recruiting co-repressor Ctbp. In other instances, Zeb1 can also activates transcription by recruiting co-activators such as p300[25]. To determine whether the DEGs identified above were directly regulated by Zeb1, we conducted a genome-wide Zeb1 binding analysis in WT cDC1 after stimulation with HKLM-OVA using Cleavage Under Targets and Tagmentation (CUT&Tag)[42]. Our analysis identified 23,896 consensus Zeb1 binding sites in cDC1 after stimulation. Zeb1 binding in the gene promoter regions and sequences surrounding a transcription starting site accounted for 46% of the regions bound by Zeb1 in cDC1 after stimulation (Supplementary Fig. 7e, f). Intragenic binding represented 25%, whereas distal intergenic binding represented another 25% of the binding sites in cDC1 (Supplementary Fig. 7e, f). As expected, known motif discovery analysis retrieved the canonical Zeb1-binding motif as one of the most enriched motifs in the Zeb1 binding peaks, indicating the reliability of CUT&Tag as a method comparable to Chromatin Immunoprecipitation with high-throughput sequencing (ChIP-seq) (Supplementary Fig. 7g)[43,44]. Comparison of Zeb1-bound genes with Zeb1-regulated genes revealed that 1,888 genes were transcriptionally regulated by Zeb1 (Supplementary Fig. 7h). KEGG

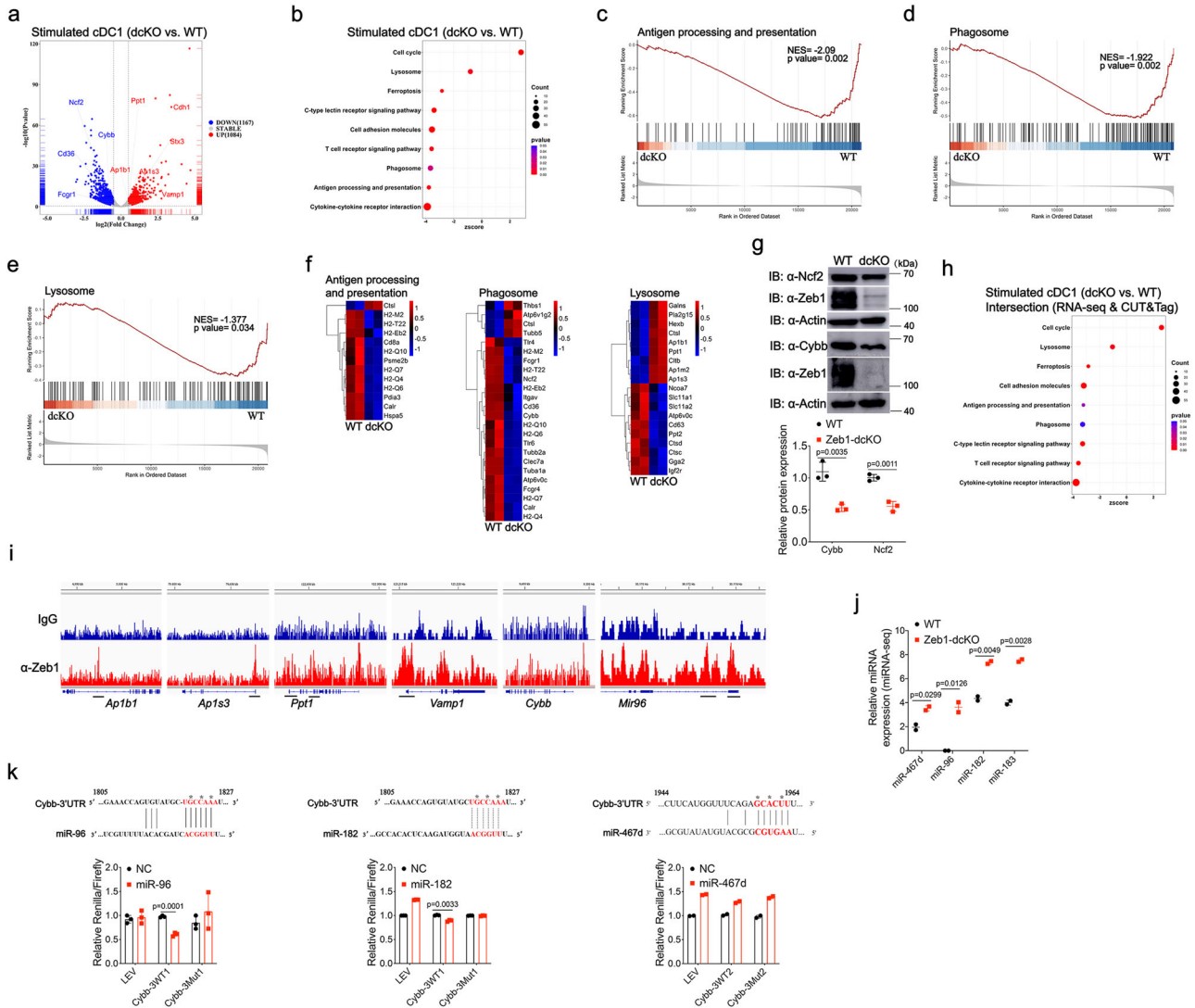

**Fig. 5 | Integrated analysis of transcriptomes and Zeb1 occupancy identifies Zeb1 target genes that support cross-presentation. a** Volcano plot illustrating DEGs in WT and Zeb1-deficient Flt3L-cDC1 challenged with HKLM-OVA for 4 h ($n = 2$). **b** Bubble plot depicting KEGG pathway analysis of DEGs in WT and Zeb1-deficient Flt3L-cDC1 after stimulation as in a ($n = 2$). GSEA profiles of KEGG pathways of antigen processing and presentation (**c**), phagosome (**d**), lysosome pathway (**e**), in WT and Zeb1-deficient Flt3L-cDC1 after stimulation as in **a**. **f** Heatmaps of differentially expressed signature genes in pathways of antigen processing and presentation, phagosome and lysosome in WT and Zeb1-deficient Flt3L-cDC1 after stimulation as in **a** ($n = 2$). **g** Immunoblot analysis of NOX2 subunits Cybb and Ncf2 in WT and Zeb1-deficient Flt3L-cDC1 after stimulation with HKLM-OVA for 4 h. Relative protein level is quantified by normalization to actin protein ($n = 3$, from 3 independent experiments). **h** Bubble plot presenting KEGG pathway analysis of DEGs in intersection of Zeb1-regulated genes obtained from above bulk RNA-seq

and Zeb1-bound genes obtained from CUT&Tag with Flt3L-cDC1 after stimulation. **i** Integrative Genomics Viewer (IGV) showing Zeb1 binding peaks (CUT&Tag) in indicated gene loci of WT Flt3L-cDC1 challenged with HKLM-OVA for 4 h. **j** miRNA-seq illustrating expression levels of the indicated miRNAs in WT and Zeb1-deficient Flt3L-cDC1 challenged with HKLM-OVA for 4 h ($n = 2$ biologically independent samples in one experiment). **k** Relative luciferase activity in HEK293T cells transfected with dual reporter vector containing WT or mutated miR-96/182- or miR-467d-binding sites in Cybb together with empty or miR-96- or miR-182- or miR-467d-expressing vector. Renilla luciferase activity is normalized to firefly luciferase activity. * marked on nucleotides represents mutated sites. Each symbol represents an individual sample, small horizontal lines indicate the mean (±s.d.). Data are representative of three independent experiments (**g**, **k**) (error bars, s.d.). Data are presented as mean ± s.d. Statistical analysis was performed using two-tailed unpaired Student's *t*-test. Source data are provided as a Source Data file.

analysis of the DEGs in the intersection revealed that Zeb1 target genes were significantly enriched in the pathways of cell adhesion molecules and lysosome (Fig. 5h). This suggested that most of DEGs in these pathways were directly bound by Zeb1 and were its direct target genes. For example, we observed strong Zeb1 binding peaks in the promoter regions of *Ap1b1* and *Ap1s3*, as well as in the promoter region and intragenic region of *Ppt1*, which suppresses cross-presentation (Fig. 5i)[45]. On the other hand, the significance of the association of Zeb1 target genes with the pathways of phagosome, antigen processing and presentation, was notably reduced compared to the association of Zeb1-regulated genes (Fig. 5h, b). This meant that significant

Zeb1 binding signal was not detected in some of DEGs in the two pathways, e.g., *Cybb* (Fig. 5i). Altogether, these results demonstrated that Zeb1 might directly regulate the transcription of most DEGs in the lysosome pathway, but indirectly regulate the transcription of some DEGs including *Cybb* in the pathways of phagosome, antigen processing and presentation.

It has been reported that Zeb1 links EMT-activation and stemness-maintenance by suppressing microRNAs (miRNAs) that inhibit stemness[46]. As *Cybb* was not directly regulated by Zeb1, we next investigated whether Zeb1 elevated the expression of Cybb by repressing a miRNA that targeted *Cybb* mRNA. To this end, we

performed microRNA deep sequencing (miRNA-seq) analysis of WT and *Zeb1*-deficient cDC1 after stimulation with HKLM-OVA. A comparison of miRNA transcriptomes of WT and *Zeb1*-deficient cDC1 led to the identification of 31 miRNA DEGs (Supplementary Fig. 7i and Supplementary Data 4), among which 19 miRNAs were direct targets of Zeb1 (Supplementary Fig. 7i), including miR-96 and miR-182 that were predicted to bind to the 3' untranslated region (UTR) of *Cybb* mRNA[47]. Two Zeb1 binding peaks were identified near the promoter of miR-183-96-182 cluster by CUT&Tag, consistent with previous reports that two Zeb1-binding motifs were found upstream of human miR-183-96-182 cluster (Fig. 5i)[46,48]. As anticipated, miR-96, miR-182, and miR-183 were all up-regulated in Zeb1-deficient cDC1 after stimulation (Fig. 5j and Supplementary Fig. 7j). Thus, our data supported the finding that Zeb1 directly represses the transcription of the miR-183-96-182 cluster[46,48]. To examine the direct regulation of *Cybb* by miR-96 or miR-182, we performed a dual-luciferase reporter assay. The results showed that miR-96 significantly reduced but miR-182 only slightly reduced protein expression under the control of *Cybb* 3'UTR, while mutation of the binding site abolished both reductions (Fig. 5k). miR-467d, on the other hand, was unable to reduce protein expression through either WT or mutated 3'UTR of *Cybb*, although miR-467d was predicted to weakly bind to 3'UTR of *Cybb* (Fig. 5k). Therefore, miR-96/182 were able to suppress the expression of *Cybb* by binding to 3'UTR of the transcript. Collectively, our data demonstrated that Zeb1 activated the expression of Cybb by repressing the transcription of *Cybb*-targeting miR-96/182 and suggested that Zeb1 might be required for phagosomal ROS production.

## Phagosomal ROS production and membrane rupture require Zeb1

DC undergo a program of maturation following recognition of microbes, which transiently enhances the capacity to phagocytose antigens and also increases the expression of costimulatory molecules and inflammatory cytokines as the second and third signal for effective T cell response[49]. First, the defective cross-presentation was not due to the maturation of *Zeb1*-deficient Flt3L-cDC1, because they expressed normal levels of MHC class I molecules and costimulatory molecules including CD80, CD86, CD40, as well as inflammatory cytokines following stimulation with HKLM-OVA or TLR ligands (Supplementary Fig. 8a and Supplementary Fig. 3). Additionally, *Zeb1*-deficient cDC1 had similar capability in uptake of soluble OVA, HKLM-OVA, and OVA-loaded $\beta 2m^{-/-}$ splenocytes (Supplementary Fig. 8b–d), relative to WT cDC1. Taken together, these data indicated that Zeb1 was dispensable for DC maturation and phagocytosis.

As we observed that Zeb1 deletion curtailed the expression of Cybb through miR-96 and miR-182 upregulation in cDC1, we then investigated whether the defective cross-presentation in *Zeb1*-deficient cDC1 was caused by miR-96/182-mediated Cybb downregulation. To address this, we transduced BM Lin⁻c-Kit^hi stem cells from Zeb1-dcKO mice with a lentiviral vector expressing *Cybb* during Flt3L-induced in vitro DC development. The infected Flt3L-cDC1 were sorted and assayed for cross-presentation. Strikingly, lentiviral expression of *Cybb* in *Zeb1*-deficient cDC1 completely restored the capability of cross-presentation of bacteria-associated antigens (Fig. 6a). Then we infected BM Lin⁻c-Kit^hi stem cells from B6 mice with control retrovirus or retrovirus encoding miR-96 or miR-182 during Flt3L-induced in vitro DC development and tested the infected Flt3L-cDC1. Consistent with the observation from dual luciferase reporter assay (Fig. 5k), miR-96 overexpression significantly but miR-182 overexpression only slightly suppressed protein expression of Cybb (Supplementary Fig. 8e). The cross-presentation of bacteria-associated antigens by infected Flt3L-cDC1 was partially and significantly inhibited by miR-96 overexpression and mildly down-regulated by miR-182 overexpression (Fig. 6b), which to some extent mimicked the defective cross-presentation in *Zeb1*-deficient cDC1. Collectively, these results

demonstrated that the Zeb1-miR-96/182-Cybb pathway plays a crucial role in controlling cross-presentation in cDC1.

We next examined whether phagosomal ROS was affected by *Zeb1*-deficient Flt3L- cDC1. First, we stimulated Flt3L-cDC1 with HKLM-OVA and measured ROS production in the cytoplasm and mitochondria after phagocytosis, using CellROX and MitoSOX, respectively. We observed that *Zeb1*-deficient Flt3L-cDC1 produced similar levels of mitochondrial ROS and mildly lower levels of cytosolic ROS compared to WT cDC1 (Fig. 6c). Then we fed Flt3L-cDC1 HKLM-OVA labeled with OxyBURST and Alexa Fluor 647 and measured the oxidative burst as indicator of phagosomal ROS production. We found that the ratio of OxyBURST⁺ to Alexa Fluor 647⁺ cDC1 population increased progressively in WT cDC1 following HKLM-OVA ingestion but this ratio remained at a very low level in *Zeb1*-deficient cDC1 (Fig. 6d, e). Additionally, *Zeb1*-deficient cDC1 generated much lower intensity of Oxidative Burst than WT cDC1 upon phagocytosis of HKLM-OVA (Fig. 6e). Phagosomal ROS is known to play a crucial role in cross-presentation[23,24], and consistent with this, inhibition of the NADPH oxidase by diphenyleneiodonium chloride (DPI) completely blocked cross-presentation of bacteria-associated antigens in both WT and *Zeb1*-deficient cDC1 (Fig. 6f). Taken together, these results demonstrated that Zeb1 was required for production of phagosomal ROS but not of mitochondrial ROS in cDC1 upon phagocytosis.

Phagosomal ROS has been shown to play a critical role in controlling phagosomal acidification, proteolysis, and membrane rupture[23,24,50]. To investigate whether the defective phagosomal ROS production affect phagosomal acidification, we monitored phagosomal pH by exposing Flt3L-cDC1 to FITC-coupled (pH-sensitive), Alexa Fluor 647-coupled (pH-insensitive) or pHrodo Red-coupled (pH indicator) HKLM-OVA. Surprisingly, we did not observe any difference in all the three kinds of fluorescence in cDC1 of both genotypes at all tested time points after phagocytosis (Supplementary Fig. 8f), suggesting that phagosomal acidification was not affected by Zeb1 deletion in cDC1, despite the defective phagosomal ROS production.

Once antigens are internalized, they are processed and loaded onto MHC-I molecules through the vacuolar or the P2C pathway[18,19]. The P2C pathway is believed to be most dominant mechanism in cross-presentation and immune surveillance. The antigen export into the cytosol is mediated by ERAD machinery or phagosomal rupture[18,19,51]. A recent study has reported that phagosomal membrane damage induced by phagosomal ROS is necessary for P2C transfer of dead cell-associated antigens and, thereby, for cross-presentation[23]. We investigated whether the defective phagosomal ROS production in Zeb1-deficient cDC1 reduced phagosomal membrane damage. To address this, we measured recruitment of cytosolic galectin-3, which binds to sugar moieties attached to membrane proteins on the luminal side of phagosomes, to the HKLM-OVA-containing phagosomes, as a marker for phagosomal membrane damage[23,52]. Confocal microscopy revealed that following uptake of Alexa Fluor 647-labeled HKLM-OVA, WT cDC1 had more galectin-3 puncta than *Zeb1*-deficient cDC1. Moreover, the galectin-3 puncta highly colocalized with HKLM-OVA⁺ phagosomes in WT cDC1 but not in *Zeb1*-deficient cDC1, indicating diminished phagosomal rupture in the latter (Fig. 6g, h). Collectively, these results suggested that impaired phagosomal ROS production caused by Zeb1 ablation in DC resulted in reduced phagosomal rupture, which likely hindered antigen export to the cytosol and impaired cross-presentation.

## Deletion of Zeb1 prevents antigen export to the cytosol in cDC1

The P2C pathway crucially depends on the export of internalized antigens from phagosome to the cytosol[51]. Given the observed reduction in phagosomal ROS production and phagosomal rupture in *Zeb1*-deficient cDC1, we sought to investigate whether *Zeb1* deficiency affected antigen export to the cytosol. To accomplish this, we performed a cytofluorimetry assay based on the enzymatic activity of β-

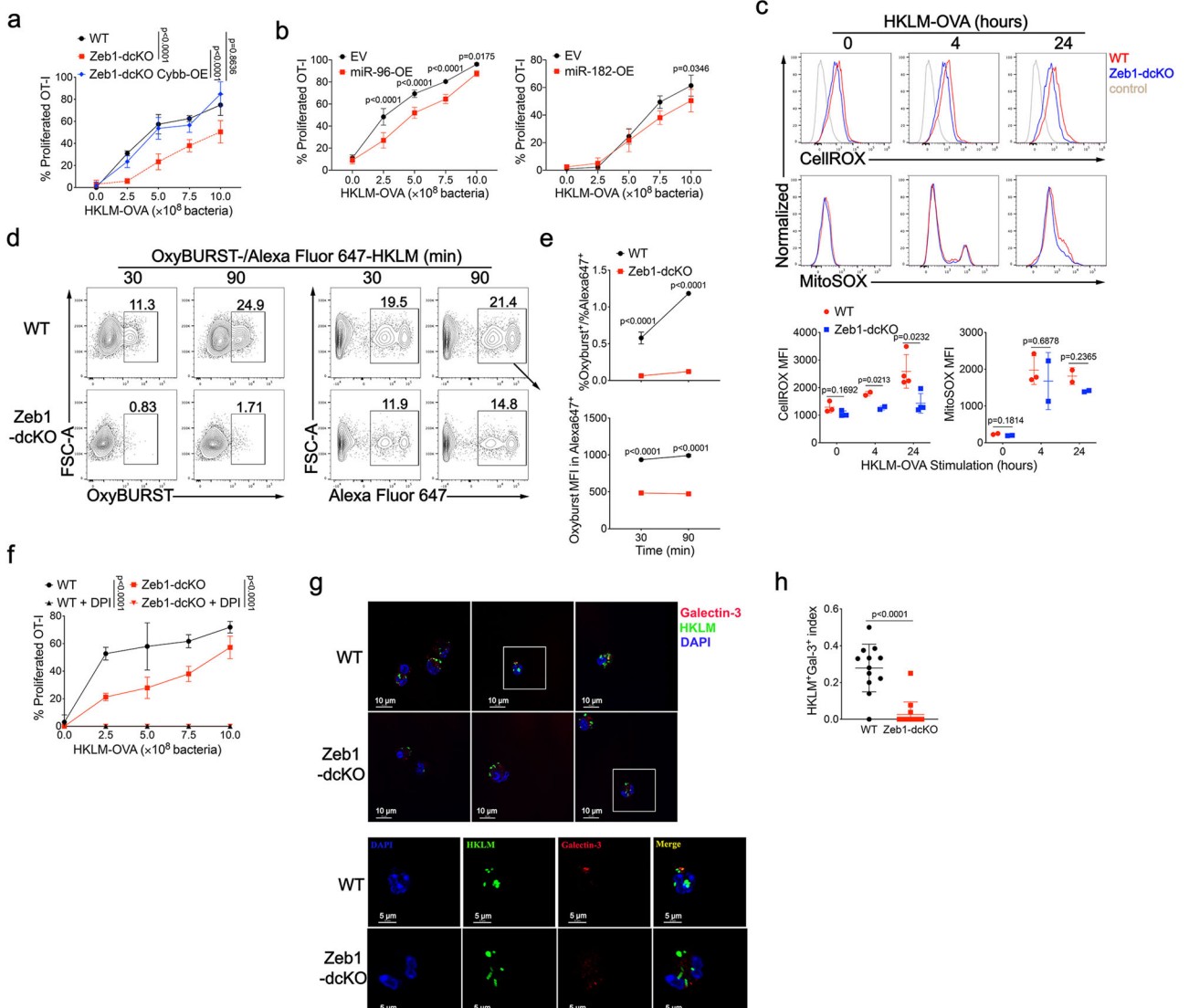

**Fig. 6 | Zeb1 controls phagosomal ROS-dependent rupture of phagosomal membrane in cDC1. a** WT or Zeb1-deficient Flt3L-cDC1, or Zeb1-deficient Flt3L-cDC1 lentivirally transduced with Cybb containing synonymously mutated miR-96 binding sites were cultured for 3 days with CFSE-labeled OT-I T cells and different dose of HKLM-OVA and assayed for OT-I proliferation and activation (CFSE⁻CD44⁺). **b** WT Flt3L-cDC1 retrovirally transduced with empty vector (EV) or miR-96 or miR-182 were cultured and analyzed as described in (**a**), with various dose of HKLM-OVA as antigens. **c** Intracellular and mitochondrial ROS production in WT and Zeb1-deficient Flt3L-cDC1 challenged with HKLM-OVA for indicated times, were measured by flow cytometric analysis of CellROX and MitoSOX fluorescence. The mean fluorescent intensity (MFI) of CellROX and MitoSOX fluorescence were quantified at the bottom. **d, e** Phagosomal ROS production in WT and Zeb1-deficient Flt3L-cDC1 exposed with OxyBURST/Alexa Fluor 647-conjugated HKLM-OVA for indicated times, were measured by flow cytometric analysis (**d**) of OxyBURST and Alexa

Fluor 647 fluorescence. The phagosomal ROS production was quantified as the ratio of OxyBURST⁺ cells to Alexa Fluor 647⁺ cells (**e**, top row) or MFI of OxyBURST in Alexa Fluor 647⁺ cells (**e**, bottom row). **f** WT and Zeb1-deficient Flt3L-cDC1 were cultured with HKLM-OVA for 30 min, and treated with or without DPI (10 μM) for 4 h before addition of CFSE-labeled OT-I T cells, and assayed for OT-I proliferation and activation (CFSE⁻CD44⁺). **g, h** Confocal microscope images of mCherry::galectin-3-expressed WT and Zeb1-deficient Flt3L-cDC1 challenged with Alexa Fluor 647-labeled HKLM-OVA. Colocalizing signal of Galectin-3⁺HKLM⁺ cells were counted and plotted as a ratio of total HKLM⁺ cells. Each symbol represents an individual sample, small horizontal lines indicate the mean (±s.d.). Data are representative of three independent experiments. Data are presented as mean ± s.d. Statistical analysis was performed using Two-tailed unpaired Student's test (**h**) or two-way ANOVA with Tukey's (**a**, **f**) or Sidak's (**b**, **e**) multiple comparisons test. Source data are provided as a Source Data file.

lactamase[53]. Purified Flt3L-cDC1 were loaded with CCF4, a cytosolic fluorescence resonance energy transfer (FRET) substrate of β-lactamase. Upon uptake of soluble β-lactamase or β-lactamase-loaded $β2m^{-/-}$ splenocytes, the enzyme is transported to the cytosol and then cleaves the CCF4 dye, resulting in a switch from 525 nm emission fluorescence to 450 nm fluorescence. Cleavage of CCF4 increased over time in WT cDC1 under both experimental settings, by contrast, it was significantly reduced in *Zeb1*-deficient cDC1, indicating a reduction in translocation of both types of β-lactamase (Fig. 7a–d). These results demonstrated that Zeb1 was required for efficient export

of both soluble and cell-associated antigens into the cytosol. Once in the cytosol, antigens are degraded into oligopeptides by proteasome. To test whether *Zeb1* deficiency impacted this process, we first incubated Flt3L-cDC1 with $β2m^{-/-}$ splenocytes loaded with DQ-OVA, a protease probe consisting of OVA proteins heavily labeled with self-quenched BODIPY dye that becomes brightly fluorescent upon OVA hydrolysis. Zeb1 ablation significantly decreased DQ-OVA fluorescence, which was blocked by proteasome inhibitor MG132, suggesting that less antigen escaped from phagosomes into the cytosol for proteasome-mediated antigen processing in *Zeb1*-deficient cDC1

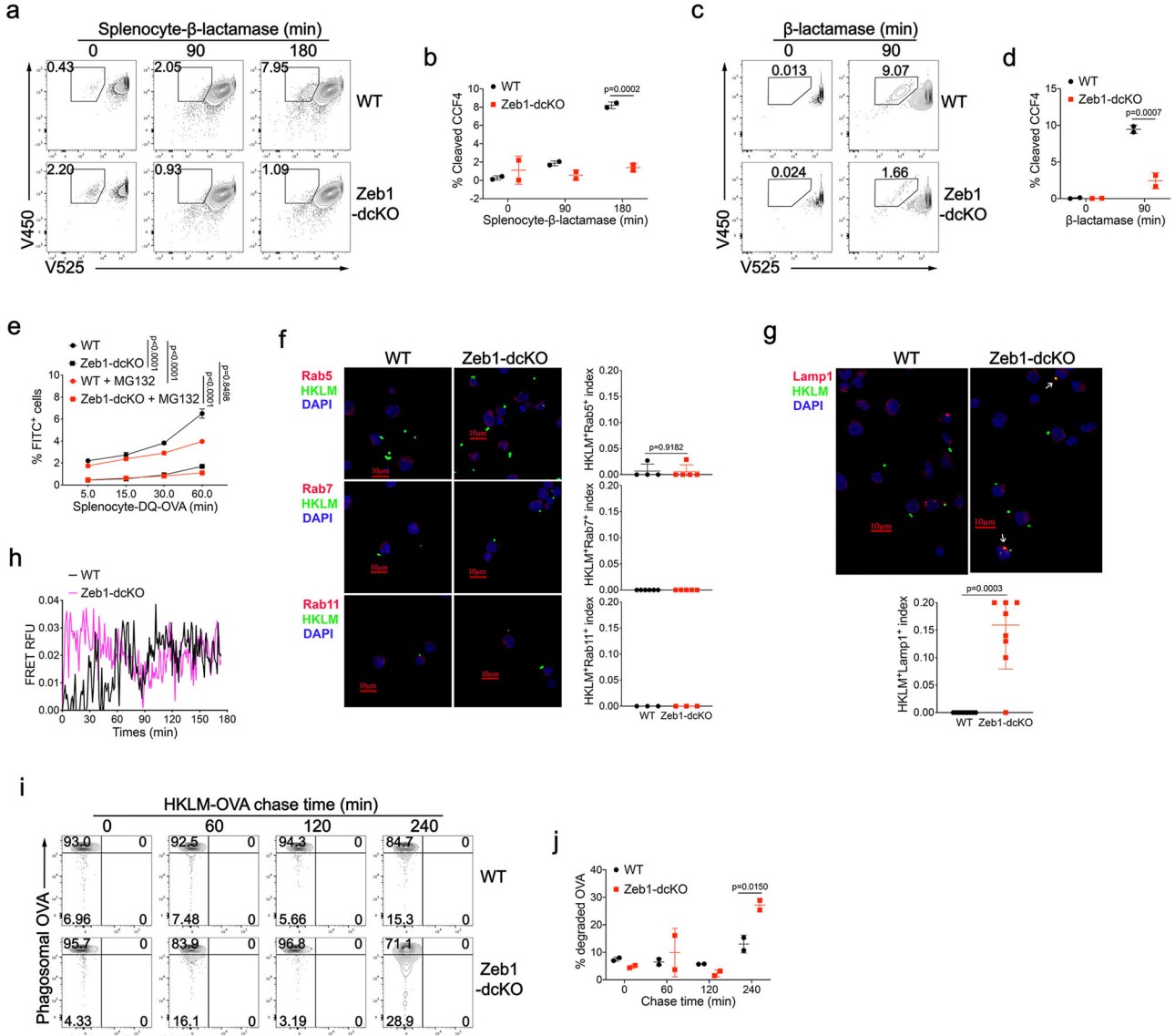

**Fig. 7 | Zeb1 deficiency prevents antigen export to the cytosol and enhances phago-lysosome fusion in cDC1.** CCF4-loaded WT and Zeb1-deficient Flt3L-cDC1 were incubated with irradiated β-lactamase-loaded β2m$^{-/-}$ splenocytes (**a, b**) or soluble β-lactamase (**c, d**) for indicated times. The cleavage of CCF4 was monitored as change in CCF4 fluorescence by flow cytometry (**a, c**). The efficiency of antigen export to the cytosol was quantified as frequency of cleaved-CCF4$^+$ cells (**b, d**) (n = 2). **e** WT and Zeb1-deficient Flt3L-cDC1 were incubated with irradiated DQ-OVA-loaded β2m$^{-/-}$ splenocytes in the presence or absence of MG132 for indicated times, and percentages of FITC$^+$ cells were monitored by flow cytometry (n = 3). Confocal microscope images of WT and Zeb1-deficient Flt3L-cDC1 stained with anti-Rab5, anti-Rab7, anti-Rab11 (**f**) or anti-Lamp1 (**g**) antibodies 4 h after phagocytosis of Alexa Fluor 647-labeled HKLM-OVA. Colocalizing signal of Rab5$^+$HKLM$^+$, Rab7$^+$HKLM$^+$, Rab11$^+$HKLM$^+$ or Lamp1$^+$HKLM$^+$ were counted and plotted as a ratio of total HKLM$^+$ cells. **h** Phago-lysosome fusion was measured by exposing WT and Zeb1-deficient Flt3L-cDC1 loaded with lysosomal FRET acceptor Alexa Fluor 594-Hydrazide to donor Alexa Fluor 488-labeled HKLM-OVA. Fluorescent measurement was taken every 1 min for 3 h. **i** Flt3L-cDC1 were allowed to internalize HKLM-OVA at 16 °C and incubated for indicated chase periods at 37 °C to allow phagosome maturation. Organelles from lysed cells were analyzed for phagosomal OVA and non-internalized cell surface OVA by flow cytometry. **j** The degradation of phagosomal OVA was quantified as frequency of phagosomes containing decreased abundance of OVA (n = 2). Each symbol represents an individual sample or section, small horizontal lines indicate the mean (±s.d.). Data are representative of three independent experiments. Data are presented as mean ± s.d. Statistical analysis was performed using Two-tailed unpaired Student's test (**g**) or two-way ANOVA with Tukey's (**e**) or Sidak's (**b, d, j**) multiple comparisons test. Source data are provided as a Source Data file.

(Fig. 7e). To assess cytosolic antigen degradation by proteasome, we introduced DQ-OVA plus Alexa Fluor 647-OVA directly into the cytosol of cDC1 by electroporation. The ratios of percentage and intensity of DQ-OVA fluorescence to that of Alexa Fluor 647 fluorescence were slightly higher after chasing in *Zeb1*-deficient cDC1 than in WT counterparts (Supplementary Fig. 8g, h). These results suggested that *Zeb1*-deficient cDC1 had a little stronger capacity to degrade endogenous antigens via proteasome than WT counterparts, which could not account for the defective cross-presentation.

It has been reported that delay of phago-lysosome fusion prevents excessive degradation of internalized antigens and enhances cross-presentation[54]. The expression of AP-1 adapter subunit encoding genes, including *Ap1b1, Ap1m2,* and *Ap1s3*, was found to be increased in *Zeb1*-deficient cDC1 after stimulation (Fig. 5f and Supplementary Fig. 7d), which suggests an increase in trafficking to the lysosome. Given the impaired export of internalized antigens into the cytosol in *Zeb1*-deficient cDC1, we next determine whether these antigens gained more opportunity to be transported from

endosome/phagosome to lysosome. We analyzed various cellular compartments of WT and *Zeb1*-deficient Flt3L-cDC1 using confocal microscopy at steady state and after phagocytosis of HKLM-OVA. We found no significant difference in the distribution of early endosomes (Rab5), late endosomes (Rab7), recycling endosomes (Rab11) and lysosomes (Lamp1) between the two groups at steady state (Supplementary Fig. 8i). 4 h after phagocytosis, we did not observe significant colocalization of HKLM-OVA and early endosomes/phagosomes, late endosomes/phagosomes, or recycling endosomes in both WT and *Zeb1*-deficient cDC1 (Fig. 7f), but did observe a higher colocalization of HKLM-OVA and Lamp1⁺ lysosome in *Zeb1*-deficient cDC1 than in WT cDC1 (Fig. 7g). These data suggested that the fusion of phagosomes containing HKLM-OVA with lysosomes was dramatically increased in *Zeb1*-deficient cDC1 at 4 h after phagocytosis. We also employed a FRET-based assay to confirm that phago-lysosome fusion was enhanced in *Zeb1*-deficient cDC1 (Fig. 7h). We next sought to evaluate the functional consequences of such enhanced phago-lysosome fusion in terms of antigen degradation in individual phagosomes. Using flow organellocytometry, we found that the kinetics and efficiency of phagosomal OVA degradation were much faster in *Zeb1*-deficient cDC1 than in WT cDC1 (Fig. 7I, j). Collectively, these results suggested that internalized antigens in *Zeb1*-deficient cDC1 lost access to the cytosol and instead gained more access to lysosomes, making them more susceptible to degradation by lysosomal proteases (Fig. 8).

## Discussion

Although transcriptional regulation of cDC1 development has been intensively investigated, the mechanisms controlling the homeostasis and function of cDC1 remain elusive. In this study, we provide genetic evidence of a critical role for Zeb1 in the homeostasis and function of cDC1. We have shown that DC-specific *Zeb1*-deficient mice had a selective reduction in splenic cDC1 accompanied by excessive cell death, which rendered them more resistant to Listeria infection. We have also shown that Zeb1 was required for cross-presentation of exogenous antigens by cDC1, and the defective cross-presentation in *Zeb1*-deficient cDC1 was due to diminished phagosomal ROS-induced phagosomal rupture and subsequent impaired antigen export to the cytosol. Using multiple omics approaches, we have identified a Zeb1-miR-96/182-Cybb regulatory axis that controls cross-presentation in cDC1 (Fig. 8).

The Immunological Genome Consortium reported that both Zeb1 and Zeb2 are widely expressed in murine myeloid immune cells[26,27], with Zeb2 being expressed in pDC and cDC2. However, its expression is down-regulated in the transition from CDP to pre-cDC1, and become nearly absent in mature cDC1, which is consistent with its role in preventing the switch from +41 kb *Irf8* enhancer to +32 kb *Irf8* enhancer during cDC1 development[8]. On the other hand, although Zeb1 expression is upregulated during DC development, our data demonstrated that Zeb1 was dispensable for cDC1 development but crucial for cDC1 homeostasis. While cDC2 homeostasis has been well studied,

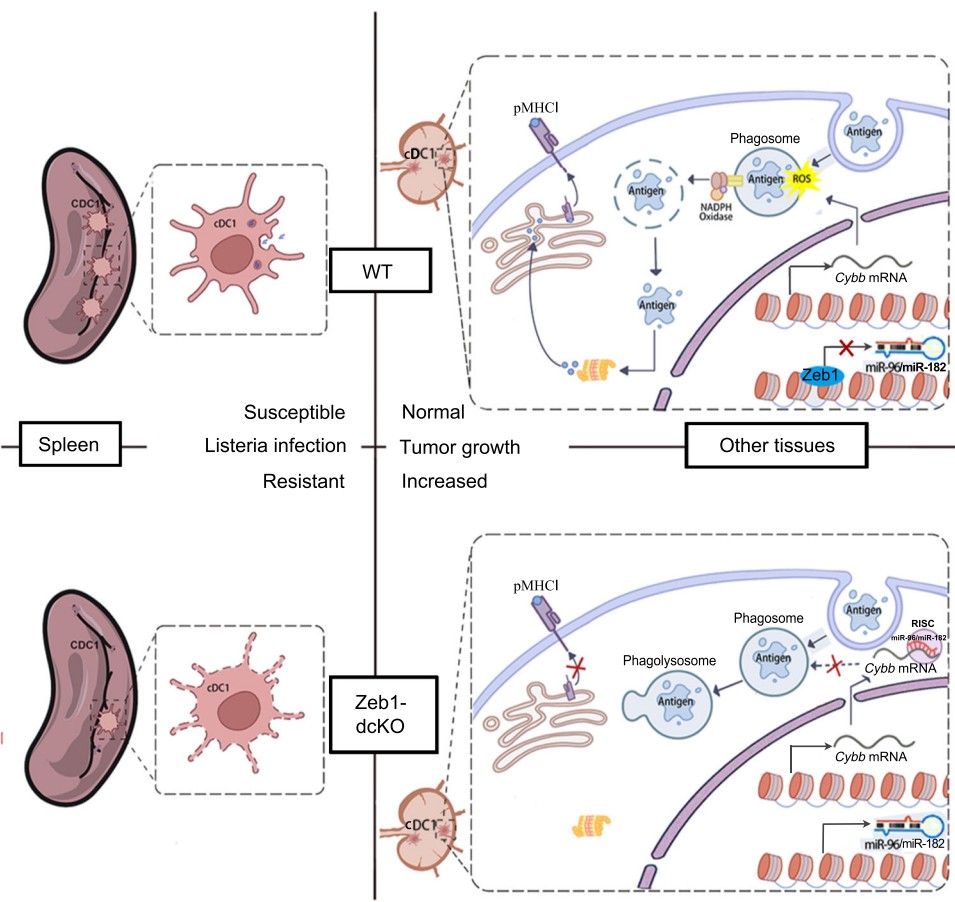

**Fig. 8 | Proposed model for the role of Zeb1 in homeostasis and function of cDC1.** *Zeb1* deficiency in dendritic cells leads to a selective reduction of splenic cDC1, associated with excessive cell death, rendering mice more resistant to Listeria infection. Additionally, cDC1 from other sources of *Zeb1*-deficient mice display impaired cross-presentation of exogenous antigens, compromising antitumor CD8⁺ T cell responses. Mechanistically, Zeb1 facilitates the production of phagosomal reactive oxygen species (ROS) by repressing the expression of microRNA-96/182 that targeted *Cybb* mRNA of NADPH oxidase Nox2. Consequently, loss of Zeb1 in cDC1 diminishes phagosomal ROS and subsequent phagosomal membrane rupture that allows antigen export to the cytosol.

our knowledge on cDC1 homeostasis is still limited. The spleen, the largest second lymphoid organ, specializes in filtering blood and trapping blood-borne pathogens or antigens. Impaired position, retention of cDC2 or the mechano-sensing of blood cells in marginal zone bridging channel of the spleen cause selective loss of splenic cDC2[11,12]. *Zeb1* deficiency in DC resulted in reduction of cDC1 only in the spleen but not in other lymphoid and non-lymphoid tissues. Moreover, the capacity of BM cells from Zeb1-dcKO mice to generate all DC populations was comparable to that of BM cells from WT mice when cultured with Flt3L in vitro. These results supported our conclusion that Zeb1 was required for the homeostasis rather than the development of cDC1. Our scRNA-seq analysis suggested that several cell death pathways were enriched in *Zeb1*-deficient splenic cDC1. We did observe a higher frequency of dying splenic cDC1 in Zeb1-dcKO mice than in WT littermates. However, the mechanism by which Zeb1 deficiency led to more cell death in splenic cDC1 is still unclear. Most splenic cDC1 localize in the white pulp or marginal zone and red pulp of the spleen, where they are readily exposed to blood-borne pathogens and dead cells and require appropriate mechano-sensing or chemoattraction[55]. It has been reported that E-cadherin, the most important target of Zeb1, augment death receptor clustering and assembly of death-inducing signaling complex (DISC), thus enhance sensitivity to death receptor-mediated apoptosis[56]. Whether excessive death of *Zeb1*-deficient splenic cDC1 is induced by these blood-borne stimuli, or aberrant mechano-sensing, chemoattraction or cell adhesion, needs to be further investigated. Interestingly, in mixed BM chimera, deletion of Zeb1 during DC development not only reduced regeneration of splenic cDC1 from Zeb1-dcKO BM cells themselves, but also impeded repopulation of splenic cDC1 from B6 BM cells in the same recipient. It has been reported that cDC1 can be selectively depleted by apoptosis triggered by translocation of internalized cytochrome c (cyt c) into the cytoplasm[57], which has been developed into a method to measure antigen export to the cytosol[53]. In fact, we found that cDC1 also underwent more cell death when co-cultured with apoptotic or necroptotic splenocytes, whose cyt c could be transported into the cytosol of cDC1 after phagocytosis by cDC1, than with live splenocytes. Hence, we speculated that cDC1 from B6 BM cells might undergo apoptosis following uptake of dying or dead cDC1 from *Zeb1*-deficient BM cells in the spleen of mixed BM chimera, resulting in similar loss of both *Zeb1*-sufficient and -deficient splenic cDC1.

Cross-presentation of particulate antigens, including bead-coupled, cell-associated, and bacterial-associated antigens, relies on the P2C pathway rather than the vacuolar pathway[18,19]. This is a highly intricate process that involves antigen uptake, export to the cytosol, and loading on MHC-I molecules. The critical step in this pathway is antigen export to the cytosol, which can be facilitated by ERAD machinery or phagosomal disruption[18,19]. The latter mechanism is particularly useful for non-protein particles like dextran or latex beads, and for enzymes to be transported into the cytosol in an enzymatically active state. However, the mechanisms that cause phagosomal membrane rupture remain unclear until recently. Two studies have shown that phagosomal ROS produced by Nox2 induced phagosomal membrane rupture probably through lipid peroxidation, leading to antigen release to the cytosol and cross-presentation[22,23]. Loss of Cybb or Vav, required for phagosomal ROS production, resulted in defective cross-presentation of particulate antigens, but not soluble antigens[23,24,58]. Conversely, a recent report demonstrated that endosomal membrane repair by endosomal sorting complex required for transport (ESCRT)-III restrained antigen export to the cytosol and inhibited cross-presentation in cDC[59]. Our findings indicated that *Zeb1*-deficient cDC1 exhibited a dramatic defect in cross-presentation of cell-associated and bacterial-associated antigens, and a mild reduction in cross-presentation of soluble antigens. We also discovered that loss of Zeb1 in cDC1 impaired phagosomal ROS production, diminished phagosomal membrane rupture, and correspondingly prevented antigen export to the cytosol. All these results suggested that the defective cross-presentation in *Zeb1*-deficient cDC1 was likely and mainly due to the diminished phagosomal ROS production. This conclusion was supported by the results that enforced expression of Cybb completely rescued the defective cross-presentation in *Zeb1*-deficient cDC1.

Nox2 is a large protein complex comprised of two membrane subunits gp91$^{phox}$ (encoded by *Cybb*) and gp22$^{phox}$, as well as three cytosolic components p40$^{phox}$, p47$^{phox}$ and p67$^{phox}$ (encode by *Ncf2*)[60]. The assembly of Nox2 on phagosomal membrane selectively in cDC1 requires the small GTPase Rac2[61], and the recruitment of Nox2 to phagosome involves another GTPase Rab27a and SNARE protein Vamp8[62,63]. Ablation of any of these proteins, Rac2, Rab27a or Vamp8 attenuated cross-presentation in DC[61-63]. However, transcriptional regulation of Nox2-encoding genes in cDC1 remains to be extensively investigated. Our data demonstrated that Zeb1 promoted expression of Cybb indirectly by repressing the expression of *Cybb*-targeting miR-96/182. Although the genetic evidence for the role of miR-96/182 in cross-presentation is still lacking, overexpression of miR-96/182 in *Zeb1*-sufficient cDC1 partly and significantly inhibited cross-presentation, which to some extent mimics the defective cross-presentation in *Zeb1*-deficient cDC1. miR-96/182 may not be the only target of Zeb1 that regulates cross-presentation in cDC1. For instance, unleash of Ppt1 expression might also contribute to the defective cross-presentation in *Zeb1*-deficient cDC1.

Excessive lysosomal (or phagosomal) proteolysis is detrimental to antigen presentation in the context of MHC-II[64,65]. In addition to inducing phagosomal membrane rupture, phagosomal ROS also plays a crucial role in inhibiting phagosomal proteolysis by reversibly oxidizing cysteine cathepsins. Attenuation of lysosomal (or phagosomal) proteolysis preserves antigen epitopes and enhances both MHC-I and MHC-II presentation[50,65,66]. Furthermore, a recent study has demonstrated that phagosomal ROS is essential for presentation of cell-associated antigens to CD4$^+$ T cells through MHC-II in cDC[67]. In *Zeb1*-deficient cDC1, where phagosomal ROS was reduced, phagosomal antigen degradation was accelerated compared to WT cDC1. Thus, the accelerated degradation of phagosomal antigen may also contribute to the impaired presentation of exogenous antigens to CD4$^+$ T cells via MHC-II in *Zeb1*-deficient cDC1. Although *Zeb1*-deficient cDC2 exhibited intact capability to present exogenous antigens to CD4$^+$ T cells, they failed to fully compensate for cDC1 during CD4$^+$ T cell priming in tumor-bearing Zeb1-dcKO mice, resulting in mild reduction of effector CD4$^+$ T cells, probably because cDC2 did not overwhelm cDC1 at tumor sites as they did in the spleen.

In summary, our findings have established a function for Zeb1 as a critical transcriptional regulator in homeostasis and function of cDC1. We have demonstrated that Zeb1 maintains splenic cDC1 survival by preventing excessive cell death. Additionally, our research has highlighted Zeb1's role in promoting cross-presentation partly by suppressing the expression of miR-96/182, which targets *Cybb* mRNA, and has emphasized the important role of phagosomal ROS-dependent phagosomal rupture in cross-presentation. Therefore, further investigations into the mechanism of Zeb1 could offer novel therapeutic targets to enhance the potency of cDC1 in antitumor immunity.

## Methods

### Ethical approval

The research in this study complies with the animal welfare guidelines and all animal protocols were approved by members of the Institutional Animal Care and Use Committee of Xiamen University.

### Mice

C57BL/6 J (B6) (#000664), B6.SJL (#002014), OT-I (#003831), OT-II (#004194), *β2m*$^{-/-}$ (#002087), CD11c-Cre (#008068) mice were originally from the Jackson Laboratory. The strain of C57BL/6J-Zeb1$^{tm1(flox-Neo)Smoc}$, in which exon 4 was flanked by *LoxP* and *FRT* sites

on a C57BL/6 N background, was generated using a standard gene-targeting strategy by Shanghai Model Organisms Center, Inc, and then were crossed with FLP deleter strain (#009086, the Jackson Laboratory) to remove *Neo* cassette to generate the floxed *Zeb1* mice (*Zeb1*<sup>fl/fl</sup>). All mice (both genders) were littermates and were 8–12 weeks old, unless otherwise indicated in the text. The mice were housed in specific pathogen-free facilities of Xiamen University Laboratory Animal Center with a 12 h day/night cycle, at a temperature about 22 °C, a relative humidity about 50%, and were fed with a standard mouse chow diet. All animal protocols were approved by members of the Institutional Animal Care and Use Committee of Xiamen University.

### Antibodies
The information of all antibodies, for cell purification, for flow cytometry, for immunofluorescence, for western blot and for ELISA, are listed in Supplementary Data 5. Annexin V Apoptosis detection Kit (Cat#559763) and Active Caspase-3 Apoptosis Kit (Cat#550914) were from BD Biosciences. Fixable Viability Dye eFluorTM 780 (1:400 dilution; Cat#65-0865-18) was purchased from eBioscience.

### Reagents and plasmids
Collagenase D (Cat#1108888200) and DNase I (Cat#1010415900) were purchased from Roche. Collagenase IV (Cat#C4-BIOC), Ovalbumin (Cat#S7951) and anti-chicken egg albumin (Ovalbumin) (Cat#C6534) were from Sigma. Ovalbumin-Alexa Fluor™ 488 (Cat#O34781), Ovalbumin-Alexa Fluor 647™ (Cat#O34784), Alexa Flour™ 647 NHS Ester (Cat#A37573), CFSE (Cat#C34554), and SIINFEKL were from Invitrogen. Diphenyleneiodonium chloride (HY-100965) was from MEC. The expression plasmid for mCherry:Galectin-3 (Cat#85662) was purchased from Addgene. PE Annexin V Apoptosis Detection Kit I was purchased from BD bioscience (Cat#559763). Mouse IL-10 ELISA Kit was from ABclonal (Cat#RK00016). TNF-α was purchased from eBioscience. Z-VAD-FMK was from Calbiochem and Smac mimetic was from APExBIO (Cat#SM164).

### Isolation of tissue leukocytes for flow cytometry
For the isolation of leukocytes from non-lymphoid tissues (tumor, liver, and lung), tissues were dissected from mice, and were minced with dissection scissors, then were digested in RPMI 1640 containing 0.1% Collagenase IV and 20 ng/ml DNase I for 1 h at 37°C with shaking. After digestion, the tissues were then ground and filtered. The mononuclear immune cells were isolated by centrifugation on a 40 to 72% Percoll gradient. For the isolation of leukocytes from lymphoid tissues (spleen, thymus and LNs), tissues were dissected from mice, and were minced with dissection scissors, then were digested in RPMI 1640 containing 1 mg/ml Collagenase D, 20 ng/ml DNase I for 30 min at 37 °C with shaking. After digestion, the tissues were then mashed and filtered. Red blood cells were removed by ACK (Ammonium-Chloride-Potassium) lysis buffer. Single cell suspensions were then washed, filtered and then collected by centrifugation.

### Cell enrichment and purification
The enrichment of myeloid cells, the purification of OT-I or OT-II T cells were performed by negative selection using Beaverbeads™ Streptavidin (Cat#22307; Beaver). Briefly, single cell suspensions were incubated with different cocktail of biotin-conjugated antibodies (BioLegend) at 4 °C for 10 min, followed by incubation with Beaverbeads™ Streptavidin for 5 to 10 min. Then cells were separated by placing the labeled cells in tube on DynaMag™−2 Magnet (Cat# 12321D; Invitrogen) for 5 min.

### Flow cytometry and cell sorting
For blocking of Fc receptors, single cell suspensions were incubated with purified anti-CD16/32 for 15 min on ice prior to immunostaining.

Surfaces of cells in suspensions were stained with fluorochrome-conjugated antibodies in flow cytometry buffer (0.5% BSA and 0.05% NaN₃ in PBS) for 30 min at 4 °C. To monitor apoptosis, cells were fixed and permeabilized with Cytofix/Cytoperm buffer after cell surface staining. Cells were then stained with anti-active Caspase-3 antibody (Cat#550914, BD Biosciences) in Perm/Wash buffer (1×). Intracellular cytokines were stained after stimulation of cells for 4 h with HKLM-OVA or phorbol 12-Myristate 13-Acetate (PMA, 50 ng/ml; Sigma-Aldrich) plus ionomycin (1 µg/ml; Sigma-Aldrich) in the presence of GolgiStop (Cat#554724, BD Biosciences). Cells were incubated with antibodies against cell surface markers, and then fixed and permeabilized with Cytofix/Cytoperm buffer (Cat#51-2090KZ, BD Biosciences). Cells were then stained with antibodies against indicated cytokines. All flow cytometry data were acquired on BD LSRFortessa and were analyzed using FlowJo software (BD Biosciences). The gating strategies for all flow cytometric analysis are presented in Supplementary Fig. 9.

For cell sorting, cell suspensions were stained with fluorochrome-conjugated antibodies in sorting buffer (1 mM EDTA, 25 mM HEPES, 1% FBS in PBS) and then sorted with BD Aria sorter. Splenic or LN DC were sorted as L/D⁻Lin⁻MHC II ⁺CD11c⁺CD317⁻Ly6C⁻ (cDC), L/D⁻Lin⁻MHC II ⁺CD11c⁺CD317⁻Ly6C⁻XCR1⁺SIRP-α⁻(cDC1). Flt3L-cDC1 were sorted as L/D⁻B220⁻MHC II ⁺CD11c⁺CD24⁺SIRP-α⁻ and Flt3L-cDC2 were sorted as L/D⁻B220⁻MHC II ⁺CD11c⁺CD24⁻SIRP-α⁺.

### Bone Marrow culture with Flt3L
Bone Marrow (BM) cells were harvested from sex- and age-matched WT and Zeb1-dcKO littermates at 8 - 12 weeks of age, and red blood cells were removed using ACK lysis buffer. Then cells were cultured at 5 × 10⁶ cell/ml in RPMI 1640 media containing 10% FBS, 50 ng/ml Flt3L, 0.1 mM NEAA, 1 mM sodium pyruvate, 2 mM GlutaMAX, 100 U/mL penicillin, 100 µg/mL streptomycin and 50 µM β-mercaptoethanol for 9 to 11 days without disturbance. Floating cells and loosely adherent cells were harvested by vigorous pipetting, and then were analyzed or sorted by flow cytometry.

Plv-GFP-mCherry-Galectin3 or Plv-EGFP-Cybb lentiviral vectors were transfected into 293 T cells and supernatants containing virus were collected 2 days and 3 days later. Sorted Lin⁻c-Kit^hi cells from the BM of WT or Zeb1-dcKO mice were infected twice with viral super-natants with 2 µg/ml polybrene at 2500 rpm at 32 °C for 2 h. Cells then were grown in Flt3L culture for 9 to 11 days before sorting for mCherry⁺CD24⁺CD11c⁺ cDC1 or GFP⁺CD24⁺CD11c⁺ cDC1, respectively. Sorted mCherry⁺CD24⁺CD11c⁺ cDC1 were stimulated with HKLM-OVA and were used in immunostaining for confocal microscopy. And sorted GFP⁺CD24⁺CD11c⁺ cDC1 were used in cross-presentation assays.

LMP-EGFP-miR-96 or LMP-EGFP-miR-182 retroviral vectors were transfected into plat-E cells and supernatant containing virus were collected 2 days later. Sorted Lin⁻c-Kit^hi cells from the BM of WT mice were infected with viral supernatants with 10 µg/ml polybrene at 2500 rpm at 32 °C for 2 h. Cells then were grown in Flt3L culture for 9 to 11 days before sorting for GFP⁺CD24⁺CD11c⁺ cDC1. Sorted cDC1 were used in cross-presentation assays.

### Mixed bone marrow chimera
Male B6.SJL mice (CD45.1⁺) at 12 weeks of age were irradiated with a sublethal dose of 6.5 Gy in an RS-2000 irradiator (Rad Source). A mixture of BM cells from B6 (CD45.2⁺) plus BM cells from either WT (CD45.1⁺CD45.2⁺) or Zeb1-dcKO (CD45.1⁺CD45.2⁺) at a ratio of 1:1 (5 × 10⁶ cells in total) were adoptively transferred into irradiated B6.SJL recipients by intravenous injection. All recipients were analyzed 8 weeks after reconstitution.

### ELISA assay
Sera were collected from mice at indicated time points after infection with L. monocytogenes. Concentration of cytokines in serum secretion

was measured by sandwich ELISA using antibody pairs according to the instructions from BioLegend.

## Immunoblot assay

Cells were collected without or after stimulation and were lysed using NP-40 lysis buffer containing 20 mM Tris-HCl (pH7.5), 1% NP-40, 150 mM NaCl, 5 mM EDTA (pH8.0), 1 mM $Na_3VO_4$, 5 mM $Na_4P_2O_7$, 5 mM NaF and protease inhibitor cocktail (Thermo Fisher Scientific). The cell lysates were diluted by 1× SDS loading buffer and denatured by being boiled for 10 min. Samples were separated by SDS-polyacrylamide gel electrophoresis, followed by electrotransfer to polyvinylidene difluoride (PVDF) membranes (Merck Millipore). The membranes were blocked with TBS containing 5% nonfat milk in and 0.1% Tween20 (TBST). After washing with TBST, membranes were analyzed by immunoblot with the appropriate antibodies, followed by horseradish peroxidase-conjugated second antibody (AS003 or AS014; ABclonal) and development with an enhanced chemiluminescence detection reagent (RPN2235; GE healthcare). Images were acquired with Amersham Imager 600 (GE Healthcare).

## Listeria infection

The stock of *L. monocytogenes* (10403 S) was purchased from Nanjing Sungyee Inc. and was inoculated in BHI solid medium. After being cultured in incubator at 37 °C for 12 h, the Listeria single colony was picked and transferred to BHI liquid medium for culture until logarithmic growth stage. Each mouse (sex- and age-matched) at 8 weeks of age was infected with $1 \times 10^6$ CFU or $2.5 \times 10^4$ CFU of Listeria by intravenous (i.v.) injection.

## Histochemistry

Spleens and livers from mice were fixed in 4% paraformaldehyde for 48 h at room temperature, and then were washed in distilled $H_2O$ overnight, followed by fixation, dehydration, embedment. 6 µm sections were cut and mounted on coated slides, and were stained with hematoxylin and eosin according to standard procedures. Slides were dehydrated with two concentrations of ethanol and dipped in xylene, and resin was used to mount coverslips.

## Tumor transplantation

B16F10 tumor cells were propagated in DMEM media supplemented with 10% FBS. Tumor cells were washed three times with PBS and resuspended at a density of $1 \times 10^6$ cells/ml in PBS and then $2 \times 10^5$ cells were injected subcutaneously (s.c.) into the dorsal area of each recipient mouse (sex- and age-matched) at 8–12 weeks of age. Tumor size was measured over time and calculated as length×(width²)/2. Survival was recorded each day. For mouse survival curve analysis, mice with tumor size reaching 2000 mm³ were euthanized with carbon dioxide and were defined as humane endpoints. To analyze the tumor-infiltrating immune cells, tumors were harvested at day 12 to 14.

## Antigen presentation assays

In vivo cross-presentation assays were performed as described previously[68]. $5 \times 10^5$ CFSE-labeled OT-I T cells were injected i.v. into sex- and age-matched mice at 8–12 weeks of age. One day later, $\beta 2m^{-/-}$ splenocytes were osmotically loaded with 10 mg/ml soluble OVA (Sigma), irradiated at 1500 rads, and $5 \times 10^5$ cells were injected i.v. into mice. After 3 days, spleen and pLN were harvested, mashed, and analyzed for CFSE dilution of OT-I T cells. For in vitro cross-presentation, $1 \times 10^4$ sorted cDC1 or cDC2 were sorted from pLNs and mLNs or from day-11-Flt3L-cultured BM cells, and were co-cultured with $2.5 \times 10^4$ CFSE-labeled OT-I or OT-II T cells in the presence of SIINFEKL peptide, or soluble OVA, or HKLM-OVA, or OVA-loaded $\beta 2m^{-/-}$ splenocytes for 3 days, and the OT-I or OT-II T cells were analyzed by flow cytometry for CFSE dilution.

Listeria expressing OVA (LM-OVA) purchased from Nanjing Sungyee Inc. and was grown in BHI broth at 37 °C for 6 h and were then frozen overnight. Then LM-OVA was thawed and washed with DPBS for three times before being heat killed at 80 °C for 1 h, and HKLM-OVA were frozen at −80 °C.

## Phagocytosis assay

$1 \times 10^6$ cDC1 from day-11-Flt3L-cultured BM cells were sorted and incubated with Alexa Flour 488-labeled OVA, or irradiated Alexa Flour 488-labeled OVA-loaded $\beta 2m^{-/-}$ splenocytes, or Alexa Fluor 647-labeled HKLM-OVA at 37 °C for indicated times. After incubation, cDC1 were stained with antibodies against cDC1 markers for flow cytometry.

## Measurement of antigen export to the cytosol

cDC1 sorted from Flt3L-cultured BM cells were resuspended at a concentration of $1 \times 10^7$ cell/ml in loading solution containing CCF4-AM (Cat#K1095; Invitrogen) and were then incubated for 30 min at room temperature protected from light. Then cells were washed with PBS and incubated with β-lactamase-loaded $\beta 2m^{-/-}$ splenocytes or soluble β-lactamase in loading solution at 37 °C for different times. Reactions were stopped with cold PBS. cDC1 were stained with Fixable Viability Dye (Cat# 65-0865-18; eBioscience) and live cells were analyzed by flow cytometry by monitoring the decrease in 525 nm fluorescence and the concomitant increase in 450 nm fluorescence resulting from CCF4 cleavage.

## DQ-Ovalbumin degradation assay

$2 \times 10^5$ cDC1 from day-11-Flt3L-cultured BM cells were incubated with $\beta 2m^{-/-}$ splenocytes osmotically loaded with 10 µg/ml DQ-Ovalbumin (Cat#D12053; Invitrogen) in the presence or absence of MG132 (Cat# HY-13259; MCE) for different times. Then cDC1 were washed with PBS for three times and stained with antibodies against cDC1 markers before analysis for FITC⁺ cells.

## Cytosolic antigen processing after OVA electroporation

$1 - 2 \times 10^6$ cDC1 from day-11-Flt3L-cultured BM cells were resuspended in 100 µl nucleofector solution. Cell suspension were mixed with 2 µg DQ-OVA and Alexa Fluor 647-OVA. The cells were transferred into nucleofector cuvette without air bubbles and were electroporated using Lonza nucleofector (AAB-1001, Program Y-001). Cells were then transferred with 100 µl DC culture medium into 96-well plate and cultured at 37 °C for indicated times. After chasing, cells were stained with antibodies against cDC1 markers before flow cytometry.

## Splenectomy

Splenectomy or sham-operation was performed on mice 6 days before tumor inoculation. Sex- and age-matched mice at 8 weeks of age were anesthetized with pentobarbital sodium, and the spleens were exposed through a left subcostal lateral laparotomy. Then, the hilar vessels were clamped by forceps, the spleens were removed and skin were sutured. Sham-operations only included laparotomy and wound closure.

## Flow Organellocytometry for phagosomal antigen degradation

$5 - 10 \times 10^6$ cDC1 from day-11-Flt3L-cultured BM cells were washed with PBS for three times and then resuspended cells in DC medium to a cell density of $1 \times 10^6$/ml. Cells were incubated with HKLM-OVA antigen at density of $1 \times 10^8$ bacteria per $1 \times 10^5$ cDC1 at 16 °C in a water bath for 20 min. Then stimulation was terminated by adding ice-cold PBS. Cells were washed with PBS twice and resuspended each sample in prewarmed DC medium to allow phagosomal antigen degradation at 37 °C and 5% $CO_2$ incubator for indicated times. Reaction was stopped by adding ice-cold PBS. Cells were blocked with anti-CD16/32 antibody and then were incubated with primary antibodies at 4 °C for 15 min.

After rinsed by PBS, cells were stained with secondary antibodies at 4 °C for 15 min. Then cells were mechanical lysate by a 22 G needle to release non-nuclear organelles from intact cells. Organelles were fixed and permeabilized with BD Cytofix/Cytoperm buffer, and then were incubated with primary antibodies at 4 °C overnight. After rinsed by PBS + 1% BSA (wt/vol), cells were stained with secondary antibodies at 4 °C for 45 min. At last, Organelles were resuspended with PBS for measurement by flow cytometry[69].

## Immunofluorescence staining and confocal microscopy
$2–3 \times 10^6$ cDC1 from day-11-Flt3L-cultured BM cells were stimulated with or without HKLM-OVA, and cells were washed with PBS for three times and then were fixed with pre-warmed 4% paraformaldehyde for 15 min. Then cells were washed with PBS for three times and permeabilized with 0.1% Triton X-100 for 5 min at room temperature. After washed three times with PBS, cells were treated with 100 mM Glycine for 10 min. Cells were rinsed with PBS and then were blocked with PBS containing 5% FBS at room temperature for 30 min. Cells were incubated with primary antibodies at 4 °C overnight. After rinsed by PBS, cells were stained with secondary antibodies at room temperature for 4 to 6 h. After washing with PBS for three times, coverslips were mounted in ProLong™ Gold Antifade Mountant with DAPI (Invitrogen). Confocal images were obtained by Leica TCS SP8 DLS and Zeiss LSM 900+Airyscan2 confocal microscopes.

## Measurement of phagosomal pH
cDC1 sorted from Flt3L-cultured BM cells were pulsed with Alexa Fluor 647-labeled HKLM-OVA (pH insensitive), or FITC-labeled HKLM-OVA (pH sensitive), or pHrodo-labeled HKLM-OVA (pH indicator). Cells were then chased for different times and analyzed by flow cytometry.

## Phago-lysosome fusion assay
cDC1 sorted from Flt3L-cultured BM cells were incubated with 100 μg/ml Alexa Fluor 594-Hydrazide (Cat#A10438; Invitrogen) for 4 h. Then cells were washed with PBS and were incubated without tracer for a further 12 h to allow accumulation of the dye in lysosomes. Cells were resuspended in new DC medium supplemented with Alexa Fluor 488-labeled HKLM-OVA. Then cells were rinsed and resuspended in medium at 37 °C for analysis by spectrofluorometer. As lysosomes fuse with phagosomes, the donor Alexa Fluor 488 comes into close proximity with acceptor Alexa Fluor 594, causing FRET. The degree of lysosomes fused with phagosomes could be calculated from the ratio between the FRET-generated emission at 620 nm and the donor emission at 510 nm using the equation RFU = FRT/DRT-FBO/DBO, where RFU represented relative FRET units, FRT represented FRET-generated fluorescent emission, DRT represented Donor emission, FBO represented FRET signal contribution of HKLM-OVA alone and DBO represented Donor emission of HKLM-OVA alone. Fluorescent measurements were taken every 1 min for 3 h.

## Quantitative PCR
cDC1 sorted from Flt3L-cultured BM cells were challenged with HKLM-OVA for 4 h, and then total RNA was isolated using Qiagen RNeasy Mini Kits (Qiagen) following the manufacturer's protocol. TransScript miRNA First-Strand cDNA Synthesis SuperMix (Cat# AT351-01; Trans) was used to obtain cDNA for miRNA Q-PCR. The primers were as follows: U6 Q-PCR forward, 5′-CGTGAAGCGTTCCATATTTTT-3′; miR-96-5p Q-PCR forward, 5′-GGCACTAGCACATTTTTGCT-3′; miR-467d-5p Q-PCR forward, 5′-AGTGCGCGCATGTATATGCG-3′.

## ROS measurements
Intracellular ROS and mitochondrial ROS were measured by using CellROX™ Deep Red Reagents (Cat#C10422; Invitrogen) and MitoSOX™ Red mitochondrial superoxide indicator (Cat#M36008; Invitrogen), respectively. cDC1 sorted from Flt3L-cultured BM cells

were stimulated with HKLM-OVA for 4 h. Cells were then incubated with 5 μM CellROX Reagent at 37 °C for 30 min or with 5 μM MitoSOX Reagent at 37 °C for 10 min. Then the cells were washed with PBS three times. The fluorescence was measured by flow cytometry. Intraphagosomal ROS was measured using OxyBURST/Alexa Fluor 647-coupled HKLM-OVA. The conjugation of OxyBurst Green $H_2$DCFDA SE (Cat#D2935; Invitrogen) or Alexa Fluor™ 647 NHS Ester (Cat#A37573; Invitrogen) to HKLM-OVA was performed following the manufacturer's instructions. OxyBurst/Alexa Fluor 647-conjugated HKLM-OVA were added to cDC1 in complete medium and were washed with PBS after incubation for 30 min or 90 min. The ratio of OxyBurst+ to Alexa Fluor 647+ cells was calculated as an index of phagosomal ROS production.

## Dual-luciferase reporter assay
WT or mutated fragments (3′end of CDS or 3′UTR) of Cybb cDNA, which contain putative miRNA binding site, were cloned into the psiCHECK2 reporter vector. Mouse miR-96 and miR-467d were cloned into the expression vector pCXN2. The reporter constructs were co-transfected with empty or miRNA expression plasmid into HEK293T cells. 48 h after transfection, cells were lysed and luciferase activity were measured using Dual-luciferase Reporter assay system (Cat#E1910; Promega) according to the manufacture's guide.

## Single cell whole transcriptome mRNA sequencing
Sorted WT and Zeb1-deficient splenic cDC were labeled with sample tags (anti-mouse CD45, Clone#30-F11, BD Biosciences) using BD Single-Cell Multiplexing Kit (Cat#633793; BD Biosciences) following the manufacturer's instructions. Cells were washed and then resuspended in BD Sample Buffer from BD Rhapsody™ Cartridge Reagent Kit (Cat#633731; BD Biosciences) with addition of viability markers (Calcein AM and Draq7). The live cells were counted and combined at 10,000 WT cells and 20,000 Zeb1-dcKO cells, and were loaded onto a BD Rhapsody nanowell cartridge followed by Capture Beads (BD Biosciences). After washing and cell lysis, the cell capture beads were retrieved for reverse transcription following the protocol of Single-Cell Capture and cDNA Synthesis (BD Biosciences). Single cell whole transcriptome and Sample Tag libraries were prepared using the BD Rhapsody Targeted mRNA and AbSeq Amplification Kit (Cat#633774; BD Biosciences), according to the BD Rhapsody System mRNA Whole Transcriptome Analysis and Sample Tag Library Preparation Protocol (BD Biosciences). Estimated the concentration by quantifying 2 μl of the final sequencing library with a Qubit Fluorometer using the Qubit dsDNA HS Kit to obtain an approximate concentration of PCR products to dilute for quantification on an Agilent 2100 Bioanalyzer. The sample Tag library showed a peak of approximate 290 bp. The libraries were sequenced on an Illumnia NovaSeq platform. The FASTQ files of Single cell whole transcriptome mRNA sequencing data was performed following the BD Biosciences Rhapsody pipeline (BD Biosciences). Read pairs with default parameter were filtered the adapter sequence using fastp and removed the low-quality reads to achieve the clean data. UMI-tools were applied for Single Cell Transcriptome Analysis to identify the whitelist of cell barcode. The clean data based on UMI was mapped to Mus musculus genome (mm10) utilizing STAR with customized parameter from UMI-tools standard pipeline to obtain the UMIs counts of each sample. Cells contained over 200 expressed genes and mitochondria UMI rate below 15% passed the cell quality filtering and mitochondria genes were removed in the expression table. We used Seurat package (version 3.0) in R (version 4.1.3) to perform normalization following the expression matrix and calculated cell cycle score using cell cycle gene lists from Itay Tirosh et al. We regression based on the expression matrix according to the UMI counts of each sample and percent of mitochondria rate and cell cycle score to yield the scaled data. Then, the scale data was analyzed by principal component analysis (PCA) with top 1000 high variable genes and top 17 principals were used for uniform manifold

approximation and projection (UMAP) construction. We used graph-based Louvain cluster cluster method (resolution = 0.5) and achieved unsupervised cell cluster results based on the top 17 principals and the marker genes were calculated by FindAllMarkers function with wilcox rank sum test algorithm under default criteria. Cell type clusters were further identified in PCA by FindClusters functions and manually adjusted based on marker gene expression.

## Bulk RNA sequencing and data analysis

cDC1 were sorted from Flt3L-cultured BM cells. At steady state or after stimulation with HKLM-OVA for 4 h, total RNA was isolated using RNeasy Mini Kit (Cat#74104; Qiagen) following the manufacturer's protocol. RNA quality and quantity were measured using the RNA Nano 6000 Assay Kit of the Bioanalyzer 2100 system (Agilent Technologies). mRNA libraries were prepared using NEBNext® UltraTM RNA Library Prep Kit for Illumina® (NEB) following manufacturer's recommendations and index codes were added to attribute sequences to each sample. Briefly, first strand cDNA was synthesized using random hexamer primer and M-MuLV Reverse Transcriptase (RNase H⁻). Second strand cDNA synthesis was subsequently performed using DNA Polymerase I and RNase H. To select cDNA fragments of preferentially 250 to 300 bp in length, the library fragments were purified with AMPure XP system (Beckman Coulter, Beverly). Then USER Enzyme (NEB) was used with size-selected fragments and PCR products were purified (AMPure XP system), and library quality was assessed on the Agilent Bioanalyzer 2100 system. The clustering of the index-coded samples was performed on a cBot Cluster Generation System using TruSeq PE Cluster Kit v3-cBot-HS (Illumia) according to the manufacturer's instructions. After cluster generation, the libraries were sequenced on an Illumnia NovaSeq platform, and 150-bp paired end reads was generated. Raw data (raw reads) of Fastq format were removed adapter to yield clean data. The clean data (clean reads) were mapped to Mus musculus reference genome (mm10) using Hisat2 (version 2.0.5) for each sample. Uniquely mapped reads were counted by FeatureCounts (version 1.5.0-p3). The counts matrix was normalized with size factors using DESeq2. Differential expression analysis of two conditions/groups (two biological replicates per condition) was performed using the DESeq2 R package. Differential expressed genes were found using DESeq2 with cutoffs abs (log2FC > 0.5). Gene set enrichment analysis were performed using clusterProfiler R package.

## miRNA sequencing and data analysis

cDC1 were sorted from Flt3L-cultured BM cells. After stimulation with HKLM-OVA for 4 h, total RNA was isolated using RNeasy Mini Kit (Cat#74104; Qiagen) following the manufacturer's protocol. The small RNA libraries were prepared from 2 μg total RNA from each sample using NEBNext® Multiplex Small RNA Library Prep Set for Illumina® (NEB) according to the manufacturer's instructions. Briefly, purified RNA was mixed with NEB 3′ SR Adapter, then the SR RT Primer hybridized to the excess of 3′SR Adapter (that remained free after the 3′ ligation reaction) and transformed the single-stranded DNA adapter into a double-stranded DNA molecule. 5′ ends adapter was ligated to 5′ ends of miRNAs, siRNA, and piRNA. Then first strand cDNA was synthesized using M-MuLV Reverse Transcriptase (RNase H⁻). PCR amplification was performed using LongAmp Taq 2X Master Mix, SR Primer for Illumina and index (X) primer. PCR products were purified on 8% polyacrylamide gel. DNA fragments corresponding to 140 to 160 bp (the length of small RNA plus the 3′ and 5′ adapters) were recovered and dissolved in 8 μl elution buffer. Library quality was assessed on the Agilent Bioanalyzer 2100 system using DNA High Sensitivity Chips. The clustering of the index-coded samples was performed on a cBot Cluster Generation System using TruSeq PE Cluster Kit v3-cBot-HS

(Illumia) according to the manufacturer's instructions. After cluster generation, the libraries were sequenced on an Illumnia NovaSeq 6000 platform, and 50-bp single end reads was generated. Raw data (raw reads) of Fastq format were firstly processed through SHRiMP2, a short-read mapping program. In this step, clean data (clean reads) were obtained by removing reads containing ploy-N, ploy A or T or G or C, or with 5′ adapter contaminants, or without 3′ adapter or the insert tag. And the 3′ adapter sequences were trimmed. Meanwhile, Q20, Q30, and GC content of the raw data were calculated. Clean reads with a definite length range were chosen for downstream analyses. The micro RNA reads were processed by Bowtie2 version 2.3.4 for read mapping to Mus musculus reference sequence (mm10). This allowed for 1 mismatch base. Processed reads of length at 18 to 35 nt for animals were then mapped to reference genome and analyzed using the bowtie soft. To identify conserved miRNAs, the predicted miRNA hairpins were compared against miRNA precursor sequences from miRBase22.0 using mirDeep2 (version 2.0.0.5) and srna-tools-cli (http://srna-tools.cm.uea.ac.uk/) were used to obtain the potential miRNA and draw the secondary structures. MirDeep2 quantifier.pl was used to obtain the miRNA counts, and custom scripts were used to obtain base bias on the first position of identified miRNA with certain length and on each position of all identified miRNA, respectively. Target gene prediction of miRNA was performed by the TargetScan (version 8.0) database. This database has a series of extensive accepted primary target prediction criteria. At the same time, we also used ENCORI database. ENCORI identifies more than 4.1 million miRNA-ncRNA, 2.9 million miRNA-mRNA, 4.1 million RBP-RNA and 1.5 million RNA-RNA interactions from multi-dimensional sequencing data.

## CUT&Tag sequencing and data analysis

CUT&Tag sequencing libraries were generated using the Hyperactive In-Situ ChIP Library Prep Kit for Illumina (Cat#TD901-01; Vazyme) following the manufacturer's instructions. Briefly, $5-10 \times 10^4$ cDC1 sorted from Flt3L-cultured BM cells of WT mice were stimulated with HKLM-OVA for 4 h. Cells were bound with concanavalin A-coated magnetic beads and then incubated with 1 μg of anti-Zeb1 antibody (Cat#21544-1-AP; Proteintech) or Rabbit IgG Isotype Control (Cat#3900; Cell Signaling Technology) with slow rotation at room temperature for 2 h. After incubation with secondary antibody (1 μg) [goat anti-rabbit IgG H&L (Cat#ab6702; Abcam)] with slow rotation at room temperature for 1 h, cells were washed and incubated with 0.04 μM Hyperactive pG-Tn5 Transposon for 1 h. Cells were washed and resuspended in tagmentation buffer and incubated at 37 °C for 1 h. DNA was purified and then amplified by 15 to 20 PCR cycles to generate libraries. Libraries were sequenced on an Illumina Novaseq 6000 and 150-bp paired-end reads were generated. Paired-end Illumina sequencing was analyzed based on the barcoded libraries following the manufacturer's description. Paired-end reads were aligned to Mus musculus reference genome (mm10) using Bowtie2 version 2.3.4 with options: -end-to-end -very-sensitive -no-mixed --no-discordant --phred33 -I 10 -X 700 -p 2. For maximum economy, up to 96 barcoded samples per 2-lane flow cell could be pooled for 25 × 25 bp sequencing. Peak calling was performed using Professor Li Qiyuan's lab platform server using MACS2 with threshold at p value of 0.05 as a cutoff based on input. For comparison of between wild-type and knock-out peak signals, peaks of identified wild-type and knock-out DC cells were merged with "bedtools" Multiple Intersect" followed by "bedtools MergeBED" function. Peak annotation was carried out using "ChIP-seeker". The Bamcoverage function was used to generate BigWig files, which were normalized to reads per kilobase per million (RPKM). The cumulative curve of peak signals at gene locus was generated using the plotProfile tool based on the bigwig files. The normalized counts of identified peaks were analyzed using the MultiBigwigSummary tool. DESeq2 was applied to differential peak analysis. The fold change of peaks of wild-type DC cells compared with knockout DC cells was

determined by computing the average signal intensity across total peaks detected at a particular gene locus. DiffBind was applied to differential peak analysis. Motif analysis was performed with HOMER.

## Statistical analysis

All data are representative of at least two to three independent experiments. The data were analyzed by two-tailed unpaired Student's *t*-test or two-way analysis of variance (ANOVA) using GraphPad Prism 8 software or R software (version 4.1.3). A Log-rank (Mantel–Cox) Test was used for comparison of survival curves. $P < 0.05$ was considered statistically significant and data are presented as mean ± s.d.

## Reporting summary

Further information on research design is available in the Nature Portfolio Reporting Summary linked to this article.

## Data availability

ScRNA-seq, Bulk RNA-seq (Unstimulated/Stimulated), miRNA-seq and CUT&Tag-seq datasets that support the findings of this study have been deposited in the Gene Expression Omnibus (GEO) (https://www.ncbi.nlm.nih.gov/geo/query/acc.cgi) with accession numbers GSE208311, GSE216059, GSE206963, GSE207161, GSE206964, respectively. All data needed to evaluate the conclusions in the paper are present in the paper and/or the Supplementary Materials. Source data are provided with this paper.

## Code availability

The scripts utilized for the subsequent analysis of ScRNA-seq, Bulk RNA-seq, miRNA-seq, and CUT&Tag-seq datasets have been published and are accessible on GitHub (https://github.com/ClaireZQ/cDC_Zeb1.git).

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

## Acknowledgements

We thank Hongling Huang and all members of the Xiao laboratory for discussion and technical assistance. We thank Lixin Hong, Xiufeng Sun, Xiaohong Ma, Qingfeng Liu, and Lei Huang at the Core Facility of Bio-medical Sciences, Xiamen University and Suqin Wu at the Xiamen University Laboratory Animal Center for technical assistance. This study was supported by the National Natural Science Foundation of China (31970851 to N.X., 82272944 to Q.L., and 81971557 to K.M.) and the Fundamental Research Funds for the Central Universities of China-Xiamen University (20720150065 to N.X., 20720190101 to Q.L., and 20720210001 to K.M.).

## Author contributions

Y.W. designed and executed the experiments, analyzed the data and prepared manuscript; Q.Z., Y.Z. analyzed high-throughput sequencing data under supervision of Q.L.; T.H., Y.C.W., T.L., Z.W., Y.Y.W., S.L. and X.W., performed the experiments; K.Y. and J.X. contributed to miRNA experiments; Y.H., K.M., W.-H.L. and S.-C.C. provided materials and study advices; X.C. helped in confocal microscopy experiments; N.X. conceived the study, designed the experiments, analyzed the data and wrote the manuscript with input from all authors.

## Competing interests

The authors declare no competing interests.
