## [Peer Review File · Nature Communications]

The transcription factor Zeb1 controls homeostasis and function of type 1 conventional dendritic cellsREVIEWER COMMENTS

Reviewer #1 (Remarks to the Author):

The Zeb protein family of transcription factors comprises both Zeb1 and Zeb2, with Zeb2 having a known role in cDC1 versus cDC2 differentiation. The current study by Wang et al. evaluates whether Zeb1 deficiency also impacts cDC differentiation and function by generating mice with conditional deletion of Zeb1 in CD11c⁺ cells. The manuscript describes a large body of work and it could be argued that it would be better presented as two independent papers. Effectively, two distinct stories are presented: one that describes the creation and phenotype of the Zeb1 conditional knockout mice (which have reduced splenic cDC1 numbers due to increased cell death, enhanced *Listeria monocytogenes* infection resistance and decreased antitumor immunity); and a second story that investigates the functional phenotype of the Zeb1 deficient cDCs that remain (compromised cross-presentation to CD8 T cells due to subpar *Cybb* expression via loss of microRNA-96 regulation). Each of the two parts could be better developed and, it could be argued, that the mouse model used is not best suited for the second part. Indeed, the functional competence of cDC1 lacking Zeb1 should be assessed using cells in which the transcription factor in question is acutely ablated after differentiation.

Specific comments:

- 1) Have the authors specifically assessed the impact of Zeb1 deficiency in alveolar macrophage populations in the lung, which also express CD11c?
- 2) Figure 1: are the migratory versus resident population of cDCs equally impacted or unaffected by Zeb1 loss? Have the authors also assessed cell death, apoptosis, and proliferation in these cells, as was done for the splenic cDC1?
- 3) Related to the above point: do the cDC1, and other cDCs present in other tissues, from the Zeb1 conditional knockout mice upregulate Zeb2 expression, which may compensate for the loss of Zeb1?
- 4) The data and interpretation in Figure 1J and 1K are very odd. Did the authors analyse neutrophils (Ly6G⁺) cells from their mixed bone marrow chimeras? These cells should display a frequency of CD45.1 versus CD45.2 cells of 50:50 and could act as quality control for chimerism.
- 5) Line 172: can the authors define what they mean by splenic myeloid cells (what criteria e.g. surface markers)
- 6) Line 250-251: the data do not suggest this. It is more likely that there is less overall dissemination of infection due to loss of splenic cDC1 in Zeb1 deficient mice
- 7) The experiment presented in Figure 4A-B is difficult to interpret owing to the fact that Zeb1 deficient mice already have reduced splenic cDC1 populations and thereby one cannot decipher whether the result stems from reduced cDC1 numbers or defects in antigen presentation.
- 8) With regards to the in vitro antigen presentation assays, do the cDC isolated from Zeb1 knockout animals have deficiencies in their survival (e.g. increased rate of cell death akin to the splenic cDC1 analysed in vivo)? This would impact the interpretation of the cross presentation experiments – a separate model using acute deletion or inhibition of Zeb1 may be more appropriate.
- 9) Why are the cDC2 in Figure 4J just as efficient as cDC1 in Figure 4I at XP of soluble OVA? This result runs counter to dogma.
- 10) The authors spend much time addressing P2C defects in the Zeb1 deficient cDC. Have the authors also checked whether endogenous processing of antigens is faster/slower in the Zeb1 knockout cDCs? This could be assessed using electroporated OVA or a virus encoding OVA
- 11) The complementation experiment shown in Figure 6A is nice. In Figure 6B, can the authors show whether *Cybb* expression is suppressed due to miR96 overexpression?
- 12) Figure 6C: MFI should be quantified with stats
- 13) Figure 7F: change colors of symbols between groups to aid in visualization
- 14) Figure 7F: these data should be quantified and analyzed statistically

Reviewer #2 (Remarks to the Author):

In the manuscript from Wang et al., the authors investigate the role of the Zeb1 transcription factor in the development and function of Type I conventional dendritic cells by using a DC-specific conditional KO mouse of Zeb1. They use these animals to study the gross phenotype of *Listeria monocytogenes* infection or implanted B16 F10 melanoma tumor growth. They further delve into the cellular processes associated with the gross phenotypes and explore antigen processing and presentation in the phagosomal/lysosomal pathways, based on pathways identified from sequencing data. Herein they study a potential connection with the miRNA-96-Cybb axis. Although the manuscript presents a large amount of work that is generally performed well, the manuscript suffers from being overly broad but only thinly supporting the interpretations at many points. The findings are provocative but lack much mechanistic explanation for most of the paper. This is true for the cell biology experiments, but also in the lack of connection between many of the in vitro experiments and the in vivo observations. Owing to the lack of mechanistic work and the thin support for many conclusions, the Discussion section reads as highly speculative in parts. Overall, the work is presented in a confusing fashion and the final interpretations are not well supported. I recommend reorganization and additional development of the mechanistic focus of the paper before the manuscript is considered for publication.

Specific comments and suggestions:

1. In general, many parts of the Results do not fit together as a coherent set of findings and leave a number of observations unaccounted for:

- It is not clear why cDC1 cells are selectively affected in the spleen and not in other lymphoid organs.
- The BM reconstitution experiments do not explain the cell extrinsic effects of Zeb1 KO in cDC1s.
- The mechanism of cDC1 homeostasis is never explained, although that is part of the title of the manuscript. What does antigen processing and presentation have to do with the homeostasis of the cDC1 population?

2. The ZEB1^{fl/fl} CD11c-cre conditional knockout mice. Fig. S1b- what size is Zeb1. The molecular weight should be listed next to blot. How were the immune cells listed in the blot separated out/sorted from total spleen? More details about the Zeb1^{fl/fl} CD11c-cre knockout mice should be included, such as whether they have any obvious phenotypes, survival differences, and whether they are generally more susceptible to infection?

3. Fig.1a-b. The authors state that the absolute number of cDC1 and cDC2 are reduced in the Zeb1-dcKO as compared to the WT in the spleen and that this is due to lower abundance of myeloid cells in the spleen of Zeb1-dcKO mice. However, the authors have not shown overall myeloid cell abundance difference between Zeb1-dcKO and WT. In order to make the statement, myeloid cells need to be evaluated and not just specific DC.

The difference in cDC1 and cDC2 in the spleens of Zeb1-dcKO vs WT is quite drastic and a key basis of this manuscript. The authors should use additional markers to distinguish cDC1/cDC2 in order to be convinced that the trend is not marker specific and truly associated with the cell types. They should also eliminate pDC and moDC in order to ensure that these groups are not contaminating the cDC data. Gating schema, as well as markers used for cDC1/cDC2, should be included.

Gating scheme should be included for Fig1Sc-d. How the authors are defining each population also needs to be clearly stated. Neutrophils (LyG+ etc), inflammatory monocytes (Ly6C+), etc.

How were cDC1/cDC2 designated in the scRNA seq? What markers were used to define these clusters? The authors should show this data in a way that conveys not only the changes in the cDC1/cDC2 between the two groups but also the statistical difference.

The authors should have more caution when making the statement that "Zeb1 deficiency in DCs impeded the generation of cDC1 in the spleen...". So far it has been shown that Zeb1 deficiency in the DC is associated with a decrease in cDC1 but how that is happening has not been investigated. Maybe Zeb1 deficiency is preventing differentiation from occurring, or if they do differentiate they are not very long-lived, or unable to proliferate. At this point in the manuscript, these other possibilities have not been investigated, so the conclusion from this data should be broader.

The scRNA seq data and statements in Fig S2a are confusing. Based on Fig 1g it was said that there was a difference in the amount of cDC1 between WT and Zeb1-dcKO. However, based on the

markers that define cDC1 in figure S2a there is no difference between WT and Zeb1-dcKO. How do you account for this discrepancy? Genes for cDC1 should be clearly different than genes for cDC2. You should clarify what genes were used to define cDC1/cDC2 from Fig 1g and which are used in fig S2a.

The annexin/PI (Fig1h) and caspase experiments (figS2d) are conflicting. How many times were each experiment repeated? Cells that have undergone apoptosis will have PS on their surface and hence how annexin binds. So it appears that these cells are undergoing apoptosis. Were the conditions different in any way between experiments? These inconsistencies make one question the conclusions.

One of the major observations by the authors is that conditional loss of Zeb1 in DCs lead to a specific loss of only splenic cDC1 subset, without any change in the other DC subsets from other sources. This is a very provocative finding, but the mechanistic underpinnings have not been addressed by the authors in any capacity. Some experimental insights will greatly improve the value of this work.

4. Figure 2. How is it that loss of Zeb1 in cDC1 enhances immune defense? Is this result specific to *L. monocytogenes*? Host defense should be tested in another way to confirm the results. What would happen if the opposite were true. Overexpression of Zeb1 in cDC1 would that cause mice to become over sensitive and die at much lower doses?

Histological analysis needs to be statistically evaluated in order to confirm that images taken were not an exception. How many images were taken per mouse?

Pro-inflammatory cytokines were tested (Fig. 2d) and showed a significant increase with the Zeb1-dcKO mice vs WT. Was the opposite trend observed for anti-inflammatory cytokines? How long were the differences in cytokine expression maintained for?

5. Figure 3. There is statistical difference between tumor growth and the survival of WT and Zeb1dcKO mice bearing B16 tumors. Zeb1dcKO have tumors that grow much faster and survival is much lower than WT. So clearly the effect would seem to be occurring within the tumor. However, the levels of cDC1/cDC2 are not different--why? Essentially the differences in B16 tumor growth/survival are attributed to cDC1/cDC2 differences only in the spleen. If this is the case then you need to be very clear about where the differences are actually occurring and how this works mechanistically.

The authors have shown a strong decrease in the activation of CD8+ CTLs, but have not looked at T-reg or ICOS+ CD4 + T cells. They may have a direct consequence in the inhibition of maturation of the CD8+ CTLs in the dcKO tumors. Further, looking at exhaustion state of CD8+ T cells (PD1/Tim3) could also provide insight as to the mechanism of reduced abundance of activated CTLs.

6. Figure 5-7. The statement that there is a strong reduction of Cybb and slight reduction of Ncf2 in the western blot Fig.5g needs to be accompanied with numerical values. The only obvious differences are with Zeb1. The data from RNAseq, in particular the statements made in line 343-353 need to be validated by another experiment. Manipulation of Zeb1 should be performed to evaluate the phagosome/lysosome pathways changes.

The Cut&Tag and chip-seq analysis are very interesting. However, no further validation of this data was performed to really narrow down the results and tie them back to the hypothesis. The conclusions need to be validated using a secondary experiment.

The targeting of miR-96 needs to be clarified further—was this performed with the miR-96 family cluster or with miR-96 specifically. This should be clarified and updated in the manuscript. Direct regulation of Cybb by m96cl has not been validated by performing WB/q-RT-PCR to look at expression of Cybb after exogenous expression or repression of m96cl in the cells. For example, multiple groups have demonstrated that Zeb1 regulates mRNA's and miRNA's throughout the genome in a concordant fashion. The authors do not seem to have analyzed the data in an unbiased fashion to understand the mechanistic effects, but rather have chosen miR-96 as a convenient potential mechanistic intermediate.

Although the authors go into mechanistic detail for the Zeb1-miR-96-Cybb axis as a potential point of control for cDC1 function, it is not soundly established that this is the core mechanism for alteration of antigen presentation. From the sequencing data, alterations in Zeb1 appear to be rather pleiotropic in nature, with both direct binding and indirect effects across the genome. The focus on just the one indirect effect on Cybb, which appears quantitatively weak, is unconvincing. I

think that the authors' statement in lines 353-355 is probably correct, but the manuscript does not explore the expression of a series of genes by Zeb1 that promote cross-presentation. Additionally, the authors do not demonstrate that the phagolysosomal and antigen export/presentation machinery effects of Zeb1 found in vitro account for the in vivo infection or tumor phenotypes in any way.

Quantitation of Fig.7f. In order to make conclusions quantitation of staining must be performed with applied statistics.

7. The manuscript would benefit from improvement in the English usage throughout. This would enhance the readability.

Reviewer #3 (Remarks to the Author):

Manuscript Nr: NCOMMS-22-52757

Wang et al., "The transcription factor Zeb1 controls homeostasis and function of type 1 conventional dendritic cells"

The authors demonstrate that Zeb1 suppresses miR-96 which in turn suppresses the Cybb subunit of NADPH oxidase 2 (NOX2). NOX2 activity is therefore reduced in Zeb1 deficient classical dendritic cells type 1 (cDC1) reducing their antigen cross-presenting capacity but also presentation of endocytosed antigens on MHC class II molecules. This results in increased splenic cDC1 cell death, their increased resistance to *Listeria* but decreased ability to induce anti-cancer immune responses. Interestingly although ovalbumin (ova) escape to the cytosol is inhibited due to Zeb1 deficiency there is no effect on phagosomal acidification. On the contrary phagocytosed antigen seemed to be more rapidly delivered to lysosomes. From these data the authors suggest that Zeb1 is required to optimize the cross-presentation capacity of cDC1s, but it remains unclear why these cells also present antigens less efficiently on MHC class II and die in the spleen.

This is an interesting study on the role of Zeb1 for cross-presentation by cDC1s but some additional information on why Zeb1 is also required for endocytosed antigen presentation on MHC class II molecules seems to be required.

Major comments:

1. How do Zeb1 deficient cDCs1 influence the survival of Zeb1 positive cDC1s in trans? The authors suggest that Zeb1 deficient cDC1s die and are taken up by Zeb1 positive cDC1s, causing their cell death. Can they also demonstrate this in vitro?
2. Why does Zeb1 deficiency also decrease MHC class II restricted antigen presentation? The authors argue in their discussion that NOX2 attenuates antigen degradation which they suggest is also beneficial for MHC class II presentation. However, the authors do demonstrate that acidification is intact as well as fusion with lysosomes. Is endocytosed ova less efficiently transported to the MHC class II containing compartment (MIIC) for loading?
3. The authors did not detect changes in phagosomal acidification in Zeb1 deficient cDC1s. Did they detect changes in phagocytosed ovalbumin (ova) maintenance?
4. The authors interpret their proteasome inhibition experiments as indication that Zeb1 deficiencies cripples proteasomal degradation. However, if less ova reaches the cytosol, less is also degraded by the proteasome. The interpretation should be reconsidered.
5. Why does Zeb1 deficiency also decrease CD4+ T cell numbers in the studied tumor model (B16F10)? Do cDC2s not compensate for cDC1s during CD4+ T cell priming? This should at least be discussed.

Minor comments:

1. Some typos, e.g. line 533 and 607, deficient instead of deficient,

Point by point response to reviewers

We thank the insightful and constructive comments from all three reviewers and the editorial office. Following the suggestions by the reviewers, we have performed new experiments and provided a large amount of new information in the revised manuscript. In particular, we have provided more evidence to support that selective reduction of splenic cDC1s in *Zeb1*-dcKO mice was caused by excessive cell death. We have also established the role of miR-182 in cross-presentation of cDC1s. Moreover, we have examined phagosomal antigen degradation in WT and *Zeb1*-deficient cDC1s after phagocytosis. The major changes are highlighted in a red underlined font in the revised manuscript, including the legends of the figures and supplementary figures. With these new results, we believe we have addressed almost all concerns from the reviewers, and improved the mechanistic insight and overall quality of our work.

A brief summary of the major new results is listed below:

1. Alveolar macrophages in bronchoalveolar lavage fluid and lung in *Zeb1*-dcKO mice (Supplementary Fig. 1f-g).
2. The generation of neutrophils in mixed bone marrow chimeric mice (Fig. 1l-m).
3. Cell death of cDC1s when cocultured with apoptotic or necroptotic splenocytes (Supplementary Fig. 2h-i).
4. IL-10 secretion into mice sera after *Listerial* infection (Fig. 2d).
5. Exhausted T cells and Treg cells in B16F10 tumors of tumor-bearing mice (Supplementary Fig. 4c-f).
6. Cell death of cDC1s during cross-presentation of HKLM-OVA (Supplementary Fig. 5a-b).
7. Parallel comparison of capacity to cross-present soluble OVA to OT- I T cells by cDC1s and cDC2s (Fig. 4i-j).
8. Validation of protein expression of *Zeb1* target genes *Ap1b1* and *Ap1m2* in WT and *Zeb1*-deficient cDC1s upon stimulation of HKLM-OVA by western blot (Supplementary Fig. 6d).
9. The effect of miR-182 overexpression on *Cybb* expression (Dual luciferase reporter assay (Fig. 5k) and western blot (Supplementary Fig. 7e)) and cross-presentation (Fig. 6b).
10. Examination of phagosomal antigen degradation in cDC1s by using flow organelloctometry (Fig. 7i-j).

We have included additional results in the additional Figures 1-3 to address reviewers' questions. We prefer not to include these data in the manuscript because of space constraint, but we will be happy to include them if the reviewers or the editors think it is necessary.

Reviewer #1 (Remarks to the Author):

The Zeb protein family of transcription factors comprises both Zeb1 and Zeb2, with Zeb2 having a known role in cDC1 versus cDC2 differentiation. The current study by Wang et al. evaluates whether Zeb1 deficiency also impacts cDC differentiation and function by generating mice with conditional deletion of Zeb1 in CD11c⁺ cells. The manuscript describes a large body of work and it could be argued that it would be better presented as two independent papers. Effectively, two distinct stories are presented: one that describes the creation and phenotype of the Zeb1 conditional knockout mice (which have reduced splenic cDC1 numbers due to increased cell death, enhanced *Listeria monocytogenes* infection resistance and decreased antitumor immunity); and a second story that investigates the functional phenotype of the Zeb1 deficient cDCs that remain (compromised cross-presentation to CD8 T cells due to subpar *Cybb* expression via loss of microRNA-96 regulation). Each of the two parts could be better developed and, it could be argued, that the mouse model used is not best suited for the second part. Indeed, the functional competence of cDC1 lacking Zeb1 should be assessed using cells in which the transcription factor in question is acutely ablated after differentiation.

Response: We thank this reviewer for the encouraging comments and insightful suggestions to revise this manuscript. We have made extensive efforts to address the raised issues by performing additional experiments and by providing explanations as described below. Based on the guidance from the editorial office, we did not divide this study into two separate manuscripts.

Specific comments:

- 1) Have the authors specifically assessed the impact of Zeb1 deficiency in alveolar macrophage populations in the lung, which also express CD11c?

Response: This is a very good suggestion. Alveolar macrophage (AM) is a distinct lineage from dendritic cells although it also expresses CD11c. As suggested, we assessed AM populations in both BAL fluid and lung of WT and Zeb1-dcKO mice. As shown in new supplementary Fig. 1f-g, Zeb1-dcKO mice showed comparable numbers of AMs in both BAL fluid and lung with wild-type counterparts. This result suggested that Zeb1 was not required for the generation of AM.

- 2) Figure 1: are the migratory versus resident population of cDCs equally impacted or unaffected by Zeb1 loss? Have the authors also assessed cell death, apoptosis, and proliferation in these cells, as was done for the splenic cDC1?

Response: This is a very good point. We did not observe clearly distinct migratory versus resident population of cDCs based on the expression of CD11c and MHC-II in both pLNs and mLNs of naive WT and Zeb1-dcKO mice that were not immunized. Nevertheless, as suggested, we assessed cell death, cleaved Caspase-3 in cDC1s and cDC2s in these lymphoid tissues from WT and Zeb1-dcKO mice, although we have already found no defect in the frequencies and absolute numbers of these populations (Fig. 1a-b). Consistently, we did not detect significant difference in cell death or apoptosis of cDC1s and cDC2s between two genotypes (Additional Fig. 1). Moreover, we did observe clearly distinct migratory versus resident population of cDCs in tumor draining lymph nodes (dLNs) of tumor-bearing mice. However, both migratory and resident cDC1s were normally presented in tumor dLNs of tumor-bearing Zeb1-dcKO mice (Fig. 3e-f). All these data clearly suggested that Zeb1 deletion in CD11c⁺ cells reduced cDC1s only in the spleen but not in the other tissues.

Additional Figure 1. Cell death of cDC1s or cDC2s in pLNs and mLN. (a) Flow cytometry of pLN and mLN cDC1s (top row) and cDC2s (middle row) from WT and Zeb1-dcKO mice. Numbers adjacent to outlined areas indicate percent dying (AnnexinV⁺7-AAD⁺) cDC1s or cDC2s. Frequencies of dying pLN and mLN cDC1s and cDC2s among total cDC1s or cDC2s (bottom row) (n=3-6 per group). (b) Flow cytometry of pLN and mLN cDC1s (top row) and cDC2s (middle row) from WT and Zeb1-dcKO mice. Numbers adjacent to outlined areas indicate percent active Caspase-3⁺ cells among pLN or mLN cDC1s or cDC2s. Frequencies of active Caspase-3⁺ cells among pLN or mLN cDC1s or cDC2s (bottom row) (n=3-6 per group).

- 3) Related to the above point: do the cDC1, and other cDCs present in other tissues, from the Zeb1 conditional knockout mice upregulate Zeb2 expression, which may compensate for the loss of Zeb1?

Response: Previous study have showed that Zeb1 and Zeb2 mutually repress each other's expression¹. Following the reviewer's suggestion, we analyzed our RNA-seq data and found that Zeb2 expression was increased in steady Zeb1-deficient cDC1s, which is consistent with previous study. However, Zeb2 expression was dramatically decreased in Zeb1-deficient cDC1s after stimulation with HKLM-OA (Additional Fig. 2a). It has been reported that Zeb2 switches the DC fate specification from cDC1 to pDC or cDC2 by antagonizing Id2 expression (Ref. 33: Wu X. et al., PNAS 2016; Ref. 34: Scott C.L. et al., J Exp Med 2016), so it seems that upregulation of Zeb2 expression in steady Zeb1-deficient cells could not compensate the effect of Zeb1 deletion on cDC1 generation. We have analyzed the cDC populations in Zeb1/2 double conditional knockout mice (*Zeb1^{fl/fl} Zeb2^{fl/fl} CD11c-Cre, Zeb1/2-dcKO*), although we did not get enough mice. Like Zeb1-dcKO mice, Zeb1/2-dcKO mice also had much fewer splenic cDC1s than WT mice (Additional Fig. 2b).

Additional Figure 2. The relation of Zeb1 and Zeb2 in cDC1s. (a) Zeb2 expression in FIt3L-cDC1 from above bulk RNA-seq between WT and Zeb1-dcKO mice at steady state and after stimulation with HKLM-OVA for 4 h. (b) Flow cytometry of live Lin⁻CD317⁻Ly6C⁻CD11c⁺MHC II⁺ cDCs in spleen from WT, Zeb1-dcKO and Zeb1/2-dcDKO mice. Numbers adjacent to outlined areas indicate percent XCR1⁺SIRPα⁻ cDC1s or XCR1⁻SIRPα⁺ cDC2s.

- 4) The data and interpretation in Figure 1J and 1K are very odd. Did the authors analyze neutrophils (Ly6G⁺) cells from their mixed bone marrow chimeras? These cells should display a frequency of CD45.1 versus CD45.2 cells of 50:50 and could act as quality control for chimerism.

Response: We thank the reviewer for pointing out this issue. Although the data in Fig. 1j-k looks odd, it is indeed true as we repeated this experiment several times. As suggested by the reviewer, we regenerated mixed bone marrow chimeric mice as described in Fig. 1j-k and analyzed neutrophils in spleens of these chimeras. Again, we observed severely reduced generation of splenic cDC1s from both Zeb1-deficient (CD45.1⁺CD45.2⁺) and B6 (CD45.2⁺) BM cells in the same recipient. In this recipient, both Zeb1-deficient and B6 BM cells reconstituted comparable frequency of CD11b⁺ myeloid cells and repopulated comparable frequency of neutrophils (CD11b⁺ Ly6G⁺) within CD11b⁺ myeloid cells, as compared to WT (CD45.1⁺CD45.2⁺) and B6 (CD45.2⁺) BM cells together in the same other recipient (new data Fig. 1l-m). A previous study has reported that cDC1s rather than other myeloid cells can be selectively depleted by apoptosis triggered by translocation of internalized cytochrome c (cyt c) into the cytoplasm (Ref. 59), so we speculated that cDC1s from B6 BM cells might undergo apoptosis following uptake of dying cDC1s from Zeb1-deficient BM cells in the spleen of mixed BM chimera, resulting in similar loss of both Zeb1-sufficient and -deficient splenic cDC1s. As suggested by the Reviewer #3, we demonstrated that Zeb1-sufficient splenic cDC1s from B6 mice underwent much more cell death when cocultured with apoptotic or necroptotic splenocytes than live splenocytes (new supplementary Fig. 2h-i).

- 5) Line 172: can the authors define what they mean by splenic myeloid cells (what criteria e.g. surface markers)

Response: Here we defined splenic myeloid cells as non-T/non-B (CD3⁻CD19⁻) splenocytes. Please see the gating strategy shown in supplementary Fig. 9a. We have also included the statistic of splenic myeloid cell numbers in new supplementary Fig. 1c.

- 6) Line 250-251: the data do not suggest this. It is more likely that there is less overall dissemination of infection due to loss of splenic cDC1 in Zeb1 deficient mice

Response: We thank the reviewer for pointing out this issue. Indeed, a previous study have already demonstrated that efficient Listerial spread in liver was mediated by splenic cDC1s (Ref. 40: Edelson B.T. et al., Immunity 2011). We agree with the reviewer and have re-written the statement: Although the decreased Listeria burden in the liver can be attributed to the severe reduction of splenic cDC1s in Zeb1-dcKO mice, it is necessary to examine the capability of cross-presentation in residual cDC1s in these mice.

- 7) The experiment presented in Figure 4A-B is difficult to interpret owing to the fact that Zeb1 deficient mice already have reduced splenic cDC1 populations and thereby one cannot decipher whether the result stems from reduced cDC1 numbers or defects in antigen presentation.

Response: We thank the reviewer for raising the important point. Although Zeb1-dcKO mice possessed normal cDC1 populations in other lymphoid tissues (peripheral lymph node (pLN) and thymus) and non-lymphoid tissues (liver and lung), we can't ignore the migration of cDC1s. So, we have re-made this statement: these data could not exclude the possibility that in vivo cross-presentation of cell-associated antigens by cDC1s from pLNs was attenuated by the absence of Zeb1.

- 8) With regards to the in vitro antigen presentation assays, do the cDC isolated from Zeb1 knockout animals have deficiencies in their survival (e.g. increased rate of cell death akin to the splenic cDC1 analysed in vivo)? This would impact the interpretation of the cross-presentation experiments – a separate model using acute deletion or inhibition of Zeb1 may be more appropriate.

Response: We thank the reviewer for raising the very important point. We examined the cell death of cDC1s during cross-presentation by staining Annexin V/7-AAD and counted the live cDC1s every day. As shown in new supplementary Fig. 5a-b, there was not significant difference in the cell death and survival between WT and Zeb1-deficient cDC1s during cross-presentation. So, we concluded that the defect of cross-presentation was unrelated to the survival of cDC1s.

- 9) Why are the cDC2 in Figure 4J just as efficient as cDC1 in Figure 4I at XP of soluble OVA? This result runs counter to dogma.

Response: We thank the reviewer for pointing out this issue. The data in old Fig. 4i and Fig. 4j were from different batches of experiments. We have performed parallel experiments to compare the capacity to cross-present soluble OVA by cDC1s and cDC2s. WT cDC1s actually exhibited stronger capacity to cross-present soluble OVA than WT cDC2s, while *Zeb1*-deficient cDC1s showed partially reduced efficiency for cross presentation of soluble OVA at the level similar to that of WT and *Zeb1*-deficient cDC2s. We replaced the old results with these new results in Fig. 4i-j.

- 10) The authors spend much time addressing P2C defects in the Zeb1 deficient cDC. Have the authors also checked whether endogenous processing of antigens is faster/slower in the Zeb1 knockout cDCs? This could be assessed using electroporated OVA or a virus encoding OVA.

Response: We apologize that we have not done experiment to assess endogenous antigen process by proteasome. Instead, we have investigated phagosomal antigen

degradation in individual phagosomes of cDC1s after phagocytosis by using flow organelloctometry. We found that the kinetics and efficiency of phagosomal OVA degradation were much faster in Zeb1-deficient cDC1s than in WT cDC1s after phagocytosis of HKLM-OVA (new Fig. 7i-j).

11) The complementation experiment shown in Figure 6A is nice. In Figure 6B, can the authors show whether Cybb expression is suppressed due to miR96 overexpression?

Response: We thank the reviewer for the positive comment. As suggested by the reviewer, we examined the protein level of Cybb in cDC1s after transduction with retroviral miR-96 or miR-182. We observed that miR-96 overexpression significantly but miR-182 overexpression only slightly suppressed protein expression of Cybb (new supplementary Fig. 7e), which is consistent with the results from dual luciferase reporter assay (Fig. 5k).

12) Figure 6C: MFI should be quantified with stats

Response: As suggested by the reviewer, we quantified the mean fluorescence intensity (MFI) of CellROX and MitoSOX in these experiments and included the statistic in Fig. 6c.

13) Figure 7F: change colors of symbols between groups to aid in visualization

Response: We moved the color symbols out of the images to the margin to aid in visualization in Fig. 6g, Fig. 7f-g.

14) Figure 7F: these data should be quantified and analyzed statistically

Response: As suggested by both reviewer #1 and reviewer #2, we quantified the frequencies of colocalization events within HKLM⁺ cells in the images and included the statistic in Fig. 7f.

Reviewer #2 (Remarks to the Author):

In the manuscript from Wang et al., the authors investigate the role of the Zeb1 transcription factor in the development and function of Type I conventional dendritic cells by using a DC-specific conditional KO mouse of Zeb1. They use these animals to study the gross phenotype of *Listeria monocytogenes* infection or implanted B16 F10 melanoma tumor growth. They further delve into the cellular processes associated with the gross phenotypes and explore antigen processing and presentation in the phagosomal/lysosomal pathways, based on pathways identified from sequencing data. Herein they study a potential connection with the miRNA-96-Cybb axis. Although the manuscript presents a large amount of work that is generally performed well, the manuscript suffers from being overly broad but only thinly supporting the interpretations at many points. The findings are provocative but lack much mechanistic explanation for most of the paper. This is true for the cell biology experiments, but also in the lack of connection between many of the in vitro experiments and the in vivo observations. Owing to the lack of mechanistic work and the thin support for many conclusions, the Discussion section reads as highly speculative in parts. Overall, the work is presented in a confusing fashion and the final interpretations are not well supported. I recommend reorganization and additional development of the mechanistic focus of the paper before the manuscript is considered for publication.

Response: We thank this reviewer for the encouraging comments and insightful suggestions to revise this manuscript. We have made extensive efforts to address the raised issues by performing additional experiments, by acquiring more information from scRNA-seq data, and by providing explanations as described below.

Specific comments and suggestions:

1. In general, many parts of the Results do not fit together as a coherent set of findings and leave a number of observations unaccounted for:

- It is not clear why cDC1 cells are selectively affected in the spleen and not in other lymphoid organs.
- The BM reconstitution experiments do not explain the cell extrinsic effects of *Zeb1* KO in cDC1s.
- The mechanism of cDC1 homeostasis is never explained, although that is part of the title of the manuscript. What does antigen processing and presentation have to do with the homeostasis of the cDC1 population?

Response: We thank the reviewer for raising these critical points.

As for selective reduction of splenic cDC1s, we found it was caused by excessive cell death, although we have not uncovered the mechanism how cDC1 cells are selectively affected in the spleen and not in other lymphoid organs. This is an important biological topic. However, we sincerely believe that it is out of the scope of this manuscript. The spleen is the largest second lymphoid organ that specializes in filtering blood and trapping blood-borne pathogens or antigens. cDC1s localize in the marginal zone and red pulp of the spleen where they are readily exposed to blood-borne pathogens and dead cells. Whether excessive death of *Zeb1*-deficient splenic cDC1s is induced by blood-borne stimuli, needs to be further investigated.

By using mixed bone marrow chimera experiments, we demonstrated that the effect of *Zeb1* deletion on splenic cDC1s was cell-intrinsic and even dominant as it also affected *Zeb1*-sufficient splenic cDC1s but did not affect other myeloid cells including cDC2s and neutrophils in the same host. We observed severely reduced generation of splenic cDC1s from both *Zeb1*-deficient (CD45.1⁺CD45.2⁺) and B6 (CD45.2⁺) BM cells in the same recipient. In this recipient, both *Zeb1*-deficient and B6 BM cells reconstituted comparable frequency of CD11b⁺ myeloid cells and repopulated comparable frequency of neutrophils (CD11b⁺ Ly6G⁺) within CD11b⁺ myeloid cells, as compared to WT (CD45.1⁺CD45.2⁺) and B6 (CD45.2⁺) BM cells together in the same other recipient (new data Fig. 11-m). As cDC1s rather than other myeloid cells can be selectively depleted by apoptosis triggered by translocation of internalized cytochrome c (cyt c) into the cytoplasm (Ref. 59), we speculated that cDC1s from B6 BM cells might undergo apoptosis following uptake of dying cDC1s from *Zeb1*-deficient BM cells in the spleen of mixed BM chimera, resulting in similar loss of both *Zeb1*-sufficient and -deficient splenic cDC1s. As suggested by the Reviewer #3, we demonstrated that *Zeb1*-sufficient splenic cDC1s from B6 mice underwent much more cell death when cocultured with apoptotic or necroptotic splenocytes than live splenocytes (new supplementary Fig. 2h-i). Therefore, the unique pathway of antigen export to cytosol in cross-presentation of cDC1s can explain why the deletion of *Zeb1*

in dendritic cells also affected *Zeb1*-sufficient splenic cDC1s but did not affect other myeloid cells including cDC2s and neutrophils in the same host.

2. The ZEB1^{fl/fl} CD11c-cre conditional knockout mice. Fig. S1b- what size is Zeb1. The molecular weight should be listed next to blot. How were the immune cells listed in the blot separated out/sorted from total spleen? More details about the Zeb1^{fl/fl} CD11c-cre knockout mice should be included, such as whether they have any obvious phenotypes, survival differences, and whether they are generally more susceptible to infection?

Response: As suggested by the reviewer, we labeled molecular size for the protein markers used in the western blot in supplementary Fig. 1b. We sorted these immune cells from total splenocytes after digestion by Collagenase D and DNase I. Briefly, spleens were dissected from WT and Zeb1-dcKO mice, and were minced with dissection scissors, then were digested in RPMI 1640 containing 1 mg/ml Collagenase D, 20 ng/ml DNase I for 30 min at 37°C with shaking. After digestion, the tissues were then mashed and filtered. Red blood cells were removed by ACK (Ammonium-Chloride-Potassium) lysis buffer. Single cell suspensions were then washed, filtered and then collected by centrifugation.

We have already investigated T cell development in the thymus, peripheral T cells in lymphoid tissues (LN and spleen), pre-DC development in the bone marrow and spleen, myeloid cell compartments in lymphoid (thymus, spleen and LNs) and non-lymphoid tissues (liver and lung) in Zeb1-dcKO mice (Fig. 1 and supplementary Fig. 1). Zeb1-dcKO mice had no visible phenotype and survive normally (Data not shown). In addition to *Listerial* infection, we also infected Zeb1-dcKO mice with LCMV Armstrong strain. At day 8 after infection, the frequencies and absolute numbers of total CD8⁺ T cells and activated (CD44⁺) CD8⁺ T cells were substantially decreased in Zeb1-dcKO mice as compared with that in WT mice (Additional Fig. 3a), while the generation of activated (CD44⁺) CD4⁺ T cells, GC B cells and plasma cells was not significantly affected in Zeb1-dcKO mice (Additional Fig. 3a-b). The selective effect on CD8⁺ T cells was also observed in tumor-infiltrating T cells of tumor-bearing mice (Fig. 3g-j). These results pointed out a potential role of Zeb1 in cross-presentation of cDC1s.

Additional Figure 3. The T cell and B cell responses after LCMV infection. (a) Flow cytometry of leukocytes (top row), activated CD4⁺ T cells (middle row), activated CD8⁺ T cells (bottom row) in spleen from WT and Zeb1-dcKO mice at day 8 after intraperitoneal injection with 1.5×10⁵ pfu LCMV Armstrong strain. (b) Frequencies and numbers of CD4⁺ or CD8⁺ T cells (top row) and CD44⁺CD4⁺ (among CD4⁺ T cells) or CD44⁺CD8⁺ (among CD8⁺ T cells) activated T cells (bottom row) in spleen from mice as in a (WT, n=6; Zeb1-dcKO, n=5). (c) Flow cytometry of B220⁺ B cells in spleen from mice as in a. (j) Frequencies and numbers of GC (Fas⁺GL7⁺) B cells and plasma cells (IgD⁺CD138⁺) among B220⁺ B cells in spleen from mice as in c (WT, n=6; Zeb1-dcKO, n=5). Each

symbol represents an individual mouse, small horizontal lines indicate the mean (\pm s.d.). *P<0.05, **P<0.01, ***P<0.001; ns, not significant (two-tailed unpaired Student's t-test).

3. Fig.1a-b. The authors state that the absolute number of cDC1 and cDC2 are reduced in the Zeb1-dcKO as compared to the WT in the spleen and that this is due to lower abundance of myeloid cells in the spleen of Zeb1-dcKO mice. However, the authors have not shown overall myeloid cell abundance difference between Zeb1-dcKO and WT. In order to make the statement, myeloid cells need to be evaluated and not just specific DC.

Response: As suggested by the reviewer, we have included the statistic of splenic myeloid cell numbers in new supplementary Fig. 1c, which showed the numbers of splenic myeloid cells were decreased in Zeb1-dcKO mice.

The difference in cDC1 and cDC2 in the spleens of Zeb1-dcKO vs WT is quite drastic and a key basis of this manuscript. The authors should use additional markers to distinguish cDC1/cDC2 in order to be convinced that the trend is not marker specific and truly associated with the cell types. They should also eliminate pDC and moDC in order to ensure that these groups are not contaminating the cDC data. Gating schema, as well as markers used for cDC1/cDC2, should be included.

Gating scheme should be included for Fig1Sc-d. How the authors are defining each population also needs to be clearly stated. Neutrophils (LyG+ etc), inflammatory monocytes (Ly6C+), etc.

Response: As suggested by the reviewer, we presented the gating strategies of all flow cytometry data in new supplementary Fig. 9. We have already excluded Ly6C⁺ cells containing pDC and moDC during analysis of flow cytometric data. We have also illustrated all the surface markers to identify each cell types in the figure legends.

How were cDC1/cDC2 designated in the scRNA seq? What markers were used to define these clusters? The authors should show this data in a way that conveys not only the changes in the cDC1/cDC2 between the two groups but also the statistical difference.

Response: We defined cDC1/cDC2 using the canonical cDC1 signature genes (*Xcr1*, *Clec9a*, *Irf8*, *CD8a*, *Itgae*, etc.) and cDC2 signature genes (*Sirpa*, *CD209a*, *Irf4*, *Esam*, *Itgam*, *Zeb2*, etc.) (Ref. 32 and 34), and DC3 signature genes (*Ccl22*, *Cxcl16*, *CD83*, *CD86*, *Adam8*, etc.)² as shown in additional Fig. 4. To clearly demonstrate the reduction in frequency of Zeb1-deficient splenic cDC1s, we performed kernel density estimation with our scRNA-seq data using the `plot_density` function from the R package *Nebulosa* to quantify cells with high expression of indicated signature genes and presented in feature plots with the statistic as shown in supplementary Fig. 1j and Table S1. These results clearly showed that the frequencies of the cells expressing high level of cDC1 signature genes were severely decreased in splenic cDC sample from Zeb1-dcKO mice.

Additional Figure 3. Heat maps show unique molecular identifier (UMI) counts of selected genes, with key indicating sample type of origin.

The authors should have more caution when making the statement that “Zeb1 deficiency in DCs impeded the generation of cDC1 in the spleen...”. So far it has been shown that Zeb1 deficiency in the DC is associated with a decrease in cDC1 but how that is happening has not been investigated. Maybe Zeb1 deficiency is preventing differentiation from occurring, or if they do differentiate they are not very long-lived, or unable to proliferate. At this point in the manuscript, these other possibilities have not been investigated, so the conclusion from this data should be broader.

Response: We thank the reviewer for pointing out this issue. We observed that the frequencies and numbers of cDC1 was decreased in spleen of Zeb1-dcKO mice. However, the pre-DC development and in vitro DC development of Zeb1-deficient BM cells was almost intact (supplementary Fig. 1k-p). So, we concluded in the next paragraph that Zeb1 might control the homeostasis rather than the development of splenic cDC1s. In this paragraph, following the reviewer’s suggestion, we changed the conclusion as following: these data suggested that Zeb1 deficiency in DCs selectively reduced the cDC1 population in the spleen but not in the other lymphoid and non-lymphoid tissues.

The scRNA seq data and statements in Fig S2a are confusing. Based on Fig 1g it was said that there was a difference in the amount of cDC1 between WT and Zeb1-dcKO. However, based on the markers that define cDC1 in figure S2a there is no difference between WT and Zeb1-dcKO. How do you account for this discrepancy? Genes for cDC1 should be clearly different than genes for cDC2. You should clarify what genes were used to define cDC1/cDC2 from Fig 1g and which are used in fig S2a.

Response: We defined cDC1/cDC2 using the canonical cDC1 signature genes (*Xcr1*, *Clec9a*, *Irf8*, *CD8a*, *Irgae*, etc.) and cDC2 signature genes (*Sirpa*, *CD209a*, *Irf4*, *Esam*, *Irgam*, *Zeb2*, etc.) (Ref. 32 and 34), and DC3 signature genes (*Ccl22*, *Cxcl16*, *CD83*, *CD86*, *Adam8*, etc.)² as shown in additional Fig. 4. To clearly demonstrate the reduction in frequency of Zeb1-deficient splenic cDC1s, we performed kernel density estimation with our scRNA-seq data using the `plot_density` function from the R package *Nebulosa*

to quantify cells with high expression of indicated signature genes and presented in feature plots with the statistic as shown in supplementary Fig. 1j and Table S1. These results clearly showed that the frequencies of the cells expressing high level of cDC1 signature genes were severely decreased in splenic cDC sample from *Zeb1*-dcKO mice.

To examine the identity of splenic cDC1 and cDC2 from WT and *Zeb1*-dcKO mice, we analyzed the expression level of cDC1 or cDC2 signature genes in single-cell level by using R package Seurat (v4.0) and presented the results in violin plots in new supplementary Fig. 2a. These results revealed that residual *Zeb1*-deficient splenic cDC1s maintained cDC1 identity as they express similar levels of cDC1 and cDC2 signature genes including *Xcr1*, *Clec9a*, *Irf8*, *Sirpa*, *CD209a*, *Irf4*, to that of WT cDC1s, although *CD8a* expression was decreased and *Itgae* (encoding CD103) expression was increased in *Zeb1*-deficient splenic cDC1s.

The annexin/PI (Fig1h) and caspase experiments (figS2d) are conflicting. How many times were each experiment repeated? Cells that have undergone apoptosis will have PS on their surface and hence how annexin binds. So it appears that these cells are undergoing apoptosis. Were the conditions different in any way between experiments? These inconsistencies make one question the conclusions.

Response: We thank the reviewer for raising this interesting point. We repeated Annexin V/7-AAD and Caspase-3 experiments more than three times, and they are repeatable. When detecting apoptotic cells, Annexin V/7-AAD double staining allows a further distinction of late necrotic (Annexin V⁺/7-AAD⁺) from early apoptotic (Annexin V⁺/7-AAD⁻) cells. Although cells that are Annexin V⁺/7-AAD⁻ can be considered to die by apoptosis at early time points, at late time points apoptotic cells can become Annexin V⁺/7-AAD⁺. Actually, it should be also noted that necroptotic cells also exposed phosphatidylserine (PS) prior to loss of plasma membrane integrity so Annexin V can bind to exposed PS or to internal PS following cell rupture^{3,4}.

One of the major observations by the authors is that conditional loss of *Zeb1* in DCs lead to a specific loss of only splenic cDC1 subset, without any change in the other DC subsets from other sources. This is a very provocative finding, but the mechanistic underpinnings have not been addressed by the authors in any capacity. Some experimental insights will greatly improve the value of this work.

Response: We thank the reviewer for the encouraging comment. As for selective reduction of splenic cDC1s, we found it was caused by excessive cell death, although we have not uncovered the mechanism how cDC1 cells are selectively affected in the spleen and not in other lymphoid organs. This is an important and big biological topic. However, we sincerely believe that it is out of the scope of this manuscript. The spleen is the largest second lymphoid organ that specializes in filtering blood and trapping blood-borne pathogens or antigens. cDC1s localize in the marginal zone and red pulp of the spleen where they are readily exposed to blood-borne pathogens and dead cells. Whether excessive death of *Zeb1*-deficient splenic cDC1s is induced by blood-borne stimuli, needs to be further investigated.

4. Figure 2. How is it that loss of *Zeb1* in cDC1 enhances immune defense? Is this result specific to *L. monocytogenes*? Host defense should be tested in another way to confirm the results. What would happen if the opposite were true. Overexpression of *Zeb1* in cDC1 would that cause mice to become over sensitive and die at much lower doses?

Histological analysis needs to be statistically evaluated in order to confirm that images taken were not an exception. How many images were taken per mouse?

Pro-inflammatory cytokines were tested (Fig. 2d) and showed a significant increase with the Zeb1-dcKO mice vs WT. Was the opposite trend observed for anti-inflammatory cytokines? How long were the differences in cytokine expression maintained for?

Response: Previous studies have established that CD8 α ⁺ dendritic cells (cDC1s) in the spleen are the obligate cellular entry points for productive infection by intracellular bacterium *L. monocytogenes*, and lack of cDC1s enhances host resistance to the infection due to the loss of access into periarterial lymphocyte sheath (PALS) (Ref. 39 and 40). Because the splenic cDC1s were dramatically reduced in Zeb1-dcKO mice, *L. monocytogenes* could not migrate to the PALS and could be trapped in the marginal zone and rapidly cleared by phagocytes as reported by ref. 40. The decreased Listeria burden and reduced tissue lesions in spleen and liver of Zeb1-dcKO mice (Fig. 2b-c) pointed out that the bacteria could not successfully reproduce in Zeb1-dcKO mice. These observations were in complete accord with the severe impairment of secretion of both inflammatory and anti-inflammatory cytokines in Zeb1-dcKO mice (Fig. 2d). Therefore, we concluded that the increased resistance to Listeria infection of Zeb1-dcKO mice was likely due to the impaired transmission of Listeria into PALS, resulted from selective reduction of splenic cDC1s, as the innate immune response was intact in Zeb1-deficient cDCs. As suggested by the reviewer, we counted the tissue lesions from the histochemical sections and included the statistic in Fig. 2c.

However, we have not checked *Listerial* infection in DC-specific conditional Zeb1 transgene mice because we have not got enough mice.

In addition to *Listerial* infection, we also infected Zeb1-dcKO mice with LCMV Armstrong strain. At day 8 after infection, the frequencies and absolute numbers of total CD8⁺ T cells and activated (CD44⁺) CD8⁺ T cells were substantially decreased in Zeb1-dcKO mice as compared with that in WT mice (Additional Fig. 3a), while the generation of activated (CD44⁺) CD4⁺ T cells, GC B cells and plasma cells was not significantly affected in Zeb1-dcKO mice (Additional Fig. 3a-b). The selective effect on CD8⁺ T cells was also observed in tumor-infiltrating T cells of tumor-bearing mice (Fig. 3g-j). These results pointed out a potential role of Zeb1 in cross-presentation of cDC1s.

5. Figure 3. There is statistical difference between tumor growth and the survival of WT and Zeb1dcKO mice bearing B16 tumors. Zeb1dcKO have tumors that grow much faster and survival is much lower than WT. So clearly the effect would seem to be occurring within the tumor. However, the levels of cDC1/cDC2 are not different--why? Essentially the differences in B16 tumor growth/survival are attributed to cDC1/cDC2 differences only in the spleen. If this is the case then you need to be very clear about where the differences are actually occurring and how this works mechanistically.

Response: We thank the reviewer for this positive comment. We already demonstrated that Zeb1 deficiency in DCs selectively reduced the cDC1 population in the spleen but not in the other lymphoid and non-lymphoid tissues. This phenotype was observed in both unimmunized and tumor-bearing Zeb1-dcKO mice (Fig. 1a-d, Fig. 3c-f and supplementary Fig. 4a-b). cDC1s are particularly adept at uptake of dead tumor cells and at cross-priming tumor-specific CD8⁺ T cells within tumor microenvironment or after migration to tumor draining lymph nodes (tumor dLNs). Many studies have demonstrated that expansion of cDC1s and their migration back and forth from tumor dLNs to tumor sites are very pivotal for cross-priming tumor-specific CD8⁺ T cells and also for immune checkpoint blockade therapy^{5,6,7,8}. So, it is reasonable to attribute the compromised tumor control in Zeb1-dcKO mice to the defect in cross-presentation of

cDC1s from tumor sites or tumor dLNs but not to the reduced numbers of splenic cDC1s.

The authors have shown a strong decrease in the activation of CD8⁺ CTLs, but have not looked at T-reg or ICOS⁺ CD4⁺ T cells. They may have a direct consequence in the inhibition of maturation of the CD8⁺ CTLs in the dcKO tumors. Further, looking at exhaustion state of CD8⁺ T cells (PD1/Tim3) could also provide insight as to the mechanism of reduced abundance of activated CTLs.

Response: This is a very good point. As suggested by the reviewer, we examined the tumor-infiltrating Treg cells and exhausted tumor-infiltrating T cells by flow cytometry. As shown in new supplementary Fig. 4c-d, the frequencies and numbers of exhausted CD8⁺ T cells but not of exhausted CD4⁺ T cells were severely decreased in tumors of Zeb1-dcKO mice. Although the frequencies of tumor-infiltrating Treg cells were reduced by half in Zeb1-dcKO mice, the absolute cell numbers of tumor-infiltrating Treg cells were similar to that of WT mice (new supplementary Fig. 4e-f). Exhausted T cells result from chronic antigen stimulation. PD-1 and Tim-3 are not only T cell exhaustion markers but also are T cell activation markers whose expressions were induced in naive T cells upon antigen stimulation. Therefore, the defective recruitment and activation of tumor infiltrating CD8⁺ T cells as well as the diminished exhausted CD8⁺ T cells in Zeb1-dcKO mice pointed out a potential role of Zeb1 in cross-presentation.

6. Figure 5-7. The statement that there is a strong reduction of Cybb and slight reduction of Ncf2 in the western blot Fig.5g needs to be accompanied with numerical values. The only obvious differences are with Zeb1. The data from RNAseq, in particular the statements made in line 343-353 need to be validated by another experiment. Manipulation of Zeb1 should be performed to evaluate the phagosome/lysosome pathways changes.

Response: As suggested by the reviewer, we quantified the normalized protein level of Cybb and Ncf2 in cDC1s after stimulation with HKLM-OVA and included the statistic in Fig. 5g. We have also validated the increased protein expression of Ap1b1 and Ap1m2 in Zeb1-deficient cDC1s after stimulation with HKLM-OVA by western blot (new supplementary Fig. 6d).

The Cut&Tag and chip-seq analysis are very interesting. However, no further validation of this data was performed to really narrow down the results and tie them back to the hypothesis. The conclusions need to be validated using a secondary experiment.

Response: We thank the reviewer for pointing out this issue. Although there is no good commercial monoclonal anti-Zeb1 antibody, some studies have already reported genome-wide binding of Zeb1 by Chip-seq using polyclonal anti-Zeb1 antibody (Ref. 46 and 47). Canonical Zeb1-binding motifs were also retrieved from our CUT&Tag data indicating that the result of CUT&Tag was reliable as that of published Zeb1 Chip-seq results. Moreover, our CUT&Tag results identified two Zeb1 binding peaks near the promoter of miR-183-96-182 cluster, consistent with previous reports that two Zeb1-binding motifs were found upstream of human miR-183-96-182 cluster (Ref. 49 and 51).

The targeting of miR-96 needs to be clarified further—was this performed with the miR-96 family cluster or with miR-96 specifically. This should be clarified and updated in the manuscript. Direct regulation of Cybb by m96cl has not been validated by performing WB/q-RT-PCR to look at expression of Cybb after exogenous expression or repression of m96cl in the cells. For example, multiple groups have demonstrated that Zeb1 regulates mRNA's

and miRNA's throughout the genome in a concordant fashion. The authors do not seem to have analyzed the data in an unbiased fashion to understand the mechanistic effects, but rather have chosen miR-96 as a convenient potential mechanistic intermediate.

Response: We thank the reviewer for raising these important issues. miR-96 and miR-182 were predicted to binding to the same site of *Cybb* mRNA, while miR-183 was predicted not to bind either *Cybb* mRNA or *Ncf2* mRNA. Hence, we performed dual luciferase reporter assay to examine the regulation of *Cybb* expression via its 3'UTR by miR-96 or miR-182. Then we examine the protein expression of *Cybb* in Flt3L-cDC1s after retroviral transduction of miR-96 or miR-182. Both experiments suggested that miR-96 overexpression significantly but miR-182 overexpression only slightly suppressed protein expression of *Cybb* (Fig. 5k and supplementary Fig. 7e).

To explore the mechanism by which *Zeb1* regulated cross-presentation, we firstly performed RNA-seq of WT and *Zeb1*-deficient Flt3L-cDC1s after stimulation with HKLM-OVA. GO and GSEA analysis revealed that most of the genes in the pathways of antigen processing and presentation, phagosome and some in the lysosome pathway were down-regulated in *Zeb1*-deficient cDC1s after stimulation. We excluded the genes encoding MHC- I b molecules in the pathway of antigen processing and presentation, as they have functions other than antigen presentation to conventional CD8⁺ T cells. Because *Nox2* activity is absolutely required for cross-presentation of particulate antigens, *Cybb* and *Ncf2* then become strong candidates for *Zeb1* target genes in cross-presentation. However, CUT&Tag experiments did not identify strong *Zeb1* binding signal in in some of DEGs including *Cybb* and *Ncf2* in the in the pathways of antigen processing and presentation, phagosome, suggesting that *Zeb1* might indirectly regulate the transcription of *Cybb* and *Ncf2*. We then performed miRNA-seq analysis of WT and *Zeb1*-deficient cDC1s after stimulation with HKLM-OVA. The intersection of *Zeb1*-bound miRNAs and *Zeb1*-regulated miRNAs revealed that 19 miRNAs were direct targets of *Zeb1*. Among these miRNAs, both miR-96 and miR-182 were predicted to bind to the 3' UTR of *Cybb* mRNA. Then the regulation of *Cybb* expression by miR-96 and miR-182 was validated using dual luciferase reporter assay and western blot. In sum, we identified the *Zeb1*-miR-96/182-*Cybb* regulatory axis in an unbiased fashion through multi-round combined analysis of three types of high-throughput sequencing data.

Although the authors go into mechanistic detail for the *Zeb1*-miR-96-*Cybb* axis as a potential point of control for cDC1 function, it is not soundly established that this is the core mechanism for alteration of antigen presentation. From the sequencing data, alterations in *Zeb1* appear to be rather pleiotropic in nature, with both direct binding and indirect effects across the genome. The focus on just the one indirect effect on *Cybb*, which appears quantitatively weak, is unconvincing. I think that the authors' statement in lines 353-355 is probably correct, but the manuscript does not explore the expression of a series of genes by *Zeb1* that promote cross-presentation.

Response: We agree with the reviewer that *Zeb1* appears to be pleiotropic in regulation of cross-presentation. However, we found that enforced expression of *Cybb* in *Zeb1*-deficient cDC1s completely restored the capability of cross-presentation. Thus, we believe that the *Zeb1*-miR-96/182-*Cybb* axis is the main mechanism by which *Zeb1* regulates cross-presentation in cDC1s.

Additionally, the authors do not demonstrate that the phagolysosomal and antigen export/presentation machinery effects of *Zeb1* found in vitro account for the in vivo infection or tumor phenotypes in any way.

Response: We have demonstrated that the increased resistance to *Listeria* infection of Zeb1-dcKO mice was likely due to selective reduction of splenic cDC1s. And it is reasonable for us to attribute the compromised tumor control in Zeb1-dcKO mice to the defect in cross-presentation of cDC1s at tumor sites or tumor dLNs rather than the reduced numbers of splenic cDC1s.

Quantitation of Fig.7f. In order to make conclusions quantitation of staining must be performed with applied statistics.

Response: As suggested by both reviewer #1 and reviewer #2, we quantified the frequencies of colocalization events within HKLM⁺ cells in the images and included the statistic in Fig. 7f.

7. The manuscript would benefit from improvement in the English usage throughout. This would enhance the readability.

Response: We thank the reviewer for pointing out this important issue. We have tried our best to polish the English writing in the manuscript.

Reviewer #3 (Remarks to the Author):

Manuscript No: NCOMMS-22-52757

Wang et al., "The transcription factor Zeb1 controls homeostasis and function of type 1 conventional dendritic cells"

The authors demonstrate that Zeb1 suppresses miR-96 which in turn suppresses the Cybb subunit of NADPH oxidase 2 (NOX2). NOX2 activity is therefore reduced in Zeb1 deficient classical dendritic cells type 1 (cDC1) reducing their antigen cross-presenting capacity but also presentation of endocytosed antigens on MHC class II molecules. This results in increased splenic cDC1 cell death, their increased resistance to *Listeria* but decreased ability to induce anti-cancer immune responses. Interestingly although ovalbumin (ova) escape to the cytosol is inhibited due to Zeb1 deficiency there is no effect on phagosomal acidification. On the contrary phagocytosed antigen seemed to be more rapidly delivered to lysosomes. From these data the authors suggest that Zeb1 is required to optimize the cross-presentation capacity of cDC1s, but it remains unclear why these cells also present antigens less efficiently on MHC class II and die in the spleen.

This is an interesting study on the role of Zeb1 for cross-presentation by cDC1s but some additional information on why Zeb1 is also required for endocytosed antigen presentation on MHC class II molecules seems to be required.

Response: We thank this reviewer for the positive comments and insightful suggestions to revise this manuscript. We have made extensive efforts to address the raised issues by performing additional experiments and by providing explanations as described below. In this study, we have investigated two roles of Zeb1 in cDC1s with one in homeostasis of splenic cDC1s and the other in cross-presentation of cDC1s. We have demonstrated that the increased resistance to *Listeria* infection of Zeb1-dcKO mice was likely due to selective reduction of splenic cDC1s. And it is reasonable for us to attribute the compromised tumor control in Zeb1-dcKO mice to the defect in cross-presentation of cDC1s at tumor sites or tumor dLNs rather than the reduced numbers of splenic cDC1s.

Major comments:

1. How do Zeb1 deficient cDCs1 influence the survival of Zeb1 positive cDC1s in trans? The authors suggest that Zeb1 deficient cDC1s die and are taken up by Zeb1 positive cDC1s, causing their cell death. Can they also demonstrate this in vitro?

Response: This is a very good point. We did not use *Zeb1*-deficient splenic cDC1s in the in-vitro experiments, because there were very few in the spleen of *Zeb1*-dcKO mice and it was very difficult to purify enough dying cells by fluorescence-activated cell sorting. Instead, we treated WT splenocytes with TNF- α +Smac to induce apoptosis, or with TNF- α +Smac+Zvad to induce necroptosis. And then we cocultured purified WT splenic cDC1s with either live splenocytes or apoptotic splenocytes, or necroptotic splenocytes. As shown in new supplementary Fig. 2h-I, WT splenic cDC1s from B6 mice underwent much more cell death when cocultured with apoptotic or necroptotic splenocytes than live splenocytes. A previous study has reported that cDC1s rather than other myeloid cells can be selectively depleted by apoptosis triggered by translocation of internalized cytochrome c (cyt c) into the cytoplasm (Ref. 59), so we speculated that cDC1s from B6 BM cells might undergo apoptosis following uptake of dying or dead cDC1s from *Zeb1*-deficient BM cells in the spleen of mixed BM chimera, resulting in similar loss of both *Zeb1*-sufficient and -deficient splenic cDC1s.

2. Why does Zeb1 deficiency also decrease MHC class II restricted antigen presentation? The authors argue in their discussion that NOX2 attenuates antigen degradation which they suggest is also beneficial for MHC class II presentation. However, the authors do demonstrate that acidification is intact as well as fusion with lysosomes. Is endocytosed ova less efficiently transported to the MHC class II containing compartment (MIIC) for loading?

3. The authors did not detect changes in phagosomal acidification in Zeb1 deficient cDC1s. Did they detect changes in phagocytosed ovalbumin (ova) maintenance?

4. The authors interpret their proteasome inhibition experiments as indication that Zeb1 deficiencies cripples proteasomal degradation. However, if less ova reaches the cytosol, less is also degraded by the proteasome. The interpretation should be reconsidered.

Response: We thank the reviewer for raising these three critical points. We agree with the reviewer that Zeb1 ablation significantly decreased DQ-OVA fluorescence because less OVA reached the cytosol. Following the reviewer's suggestion, we changed this conclusion as following: Zeb1 ablation significantly decreased DQ-OVA fluorescence, which was blocked by proteasome inhibitor MG132, suggesting that less antigen escaped from phagosomes into the cytosol for proteasome-mediated antigen processing in Zeb1-deficient cDC1s.

Although we have not examined the efficiency of antigen transportation to the MHC class II containing compartment (MIIC), we have investigated phagosomal antigen degradation in individual phagosomes of cDC1s after phagocytosis by using flow organellometry. We found that the kinetics and efficiency of phagosomal OVA degradation were much faster in *Zeb1*-deficient cDC1s than in WT cDC1s after phagocytosis of HKLM-OVA (new Fig. 7i-j). Therefore, the accelerated degradation of phagosomal antigen may also contribute to the impaired presentation of exogenous antigens to CD4⁺ T cells via MHC- II in *Zeb1*-deficient cDC1s.

5. Why does Zeb1 deficiency also decrease CD4+ T cell numbers in the studied tumor model (B16F10)? Do cDC2s not compensate for cDC1s during CD4+ T cell priming? This should at least be discussed.

Response: We thank the reviewer for raising this issue. When we enumerated tumor-infiltrating immune cells, we always normalized the total cell number by dividing it by the corresponding tumor weight from tumor-bearing mouse. Since the tumor weight was higher in Zeb1-dcKO mice than in WT mice, after such normalization, the difference of tumor-infiltrating CD4⁺ T cell numbers between WT and Zeb1-dcKO mice was increased from twofold to threefold (additional Fig. 5, main Fig. 3g-h). Anyway, Zeb1 deletion in DCs did decrease CD4⁺ T cell numbers in B16F10 tumors. We agree with the reviewer that cDC2s in tumors did not compensate for cDC1s in CD4⁺ T cell priming in tumor-bearing Zeb1-dcKO mice, probably because the cDC2 numbers were not dominant over cDC1 numbers at tumor sites as they did in spleen (Fig. 3c-d).

Additional Figure 5. Absolute numbers of tumor-infiltrating CD4⁺ or CD8⁺ T cells in tumors from WT and Zeb1-dcKO mice at day 12 after s.c. injection with 2×10⁵ B16F10 melanoma cells.

Minor comments:

1. Some typos, e.g. line 533 and 607, deficient instead of deficient,

Response: We apologize for these spelling mistakes. We have corrected them.

Additional References

1. Guan, T. *et al.* ZEB1, ZEB2, and the miR-200 family form a counterregulatory network to regulate CD8(+) T cell fates. *J Exp Med* **215**, 1153-1168 (2018).
2. Maier, B. *et al.* A conserved dendritic-cell regulatory program limits antitumour immunity. *Nature* **580**, 257-262 (2020).
3. Shlomovitz, I., Speir, M. & Gerlic, M. Flipping the dogma - phosphatidylserine in non-apoptotic cell death. *Cell Commun Signal* **17**, 139 (2019).

4. Gong, Y.N. *et al.* ESCRT-III Acts Downstream of MLKL to Regulate Necroptotic Cell Death and Its Consequences. *Cell* **169**, 286-300 e216 (2017).
5. Salmon, H. *et al.* Expansion and Activation of CD103(+) Dendritic Cell Progenitors at the Tumor Site Enhances Tumor Responses to Therapeutic PD-L1 and BRAF Inhibition. *Immunity* **44**, 924-938 (2016).
6. Roberts, E.W. *et al.* Critical Role for CD103(+)/CD141(+) Dendritic Cells Bearing CCR7 for Tumor Antigen Trafficking and Priming of T Cell Immunity in Melanoma. *Cancer Cell* **30**, 324-336 (2016).
7. Spranger, S., Dai, D., Horton, B. & Gajewski, T.F. Tumor-Residing Batf3 Dendritic Cells Are Required for Effector T Cell Trafficking and Adoptive T Cell Therapy. *Cancer Cell* **31**, 711-723 e714 (2017).
8. Schenkel, J.M. *et al.* Conventional type I dendritic cells maintain a reservoir of proliferative tumor-antigen specific TCF-1(+) CD8(+) T cells in tumor-draining lymph nodes. *Immunity* **54**, 2338-2353 e2336 (2021).

REVIEWER COMMENTS

Reviewer #1 (Remarks to the Author):

Wang et al. revised

The authors have included new data in the revised manuscript that address some of the comments that were put forth in the initial review. However, there are a few points that remain either unaddressed or unclear. The manuscript text should be modified to take this into consideration:

1. The data presented in Figure 1 with the bone marrow chimeras remain odd. The mechanism of how the wild-type cDC1 are affected in this competitive setting is speculative and suggestive, not conclusive. The text in Line 220 should be modified to reflect this. The same language should be applied in regards to the splenic cDC1 reduction in Zeb1 KO animals. The abstract and results should not state that the cDC1 reduction in Zeb1 KO is "due to" excessive cell death as this is not formally proven.
2. The modified text in Lines 311-313 still needs rewording. The authors cannot exclude the possibility that Zeb1 KO mice have deficiencies in antitumour responses (in vivo cross priming) due to deficits in cDC1 numbers.
3. The authors should highlight that cDC2 numbers were different in WT vs. Zeb1 KO mice (Fig. 3e-f) and address whether this might affect the interpretation of the antitumour data.
4. The authors did not address whether endogenous processing of antigens is faster/slower in Zeb1 knockout cDCs. This point can be made in the discussion.

Reviewer #2 (Remarks to the Author):

In the revised manuscript from Wang et al., the authors investigate the specific role of the Zeb1 transcription factor in Type I conventional dendritic cells by using a DC-specific conditional KO mouse of Zeb1. They stress three main findings in the manuscript: 1) that cDC1 homeostasis is affected ONLY in the spleen of the animals and not in any other lymphoid, normal, or tumor tissue; 2) that the Zeb1-deficient cDC1s have defective cross-presentation of antigens due to altered phagosomal ROS-induced rupture and enhanced lysosomal trafficking of antigens; and 3) discovery of a regulatory axis via Zeb1-miR-96-Cybb that controls the cross presentation pathway. The revised manuscript presents a large amount of work, with many additions during the revision process, that is generally performed well. Many of the points have been firmed up to better support their conclusions, with the strongest work in the manuscript being the defect in cross-presentation of antigens due to the altered phagosomal ROS-induced rupture. However, I still find that the manuscript suffers from being too broad and speculative at points, which makes portions of it difficult to read and follow. As noted by Reviewer 1, this could really be reorganized into two distinct manuscripts that are more focused and better explain the findings. Unfortunately, the in vivo observations of the spleen-specific homeostasis and the tumor cell effects are still without mechanistic explanation, thus providing no strong connection between many of the in vitro cell biologic experiments and the in vivo experiments. Although the Discussion section has been revised, it still relies heavily on supposition or with caveats, and thus comes across to the reader as highly speculative rather than being based on solid findings.

Major points:

1. The several main themes of the Results do not fit well together as a coherent set of findings and leave a number of observations incompletely explained. Importantly, it is not clear why cDC1 cells are selectively affected in the spleen and not in other lymphoid organs. The authors do not believe that this is something that should be addressed in the manuscript, but the reader is still left wondering how excessive non-apoptotic cDC1 cell death occurs and why it only occurs in the spleen.

Additionally, it is unclear if this observation is connected with the antigen cross-presentation phenotype or is distinct, and whether the spleen-specific homeostasis has any relationship with the

Zeb1/miR-96 axis described later in the paper.

2. The question of how B16F10 melanoma tumors in the Zeb1dcko syngeneic animals grow much faster and with worse outcome than WT is not fully addressed and the findings raise confusion. In their reply the authors attribute this to cDC1s in the tumor microenvironment or draining lymph nodes, but per their own data (in Figure 3) there are no differences in the numbers or composition of cDCs in the tumors or dLNs. There is no mechanistic support to explain how the differences in B16 tumor growth/survival due to cytotoxic CD8 T cells are attributed to cDC1 differences in the spleen, or alternatively to effectively demonstrate that cross-presentation of tumor-specific antigens by cDC1s in the tumors or tumor draining lymph nodes is altered.

Reviewer #3 (Remarks to the Author):

Manuscript Nr: NCOMMS-22-52757A

Wang et al., "The transcription factor Zeb1 controls homeostasis and function of type 1 conventional dendritic cells"

The authors demonstrate that Zeb1 suppresses miR-96 which in turn suppresses the Cybb subunit of NADPH oxidase 2 (NOX2). NOX2 activity is therefore reduced in Zeb1 deficient classical dendritic cells type 1 (cDC1) reducing their antigen cross-presenting capacity but also presentation of endocytosed antigens on MHC class II molecules. This results in increased splenic cDC1 cell death, their increased resistance to Listeria but decreased ability to induce anti-cancer immune responses. Interestingly although ovalbumin (ova) escape to the cytosol is inhibited due to Zeb1 deficiency there is no effect on phagosomal acidification. On the contrary phagocytosed antigen seemed to be more rapidly delivered to lysosomes. From these data the authors suggest that Zeb1 is required to optimize the cross-presentation capacity of cDC1s, but it remains unclear why these cells also present antigens less efficiently on MHC class II and die in the spleen.

In their revised manuscript version, the authors have addressed some of my concerns. Primarily they demonstrate that DQ-OVA is degraded much faster in phagosomes of Zeb1 deficient cDC1s, presumably not allowing efficient delivery to MIICs and escape to the cytosol for MHC class II presentation and cross-presentation on MHC class I molecules, respectively. The author, however, provide no explanation why cDC2s do not compensate for cDC1s to stimulate CD4+ T cells or why Zeb1 deficient cDC1s undergo more apoptosis. Nevertheless, the manuscript is improved.

Minor comments:

1. In order to substantiate that elevated lysosomal degradation of phagocytosed antigens is detrimental to MHC class II presentation, the authors could cite early literature on attenuation of lysosomal activity that enhanced MHC class II presentation, such as Delamarre et al., Science 2005; Delamarre et al., J Exp Med 2006; Brazil et al., Eur J Immunol 1997)

Point by point response to the reviewers' comments

We are grateful to the reviewers for their further review of our revised manuscript and for their positive and constructive comments. Following the suggestions by the reviewers, we have performed new experiments and provided more explanation in the new revised manuscript. In particular, we have examined antitumor immunity of WT and Zeb1-dcKO mice after splenectomy (new Supplementary Fig. 5). We have also assessed endogenous antigen processing through electroporation of DQ-OVA (new Supplementary Fig. 8g-h). The major changes are highlighted in a red underlined font in the new revised manuscript, including the legends of the new supplementary figures. With these new results, we believe we have addressed almost all concerns from the reviewers, and improved the quality of our work.

REVIEWER COMMENTS

Reviewer #1 (Remarks to the Author):

Wang et al. revised

The authors have included new data in the revised manuscript that address some of the comments that were put forth in the initial review. However, there are a few points that remain either unaddressed or unclear. The manuscript text should be modified to take this into consideration:

Response: We thank reviewer #1 for acknowledging the effort done to follow his/her advises to improve the manuscript.

1. The data presented in Figure 1 with the bone marrow chimeras remain odd. The mechanism of how the wild-type cDC1 are affected in this competitive setting is speculative and suggestive, not conclusive. The text in Line 220 should be modified to reflect this. The same language should be applied in regards to the splenic cDC1 reduction in Zeb1 KO animals. The abstract and results should not state that the cDC1 reduction in Zeb1 KO is “due to” excessive cell death as this is not formally proven.

Response: We thank the reviewer for raising this question. Following the reviewer's suggestion, we replaced “due to” with “accompanied by” or “associated with” to describe the relation of cell death and splenic cDC1 reduction in the abstract, introduction, results, and discussion including the text in line 220-221.

2. The modified text in Lines 311-313 still needs rewording. The authors cannot exclude the possibility that Zeb1 KO mice have deficiencies in antitumour responses (in vivo cross priming) due to deficits in cDC1 numbers.

Response: We thank both reviewers #1 and #2 for raising this important question. To exclude the influence of deficits in splenic cDC1 numbers on the antitumor T cell response in *Zeb1*-dcKO mice, we performed splenectomy before tumor inoculation. Interestingly, splenectomy slightly inhibited B16F10 melanoma growth in both WT and *Zeb1*-dcKO mice, consistent with an early study. However, splenectomised *Zeb1*-dcKO mice still had compromised tumor control compared to splenectomised WT mice (new Supplementary Fig. 5a). Moreover, splenectomy neither abrogated the difference of antitumor T cell response between WT and *Zeb1*-dcKO mice, nor hindered infiltration of cDC1s into tumors (new Supplementary Fig. 5a-g). So, we attribute the compromised tumor control in *Zeb1*-dcKO mice to the defective function of cDC1s from tumor sites or tumor dLNs but not to the reduced numbers of splenic cDC1s.

3. The authors should highlight that cDC2 numbers were different in WT vs. *Zeb1* KO mice (Fig. 3e-f) and address whether this might affect the interpretation of the antitumour data.

Response: We thank the reviewer for pointing out this issue. We add a statement “but there were slightly fewer resident cDC2s in tumor dLNs of *Zeb1*-dcKO mice” in line 279-280. We think this is a mild phenotype, because all other populations of both cDC1 and cDC2 presented normally in other lymphoid tissues (except for spleen) and non-lymphoid tissues of unimmunized *Zeb1*-dcKO mice, also in tumor and dLNs of tumor-bearing *Zeb1*-dcKO mice. More importantly, the defect of antitumor CD4⁺ T cell responses in tumor-bearing *Zeb1*-dcKO mice were milder than the defect of antitumor CD8⁺ T cell responses (new Supplementary Fig. 5a-g).

4. The authors did not address whether endogenous processing of antigens is faster/slower in *Zeb1* knockout cDCs. This point can be made in the discussion.

Response: We apologize that we did not assess endogenous antigen processing by proteasome in the first revision of revised manuscript. As suggested by reviewer #1, we introduced DQ-OVA plus Alexa Fluor 647-OVA directly into the cytosol of cDC1s by electroporation, and analyzed the fluorescence of both DQ-OVA and Alexa Fluor 647-OVA after chasing by flow cytometry. The ratios of percentage and intensity of DQ-OVA fluorescence to that of Alexa Fluor 647 fluorescence were slightly higher in *Zeb1*-deficient cDC1s than in WT counterparts, suggesting that *Zeb1*-deficient cDC1s had a little stronger capacity to degrade cytosolic antigens than WT counterparts (new Supplementary Fig. 8g-h). Thus, we can conclude that the defective cross-presentation in *Zeb1*-deficient cDC1s was not caused by endogenous antigen processing.

Reviewer #2 (Remarks to the Author):

In the revised manuscript from Wang et al., the authors investigate the specific role of the Zeb1 transcription factor in Type I conventional dendritic cells by using a DC-specific conditional KO mouse of Zeb1. They stress three main findings in the manuscript: 1) that cDC1 homeostasis is affected ONLY in the spleen of the animals and not in any other lymphoid, normal, or tumor tissue; 2) that the Zeb1-deficient cDC1s have defective cross-presentation of antigens due to altered phagosomal ROS-induced rupture and enhanced lysosomal trafficking of antigens; and 3) discovery of a regulatory axis via Zeb1-miR-96-Cybb that controls the cross-presentation pathway. The revised manuscript presents a large amount of work, with many additions during the revision process, that is generally performed well. Many of the points have been firmed up to better support their conclusions, with the strongest work in the manuscript being the defect in cross-presentation of antigens due to the altered phagosomal ROS-induced rupture. However, I still find that the manuscript suffers from being too broad and speculative at points, which makes portions of it difficult to read and follow. As noted by Reviewer 1, this could really be reorganized into two distinct manuscripts that are more focused and better explain the findings. Unfortunately, the in vivo observations of the spleen-specific homeostasis and the tumor cell effects are still without mechanistic explanation, thus providing no strong connection between many of the in vitro cell biologic experiments and the in vivo experiments. Although the Discussion section has been revised, it still relies heavily on supposition or with caveats, and thus comes across to the reader as highly speculative rather than being based on solid findings.

Response: We thank reviewer #2 for acknowledging the effort done to follow his/her advises to improve the manuscript and also for the constructive criticisms. In this study, we have investigated two roles of Zeb1 in cDC1s with one in homeostasis of splenic cDC1s and the other in cross-presentation of cDC1s. Although we did not answer why cDC1s are selectively reduced in the spleen of Zeb1-dcKO mice, we found that this reduction was associated with excessive cell death. Based on the guidance from the editorial office, we did not divide this study into two separate manuscripts.

Major points:

1. The several main themes of the Results do not fit well together as a coherent set of findings and leave a number of observations incompletely explained. Importantly, it is not clear why cDC1 cells are selectively affected in the spleen and not in other lymphoid organs. The authors do not believe that this is something that should be addressed in the manuscript, but the reader is still left wondering how excessive non-apoptotic cDC1 cell death occurs and why it only occurs in the spleen.

Response: We thank the reviewer for the constructive criticisms. We agree with the reviewer that it should be addressed why cDC1s were selectively affected in the spleen of *Zeb1*-dcKO mice. However, this is a very challenging question and will be a long-time story. We have tried to obtain useful information from the sc-RNA-seq data of splenic cDC samples from WT and *Zeb1*-dcKO mice, but, due to the limitation in sequencing depth, we can only conclude that several death pathways were activated in *Zeb1*-deficient splenic cDC1s. We confirmed that *Zeb1*-deficient splenic cDC1s underwent more non-apoptotic cell death by flow cytometry. It has been reported that E-cadherin, the most important target of *Zeb1*, augmented death receptor clustering and assembly of DISC, thus enhanced sensitivity to death receptor-mediated apoptosis¹. We crossed *Cdh1*^{fl/fl} mice to *Zeb1*^{fl/fl} CD11c-Cre to generate double conditional KO mice (*Zeb1*/*Cdh1*-dcDKO). We found that *Zeb1*/*Cdh1*-dcDKO also had selective reduction in splenic cDC1s, similar to *Zeb1*-dcKO mice (additional Fig. 6). It suggested that genetic deletion of E-cadherin did not rectify the reduction of splenic cDC1s in *Zeb1*-dcKO mice.

The spleen is the largest second lymphoid organ, specializing in filtering blood and trapping blood-borne pathogens or antigens. Impaired position, retention of cDC2s or the mechano-sensing of blood cells in marginal zone bridging channel of the spleen cause selective loss of splenic cDC2s (Ref. 11 and 12). Whether the reduction of splenic cDC1s in *Zeb1*-dcKO mice was also caused by similar mechanism, remains to be intensively investigated. Anyway, we are going to perform bulk RNA-seq using splenic cDC1 samples from WT and *Zeb1*-dcKO mice and try to find any *Zeb1* target involved in either mechano-sensing or chemoattraction. We will then perform CRISPR-Cas9 screening with the candidate genes by making BM chimeras reconstituted with Cas9-expressing BM cells with gene-specific sgRNA.

Additional Figure 6. Deletion of E-cadherin did not rectify the reduction of splenic cDC1s in Zeb1-dcKO mice. (a) Flow cytometry of live Lin⁻CD317⁻Ly6C⁻CD11c⁺MHC II⁺ cDCs in spleen from WT, Zeb1-dcKO and Zeb1/Cdh1-dcDKO mice (Lin⁻=CD3⁻CD19⁻ hereinafter). Numbers adjacent to outlined areas indicate percent XCR1⁺SIRP α ⁻ cDC1s or XCR1⁻SIRP α ⁺ cDC2s. (b) Frequencies (among CD11c⁺MHC II⁺ cDC) and numbers of cDC1s and cDC2s in the spleen from mice as in a (WT, n=4; Zeb1-dcKO, n=3; Zeb1/Cdh1-dcDKO, n=2).

Additionally, it is unclear if this observation is connected with the antigen cross-presentation phenotype or is distinct, and whether the spleen-specific homeostasis has any relationship with the Zeb1/miR-96 axis described later in the paper.

Response: We agree with the reviewer that it is still unclear if excessive non-apoptotic splenic cDC1 cell death is connected with defective cross-presentation in *Zeb1*-deficient cDC1s. Anyway, we are crossing Zeb1-dcKO mice to miR-183-96-182 KO mice. If genetic ablation of miR-183-96-182 cluster could restore the reduced splenic cDC1 numbers in Zeb1-dcKO mice, the Zeb1-miR-96/182-Cybb regulatory axis might also control splenic cDC1 homeostasis. If not, these two phenotypes might be disconnected and under control of different targets of Zeb1.

2. The question of how B16F10 melanoma tumors in the Zeb1dcKO syngeneic animals grow much faster and with worse outcome than WT is not fully addressed and the findings raise confusion. In their reply the authors attribute this to cDC1s in the tumor microenvironment or draining lymph nodes, but per their own data (in Figure 3) there are no differences in the numbers or composition of cDCs in the tumors or dLNs. There is no mechanistic support to explain how the differences in B16 tumor growth/survival due to cytotoxic CD8 T cells are attributed to cDC1 differences in the spleen, or alternatively to effectively demonstrate that cross-presentation of tumor-specific antigens by cDC1s in the tumors or tumor draining lymph nodes is altered.

Response: We thank both reviewers #1 and #2 for raising this important question. To exclude the influence of deficits in splenic cDC1 numbers on the antitumor T cell response in Zeb1-dcKO mice, we performed splenectomy before tumor inoculation. Interestingly, splenectomy slightly inhibited B16F10 melanoma growth in both WT and Zeb1-dcKO mice, consistent with an early study. However, splenectomised Zeb1-dcKO mice still had compromised tumor control compared to splenectomised WT mice (new Supplementary Fig. 5a). Moreover, splenectomy neither abrogated the difference of antitumor T cell response between WT and Zeb1-dcKO mice, nor hindered infiltration of cDC1s into tumors (new Supplementary Fig. 5a-g). So, we attribute the compromised tumor control in Zeb1-dcKO mice to the defective function of cDC1s from tumor sites or tumor dLNs but not to the reduced numbers of splenic cDC1s.

Reviewer #3 (Remarks to the Author):

Manuscript No: NCOMMS-22-52757A

Wang et al., "The transcription factor Zeb1 controls homeostasis and function of type 1 conventional dendritic cells"

The authors demonstrate that Zeb1 suppresses miR-96 which in turn suppresses the Cybb subunit of NADPH oxidase 2 (NOX2). NOX2 activity is therefore reduced in Zeb1 deficient classical dendritic cells type 1 (cDC1) reducing their antigen cross-presenting capacity but also presentation of endocytosed antigens on MHC class II molecules. This results in increased splenic cDC1 cell death, their increased resistance to *Listeria* but decreased ability to induce anti-cancer immune responses. Interestingly although ovalbumin (ova) escape to the cytosol is inhibited due to Zeb1 deficiency there is no effect on phagosomal acidification. On the contrary phagocytosed antigen seemed to be more rapidly delivered to lysosomes. From these data the authors suggest that Zeb1 is required to optimize the cross-presentation capacity of cDC1s, but it remains unclear why these cells also present antigens less efficiently on MHC class II and die in the spleen.

In their revised manuscript version, the authors have addressed some of my concerns. Primarily they demonstrate that DQ-OVA is degraded much faster in phagosomes of Zeb1 deficient cDC1s, presumably not allowing efficient delivery to MHCs and escape to the cytosol for MHC class II presentation and cross-presentation on MHC class I molecules, respectively. The author, however, provide no explanation why cDC2s do not compensate for cDC1s to stimulate CD4⁺ T cells or why Zeb1-deficient cDC1s undergo more apoptosis. Nevertheless, the manuscript is improved.

Response: We thank reviewer #3 for acknowledging the effort done to follow his/her advises to improve the manuscript.

Although *Zeb1*-deficient cDC2s exhibited intact capability to present exogenous antigens to CD4⁺ T cells, they failed to fully compensate for cDC1s during CD4⁺ T cell priming in tumor-bearing *Zeb1*-dcKO mice, probably because cDC2s were not many more than cDC1s at tumor sites as they did in the spleen (Fig. 3c-d and Supplementary Fig. 4a-b). Actually, the defect of antitumor CD4⁺ T cell responses in tumor-bearing *Zeb1*-dcKO mice were milder than the defect of antitumor CD8⁺ T cell responses (Fig. 3g-j and new Supplementary Fig. 5d-g).

We thank both reviewers #2 and #3 for raising similar question. We agree with the reviewer that it should be addressed why cDC1s were selectively affected in the spleen of Zeb1-dcKO mice. However, this is a very challenging question and will be a long-time story. We have tried to obtain useful information from the sc-RNA-seq data of splenic cDC samples from WT and Zeb1-dcKO mice, but, due to the limitation in sequencing depth, we can only conclude that several death pathways were activated in Zeb1-deficient splenic cDC1s. We confirmed that Zeb1-deficient splenic cDC1s underwent more non-apoptotic cell death by flow cytometry. It has been reported that E-cadherin, the most important target of Zeb1, augmented death receptor clustering and assembly of DISC, thus enhanced sensitivity to death receptor-mediated apoptosis¹. We crossed *Cdh1*^{fl/fl} mice to Zeb1^{fl/fl} CD11c-Cre to generate double conditional KO mice (Zeb1/Cdh1-dcDKO). We found that Zeb1/Cdh1-dcDKO also had selective reduction in splenic cDC1s, similar to Zeb1-dcKO mice (additional Fig. 6, see response to question 1 of reviewer #2). It suggested that genetic deletion of E-cadherin did not rectify the reduction of splenic cDC1s in Zeb1-dcKO mice.

The spleen is the largest second lymphoid organ, specializing in filtering blood and trapping blood-borne pathogens or antigens. Impaired position, retention of cDC2s or the mechano-sensing of blood cells in marginal zone bridging channel of the spleen cause selective loss of splenic cDC2s (Ref. 11 and 12). Whether the reduction of splenic cDC1s in Zeb1-dcKO mice was also caused by similar mechanism, remains to be intensively investigated. Anyway, we are going to perform bulk RNA-seq using splenic cDC1 samples from WT and Zeb1-dcKO mice and try to find any Zeb1 target involved in either mechano-sensing or chemoattraction. We will then perform CRISPR-Cas9 screening with the candidate genes by making BM chimeras reconstituted with Cas9-expressing BM cells with gene-specific sgRNA.

Minor comments:

1. In order to substantiate that elevated lysosomal degradation of phagocytosed antigens is detrimental to MHC class II presentation, the authors could cite early literature on attenuation of lysosomal activity that enhanced MHC class II presentation, such as Delamarre et al., Science 2005; Delamarre et al., J Exp Med 2006; Brazil et al., Eur J Immunol 1997)

Response: We thank the reviewer for referring to these interesting papers. We have followed the suggestion and cited the first two in the discussion section (Line 659-660) of the second version of revised manuscript. We did not include the last one because it talked about excessive degradation of endogenous antigen rather than exogenous antigen.

Additional References

1. Lu, M. *et al.* E-cadherin couples death receptors to the cytoskeleton to regulate apoptosis. *Mol Cell* **54**, 987-998 (2014).

REVIEWERS' COMMENTS

Reviewer #1 (Remarks to the Author):

The authors have largely addressed my comments.

Reviewer #2 (Remarks to the Author):

The authors have attempted to address the prior concerns raised during review in this new round of revisions. The manuscript and its interpretation have been improved and it is suitable for publication.

Reviewer #3 (Remarks to the Author):

Manuscript Nr: NCOMMS-22-52757B

Wang et al., "The transcription factor Zeb1 controls homeostasis and function of type 1 conventional dendritic cells"

The authors demonstrate that Zeb1 suppresses miR-96 which in turn suppresses the Cybb subunit of NADPH oxidase 2 (NOX2). NOX2 activity is therefore reduced in Zeb1 deficient classical dendritic cells type 1 (cDC1) reducing their antigen cross-presenting capacity but also presentation of endocytosed antigens on MHC class II molecules. This results in increased splenic cDC1 cell death, their increased resistance to *Listeria* but decreased ability to induce anti-cancer immune responses. Interestingly although ovalbumin (ova) escape to the cytosol is inhibited due to Zeb1 deficiency there is no effect on phagosomal acidification. On the contrary phagocytosed antigen seemed to be more rapidly delivered to lysosomes. From these data the authors suggest that Zeb1 is required to optimize the cross-presentation capacity of cDC1s, but it remains unclear why these cells also present antigens less efficiently on MHC class II and die in the spleen.

In their revised manuscript version, the authors have addressed some of my concerns. Primarily they demonstrate that DQ-OVA is degraded much faster in phagosomes of Zeb1 deficient cDC1s, presumably not allowing efficient delivery to MIICs and escape to the cytosol for MHC class II presentation and cross-presentation on MHC class I molecules, respectively. The author, however, provide no explanation why cDC2s do not compensate for cDC1s to stimulate CD4+ T cells or why Zeb1 deficient cDC1s undergo more apoptosis.

In their second revision the authors have incorporated the additional references that I suggested as minor comment. Therefore, the manuscript is further improved.

Point by point response to the reviewers' comments

We are grateful to the reviewers for their positive and encouraging comments on our revised manuscript. As suggested by reviewer#2, we include more discussion on the mild reduction of tumor infiltrating effector CD4⁺ T cells and selective splenic cDC1 death in Zeb1-dcKO mice. The major changes are highlighted in red font in the new revised manuscript.

REVIEWER COMMENTS

Reviewer #1 (Remarks to the Author):

Wang et al. revised

The authors have largely addressed my comments.

Response: We thank the reviewer for the positive comments and for acknowledging the effort done to follow his/her advises to improve the manuscript.

Reviewer #2 (Remarks to the Author):

The authors have attempted to address the prior concerns raised during review in this new round of revisions. The manuscript and its interpretation have been improved and it is suitable for publication.

Response: We thank the reviewer for the encouraging comments and the suggestions to improve the manuscript.

Reviewer #3 (Remarks to the Author):

Manuscript No: NCOMMS-22-52757A

Wang et al., "The transcription factor Zeb1 controls homeostasis and function of type 1 conventional dendritic cells"

The authors demonstrate that Zeb1 suppresses miR-96 which in turn suppresses the Cybb subunit of NADPH oxidase 2 (NOX2). NOX2 activity is therefore reduced in Zeb1 deficient classical dendritic cells type 1 (cDC1) reducing their antigen cross-presenting capacity but also presentation of endocytosed antigens on MHC class II molecules. This results in increased splenic cDC1 cell death, their increased resistance to Listeria but decreased ability to induce anti-cancer immune responses. Interestingly although ovalbumin (ova) escape to the cytosol is inhibited due to Zeb1 deficiency there is no effect on phagosomal acidification. On the contrary phagocytosed antigen

seemed to be more rapidly delivered to lysosomes. From these data the authors suggest that Zeb1 is required to optimize the cross-presentation capacity of cDC1s, but it remains unclear why these cells also present antigens less efficiently on MHC class II and die in the spleen.

In their revised manuscript version, the authors have addressed some of my concerns. Primarily they demonstrate that DQ-OVA is degraded much faster in phagosomes of Zeb1 deficient cDC1s, presumably not allowing efficient delivery to MHCs and escape to the cytosol for MHC class II presentation and cross-presentation on MHC class I molecules, respectively. The author, however, provide no explanation why cDC2s do not compensate for cDC1s to stimulate CD4⁺ T cells or why Zeb1 deficient cDC1s undergo more apoptosis.

In their second revision the authors have incorporated the additional references that I suggested as minor comment. Therefore, the manuscript is further improved.

Response: We thank the reviewer for the positive comments and for acknowledging the effort done to follow his/her advises to improve the manuscript. We are happy to convinced the reviewer that the accelerated degradation of phagosomal antigen may account for the impaired presentation of exogenous antigens to CD4⁺ T cells via MHC-II in Zeb1-deficient cDC1.

Following the reviewer's suggestion, we include more discussion to explain the mild reduction of tumor infiltrating effector CD4⁺ T cells and selective splenic cDC1 death in Zeb1-dcKO mice as below.

Most splenic cDC1 localize in the white pulp or marginal zone and red pulp of the spleen, where they are readily exposed to blood-borne pathogens and dead cells and require appropriate mechano-sensing or chemoattraction. It has been reported that E-cadherin, the most important target of Zeb1, augment death receptor clustering and assembly of DISC, thus enhance sensitivity to death receptor-mediated apoptosis. Whether excessive death of Zeb1-deficient splenic cDC1 is induced by these blood-borne stimuli, or aberrant mechano-sensing, chemoattraction or cell adhesion, needs to be further investigated.

Although Zeb1-deficient cDC2 exhibited intact capability to present exogenous antigens to CD4⁺ T cells, they failed to fully compensate for cDC1 during CD4⁺ T cell priming in tumor-bearing Zeb1-dcKO mice, resulting in mild reduction of effector CD4⁺ T cells, probably because cDC2 did not overwhelm cDC1 at tumor sites as they did in the spleen (Fig. 3c-d and Supplementary Fig. 4a-b). Actually, the defect of antitumor CD4⁺ T cell responses in tumor-bearing Zeb1-dcKO mice were milder than the defect of antitumor CD8⁺ T cell responses (Fig. 3g-j and new Supplementary Fig. 5d-g).